# Detection of cell-type-specific risk-CpG sites in epigenome-wide association studies

Xiangyu Luo[1,2], Can Yang[3] & Yingying Wei [2]

In epigenome-wide association studies, the measured signals for each sample are a mixture of methylation profiles from different cell types. Current approaches to the association detection claim whether a cytosine-phosphate-guanine (CpG) site is associated with the phenotype or not at aggregate level and can suffer from low statistical power. Here, we propose a statistical method, HIgh REsolution (HIRE), which not only improves the power of association detection at aggregate level as compared to the existing methods but also enables the detection of risk-CpG sites for individual cell types.

[1] Institute of Statistics and Big Data, Renmin University of China, 100872 Beijing, China. [2] Department of Statistics, The Chinese University of Hong Kong, Hong Kong SAR, China. [3] Department of Mathematics, The Hong Kong University of Science and Technology, Hong Kong SAR, China. Correspondence and requests for materials should be addressed to C.Y. (email: macyang@ust.hk) or to Y.W. (email: yweicuhk@gmail.com)

Epigenome-wide association studies (EWAS) aim to identify cytosine-phosphate-guanine (CpG) sites associated with phenotypes of interest, such as disease status[1–3], smoking history[4,5], body mass index[6], and age[7,8]. However, because the samples in EWAS are measured at the bulk level rather than at the single-cell level, the obtained methylome for each sample shows the signals aggregated from distinct cell types[3,9,10], which leads to two main challenges in the analysis of EWAS data. On the one hand, the cell type compositions differ among samples and can be associated with phenotypes[3,10]. Both binary phenotypes, such as the diseased or normal status[3], and continuous phenotypes, such as age[10], have been found to affect the cell type compositions. As a result, ignoring the cellular heterogeneity in EWAS can lead to many spurious associations[10–13]. On the other hand, the phenotype may change the methylation level of a CpG site in some but not all of the cell types. Identification of the exact cell types that carry the risk-CpG sites can deepen our understandings of disease mechanisms. However, such identification is challenging because only the aggregated-level signals can be observed.

To the best of our knowledge, no existing statistical method for EWAS can detect cell-type-specific associations despite active research to account for cell-type heterogeneity. The existing approaches can be categorized into two schools[14]: reference-based and reference-free methods. The reference-based methods[9,15] require the reference methylation profiles for each cell type to be known a priori, and they regress the aggregated methylation levels observed from each sample on the same set of references to learn the sample's cellular compositions. However, because samples have different attributes, such as age and gender, the methylation levels of a given cell type can vary among samples. It is thus problematic to assume that all of the samples have the same set of reference profiles[10,14]. Furthermore, high-quality references are difficult to obtain for most EWAS due to the existence of unknown cell types, the high cost of cell sorting, and confounding effects[14]. Consequently, a large amount of recent EWAS literature was devoted to identification of risk-CpG sites without the need for the reference methylation profiles.

The reference-free methods can generally be further divided into two classes according to whether they estimate the cell-type mixing proportions directly. The direct-decomposition-based procedures consist of two stages. In the first stage, they simultaneously estimate the cellular compositions of each sample and the cell-type-specific reference methylomes via quadratic programming[16]; and in the second stage, they treat the estimated cell-type proportions as covariates with additive effects in the linear models to conduct association tests. However, when estimating cellular compositions during the first stage, the direct-decomposition-based methods also do not consider samples' phenotype information, thus suffering from the same problem of biasing the cellular composition estimates as the reference-based approaches[9]. Moreover, similar to tumor purity[17], we argue that the estimated cellular composition has a multiplicative rather than an additive effect on the observed methylation level (Methods). The second class of methods, which is exemplified by SVA[18], RefFreeEWAS[19], and ReFACTor[13], does not carry out cell-type decompositions. They resort to singular value decomposition, which includes the principal component analysis, to construct surrogates for the underlying cell-type composition. EWASher, a linear mixed model, also belongs to this class because it is equivalent to the use of principal components as fixed-effect covariates[11]. However, the use of principal components as the covariates in the regression undergoes the same issue of additive effects as the direct-decomposition-based methods. Therefore, the existing reference-free methods have low power in detecting risk-CpG sites[12].

Although the existing methods aim to address the cellular heterogeneity problem in EWAS and claim whether a CpG site is associated with phenotypes at the aggregate level, none of them can identify the risk-CpG sites for each individual cell type, thus missing the opportunity to obtain finer-grained results in EWAS.

Here, we propose a method, HIRE, to identify the association in EWAS at a HIgh REsolution: detecting whether a CpG site has any associations with the phenotypes in each cell type (Methods). The keys to HIRE's success are twofold. First, HIRE links the underlying cell-type-specific methylation profiles for each sample to the sample's phenotypes, thus avoiding the bias in estimating the cellular composition by the reference-based and direct-decomposition-based methods. Second, HIRE correctly characterizes the cellular compositions as the multiplicative effects, whereas the existing methods inappropriately treat the cell proportions as additive effects (Methods). HIRE is applicable to EWAS with binary phenotypes, continuous phenotypes, or both. By helping researchers understand in which cell types the CpG sites are affected by a disease, HIRE can ultimately facilitate the development of epigenetic therapies by targeting the specifically affected cell types.

## Results

**Method overview.** HIRE is a hierarchical model that closely follows the data generation process. Its elaborate modeling depicts how phenotypes affect the methylation levels of each sample. Here, we briefly introduce the method. The technical details are provided in the Methods section and the Supplementary Methods.

Let us first review the cornerstone in most EWAS approaches. These methods model the observed methylation levels of the $m$ CpG sites for sample $i$, $\mathbf{O}_i = (O_{1i}, O_{2i}, \ldots, O_{mi})^T$, as the weighted average of the methylation profiles of $K$ cell types, $\mathbf{u}_i = (\mathbf{u}_{i1}, \mathbf{u}_{i2}, \ldots, \mathbf{u}_{iK})$. The weights are the cellular compositions $\mathbf{p}_i = (p_{1i}, p_{2i}, \ldots, p_{Ki})^T$ of sample $i$ (see the top panel of Fig. 1a). However, regardless of whether the reference is known or not, the existing methods assume that the cell-type-specific methylation profiles $\mathbf{u}_i$s remain the same for all samples: $\mathbf{u}_i = \mathbf{M}$, for $i = 1, \ldots, n$. Unfortunately, because the methylation levels can actually change with covariates such as age and disease status, ignoring the covariates' effects and enforcing static reference methylomes can bias the estimation of $\mathbf{p}_i$ and thus affect all downstream analyses[14]. More importantly, the assumption that cell-type-specific methylation profiles are the same for each sample prevents the detection of cell-type-specific risk-CpG sites.

For association detection at the aggregate level, after estimation of $\mathbf{p}_i$ using the deconvolution-based approach or its surrogates from principal component-based methods, the existing methods examine a linear model in which the phenotypes $\mathbf{x}_i = (x_{i1}, \ldots, x_{i\ell}, \ldots, x_{iq})^T$ and the cellular proportions $\mathbf{p}_i$ exert additive effects on the methylation level $\mathbf{O}_i$:

$$\mathbf{O}_i = \mathbf{T}\mathbf{x}_i + \mathbf{M}\mathbf{p}_i. \tag{1}$$

A CpG-site $j$ is then associated with phenotype $\ell$ if we reject the null hypothesis that the covariate coefficient $T_{j\ell}$ equals zero.

In contrast, HIRE further models the effect of each phenotype on each cell type as shown in the bottom panel of Fig. 1a. In cell type $k$, sample $i$'s cell-type-specific methylation profile, $\mathbf{u}_{ik}$, is the summation of the corresponding baseline cell-type-specific methylation levels, $\boldsymbol{\mu}_k$, and the phenotype effects $\mathbf{B}_{k\ell}x_{i\ell}$ on sample $i$ from all the $l = 1, \ldots, q$ phenotypes: $\mathbf{u}_{ik} = \boldsymbol{\mu}_k + \sum_{l=1}^{q} \mathbf{B}_{k\ell}x_{i\ell}$, where $x_{i\ell}$ is the phenotype $\ell$ of sample $i$ and $\mathbf{B}_{k\ell} = (\beta_{1k\ell}, \ldots, \beta_{mk\ell})^T$—the $k$th column of $\mathbf{B}_\ell$—reflects the association of phenotype $\ell$ with each of the $m$ CpG sites in cell type $k$. Thus, by collecting the baseline cell-type-specific

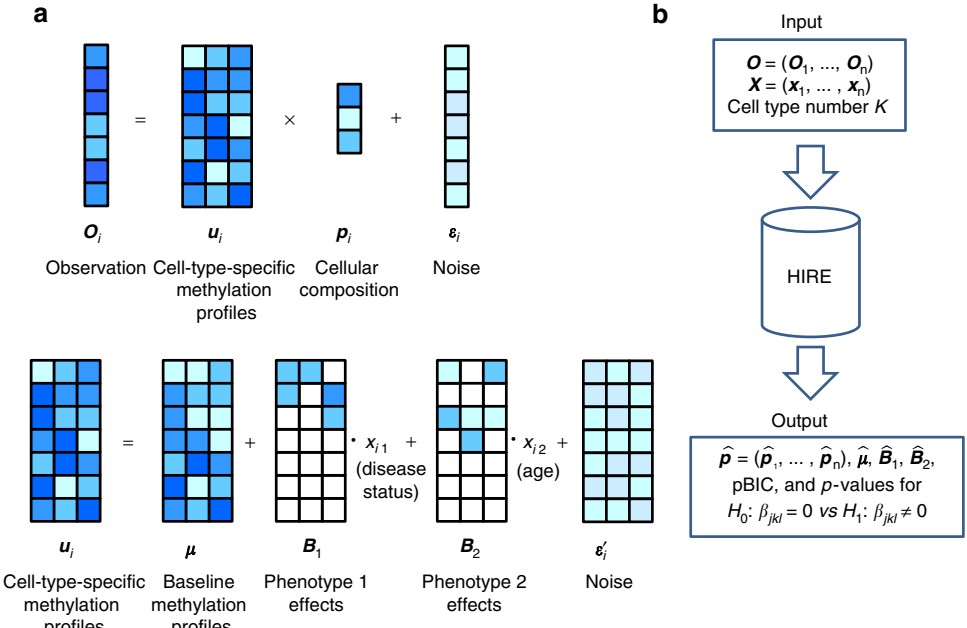

**Fig. 1** A simple cartoon illustration of the HIRE model with three cell types ($K = 3$) and two phenotypes (disease status and age; $q = 2$). **a** Data generation procedure for the observed methylation vector $\mathbf{O}_i$ for sample $i$ ($i = 1, ..., n$). In the top panel, $\mathbf{O}_i$ is the convolution of cell-type-specific methylation profiles $\mathbf{u}_i$ with cellular compositions $\mathbf{p}_i$. Both $\mathbf{u}_i$ and $\mathbf{p}_i$ depend on the attributes of sample $i$. The bottom panel describes how sample $i$'s phenotypes affect $\mathbf{u}_i$ via two phenotype-effect matrices $\mathbf{B}_1$ and $\mathbf{B}_2$. In $\mathbf{B}_1$ and $\mathbf{B}_2$, the white square represents zero, which indicates that the phenotype exerts no influence on the corresponding methylation level in $\mathbf{u}_i$. **b** Inputs and outputs of HIRE. We input the observed methylation matrix $\mathbf{O}$, the phenotype data matrix $\mathbf{X}$, and a predetermined cell type number $K$ into HIRE, and HIRE outputs the estimates for the cellular compositions $\hat{\mathbf{p}}$, the baseline methylation profiles $\hat{\boldsymbol{\mu}}$, the phenotype effects $\hat{\mathbf{B}}_\ell$, and the penalized BIC value. In addition, HIRE tests whether there is any association between CpG site $j$ and phenotype $\ell$ in cell type $k$ —$H_0 : \beta_{jk\ell} = 0$ vs $H_1 : \beta_{jk\ell} \neq 0$—and provides the p-values

methylation profiles to $\boldsymbol{\mu} = (\boldsymbol{\mu}_1, ..., \boldsymbol{\mu}_k)$ and denoting the $m$ by $K$ phenotype coefficient matrix ($\beta_{jk\ell} : 1 \leq j \leq m, 1 \leq k \leq K$) by $\mathbf{B}_\ell$, we now have:

$$\mathbf{O}_i = \mathbf{u}_i \mathbf{p}_i = \sum_{l=1}^{q} \mathbf{B}_\ell x_{i\ell} \mathbf{p}_i + \boldsymbol{\mu} \mathbf{p}_i = (\mathbf{B}_1 \mathbf{p}_i, ..., \mathbf{B}_q \mathbf{p}_i) \mathbf{x}_i + \boldsymbol{\mu} \mathbf{p}_i. \quad (2)$$

A comparison of $\mathbf{x}_i$ in Eq. (1) and $x_{i\ell} \boldsymbol{p}_i, \ell = 1, ..., q$ in Eq. (2) reveals that via the two-layer hierarchical model HIRE correctly captures the multiplicative effects of the cellular compositions on the phenotype effects (see also Methods and the Supplementary Methods). As a result, HIRE achieves greater statistical power for association detection at the aggregate level and enables the fine-scale resolutions that were previously infeasible. We mathematically prove that the HIRE model is identifiable under mild conditions that are easily met in reality (see Theorem 1 and its proof in Methods).

Figure 1b summarizes the inputs and outputs of HIRE. Given the methylation measurements at the aggregate level of $n$ samples, HIRE can estimate all parameters of interests —$\mathbf{p}_i$ ($i = 1, ..., n$), $\boldsymbol{\mu}$, and $\mathbf{B}_\ell$ ($\ell = 1, ..., q$). HIRE then determines whether any association exists between CpG site $j$ and phenotype $\ell$ in each individual cell type by testing the hypotheses $H_0 : \beta_{jk\ell} = 0$ versus $H_1 : \beta_{jk\ell} \neq 0$. When the null hypothesis $H_0 : \beta_{jk\ell} = 0$ is rejected, HIRE calls CpG site $j$ as a risk-CpG site for phenotype $\ell$ in cell type $k$. The detection of cell-type-specific risk-CpG sites cannot be performed with any of the existing state-of-the-art methods.

Moreover, HIRE allows users to prespecify the number of cell types $K$. When $K$ is unknown, HIRE selects the number of cell types according to the penalized Bayesian information criterion (pBIC)[20] (Supplementary Methods).

**Simulation**. As the definition of the gold standard for real data is debatable[21,22], we designed extensive simulation studies to evaluate the performance of HIRE and compared it with commonly used methods—unadjusted analysis, SVA, RefFreeEWAS, EWASHer, and ReFACTor (Methods). We generated datasets in which the observed methylation was a mixture of several cell types and each sample was accompanied with a diseased or normal status and a continuous age attribute. We deliberately designed some cell types to have similar baseline methylation profiles to mimic cell types from the same cell lineage. We set the sample size $n$ to 180, 300, and 600 and let the underlying cell type number $K$ be 3, 5, and 7. For each pair of ($n$, $K$), we investigated two scenarios in which (1) all phenotype effects $\beta_{jk\ell}$s are zero— the true null case—to compare the ability of each method to control false positives; and (2) a small portion of $\beta_{jk\ell}$s are non-zero—the true alternative case—to study each method's power to detect risk-CpG sites. Under the true alternative, both the binary and the continuous phenotypes were assumed to have cell-type-specific risk-CpG sites and to affect the cell-type proportions among the samples[10]. We further simulated phenotype effects with various directions and magnitudes.

Under the true null, HIRE, EWASHer, and ReFACTor control the false positive rates (FPRs) very well: none are greater than 0.05% (Table 1 and Supplementary Figs. 1–9). In comparison, RefFreeEWAS often has FPRs greater than 0.1% and thus does not perform as well as HIRE, and the unadjusted analysis and SVA further suffer from the dramatic inflation of false positives. For the true alternative settings, given that the FPRs are well-controlled, with FPRs below 0.05%, HIRE achieves the highest true positive rates (TPR) of all methods in every simulation setting (see also Fig. 2a and Supplementary Figs. 10–17). As expected, as the sample size increases, HIRE's power increases.

**Table 1 Performance of HIRE and other competing methods in simulation studies**

| | Cell type number | Sample size | | HIRE | SVA | Unadjusted | RefFreeEWAS | EWASHer | ReFACTor |
|---|---|---|---|---|---|---|---|---|---|
| True null | $K = 3$ | $n = 180$ | FPR | 0 | 0.35% | 21.72% | 0.12% | 0 | 0 |
| True null | $K = 3$ | $n = 300$ | FPR | 0.00% | 3.13% | 53.40% | 0.12% | 0.00% | 0 |
| True null | $K = 3$ | $n = 600$ | FPR | 0 | 22.06% | 77.2% | 0.1% | 0.00% | 0 |
| True null | $K = 5$ | $n = 180$ | FPR | 0.05% | 0.3% | 18.5% | 0.13% | 0 | 0 |
| True null | $K = 5$ | $n = 300$ | FPR | 0.01% | 1.54% | 36.89% | 0.08% | 0 | 0 |
| True null | $K = 5$ | $n = 600$ | FPR | 0 | 8.09% | 55.66% | 0.12% | 0 | 0 |
| True null | $K = 7$ | $n = 180$ | FPR | 0.00% | 0.04% | 5.13% | 0.11% | 0 | 0 |
| True null | $K = 7$ | $n = 300$ | FPR | 0.00% | 0.22% | 22.84% | 0.12% | 0 | 0 |
| True null | $K = 7$ | $n = 600$ | FPR | 0 | 6.80% | 48.12% | 0.11% | 0 | 0 |
| True alternative | $K = 3$ | $n = 180$ | FPR | 0.01% | 1.24% | 32.20% | 0.11% | 0 | 0 |
| True alternative | $K = 3$ | $n = 180$ | TPR | 98.67% | 87.33% | 60.67% | 79.33% | 21.33% | 60.67% |
| True alternative | $K = 3$ | $n = 300$ | FPR | 0.00% | 7.60% | 65.03% | 0.09% | 0 | 2.21% |
| True alternative | $K = 3$ | $n = 300$ | TPR | 96.67% | 86% | 79.33% | 63.33% | 24.67% | 56.67% |
| True alternative | $K = 3$ | $n = 600$ | FPR | 0.03% | 16.91% | 73.55% | 0.11% | 0.00% | 1.34% |
| True alternative | $K = 3$ | $n = 600$ | TPR | 100% | 84% | 88.67% | 90% | 35.33% | 45.33% |
| True alternative | $K = 5$ | $n = 180$ | FPR | 0.00% | 2.70% | 20.06% | 0.14% | 0 | 0 |
| True alternative | $K = 5$ | $n = 180$ | TPR | 66% | 80.8% | 35.2% | 71.2% | 6.4% | 41.6% |
| True alternative | $K = 5$ | $n = 300$ | FPR | 0.01% | 1.76% | 36.11% | 0.17% | 0 | 0.01% |
| True alternative | $K = 5$ | $n = 300$ | TPR | 89.6% | 86.4% | 69.2% | 76.4% | 11.2% | 53.6% |
| True alternative | $K = 5$ | $n = 600$ | FPR | 0.01% | 15.31% | 56.25% | 0.09% | 0 | 0.11% |
| True alternative | $K = 5$ | $n = 600$ | TPR | 98.4% | 82.8% | 62% | 84.4% | 18% | 62.8% |
| True alternative | $K = 7$ | $n = 180$ | FPR | 0 | 0.37% | 9.30% | 0.11% | 0 | 0 |
| True alternative | $K = 7$ | $n = 180$ | TPR | 43% | 58.67% | 35% | 54.67% | 5% | 26% |
| True alternative | $K = 7$ | $n = 300$ | FPR | 0.00% | 1.08% | 26.67% | 0.11% | 0 | 0 |
| True alternative | $K = 7$ | $n = 300$ | TPR | 63.33% | 73% | 45% | 76.33% | 5% | 35.67% |
| True alternative | $K = 7$ | $n = 600$ | FPR | 0.04% | 25.80% | 56.76% | 0.10% | 0 | 0.00% |
| True alternative | $K = 7$ | $n = 600$ | TPR | 82.67% | 83% | 66% | 74.33% | 5% | 51.67% |

Performance of HIRE and other competing methods in simulation studies in detecting risk-CpG sites at the aggregate level. For the true null cases in which no CpG site is at risk, the average of the false positive rates (FPRs) based on five replicates is reported. For the true alternative cases, the averages of the FPRs and the true positive rates (TPRs) based on five replicates are reported. The number of CpG sites at risk is 30, 50, and 60 for the cell type number $K = 3$, 5, and 7, respectively. HIRE calls a CpG site as significant at the aggregate level if it is at risk in at least one cell type. We used Bonferroni correction for each method to control the family-wise error rate (FWER) below $\alpha = 0.01$. Because HIRE can provide the p-values of CpG sites for all cell types and phenotypes, the p-value threshold for significance is $\alpha/(mKq)$, where $m = 10{,}000$ is the number of CpG sites and $q = 2$ is the phenotype number. For the other five methods, the p-value threshold is set to $\alpha/m$. Notice that "0" represents exact zero, and "0.00%" indicates a very small positive number that is rounded down to zero using four decimal places

For example, when the data include five cell types, HIRE can identify 89.6% of the risk-CpG sites with 300 samples, and HIRE can detect almost all risk-CpG sites when the sample size reaches 600, which is a typical sample size for EWAS. Although EWASHer and ReFACTor have low FPRs, they miss a large proportion of risk-CpG sites. EWASHer's maximum TPR is only 35.33%, and ReFACTor's maximum TPR is slightly over 60%. However, in those cases, HIRE's power is greater than 95%. Consistent with the true null scenario, in the true alternative, RefFreeEWAS has inflated FPRs compared to HIRE, and the unadjusted analysis and SVA always have huge false positives. Therefore, HIRE substantially improves the power of association detection at the aggregate level compared with existing methods.

In the multiple hypothesis testing, the p-values from the truly null features should follow a uniform distribution on (0, 1), whereas those for the truly alternative features are concentrated near zero[23]. Both the histograms (Fig. 2d–i) and the Q-Q plots (Fig. 2j–o) show that the p-value distribution of HIRE is the best fit to the underlying truth—there are only a small proportion of signals, followed by RefFreeEWAS and ReFACTor. EWASHer easily overcorrects signals with its p-value density having a dip near zero (Fig. 2h), thus failing to detect the true associations. In contrast, the unadjusted analysis and SVA generate very small p-values clustered near zero, resulting in inflated type I errors.

In addition to the traditional association detection at the aggregate level, HIRE can identify the association for each CpG site with the phenotypes under each cell type. Table 2 shows the FPR and TPR of HIRE for each cell type in various simulation settings. Such fine analysis is not possible with the other methods. Consistent with association detection at the aggregate level, HIRE always controls the FPR well. When $K = 3$ and $n = 180$, HIRE accurately detects the risk-CpG sites associated with disease status a TPR of greater than 83% and an FPR of 0.01% or less in all three

cell types. Similarly, most of the CpG sites affected by age are also correctly identified in each cell type. HIRE's learned cell-type-specific association patterns closely matches the underlying true associations (see Fig. 2b, c and Supplementary Figs. 18–26). Once again, HIRE's power decreases with the number of cell types and increases with the sample size. When the samples consist of seven cell types and the proportion of the least abundant cell type is as low as 4.2%, given a typical current EWAS with around 600 samples, HIRE can detect most cell-type-specific risk-CpG sites reasonably well. Moreover, HIRE's estimates for the baseline methylation profiles, cellular compositions, and phenotype effects have little bias (Supplementary Figs. 27–62); therefore, HIRE can provide accurate estimates and is powerful in detecting cell-type-specific risk-CpG sites.

In the HIRE model, we assume that different CpG sites are independent, and we investigate the performance of HIRE when such a model assumption is violated and dependences exist among nearby CpG sites. Specifically, we assume that every 50 consecutive CpG sites belongs to a block. For CpG sites within the same block, their random noises $\epsilon$ follow a multivariate normal distribution with mean zero and $50 \times 50$ covariance matrix $\Sigma$, and $\Sigma$'s corresponding correlation matrix has its $(i, j)$ entry equal to $\rho^{|i-j|}$. We vary $\rho$ to 0.8, 0.6, and 0.4. A comparison of Supplementary Tables 1–3 with Supplementary Table 4 shows that even when strong correlations exit among nearby CpG sites, HIRE still provides good performances in controlling the FPR and detecting the risk-CpG sites under the model misspecification setting.

To further evaluate HIRE's performance on experimentally mixed samples, we conducted another semi-simulated dataset that includes six samples mixed with six purified cell types in predetermined proportions[24]. Once again, HIRE successfully recovers the six underlying reference cell types and estimates the cellular compositions well (see Methods).

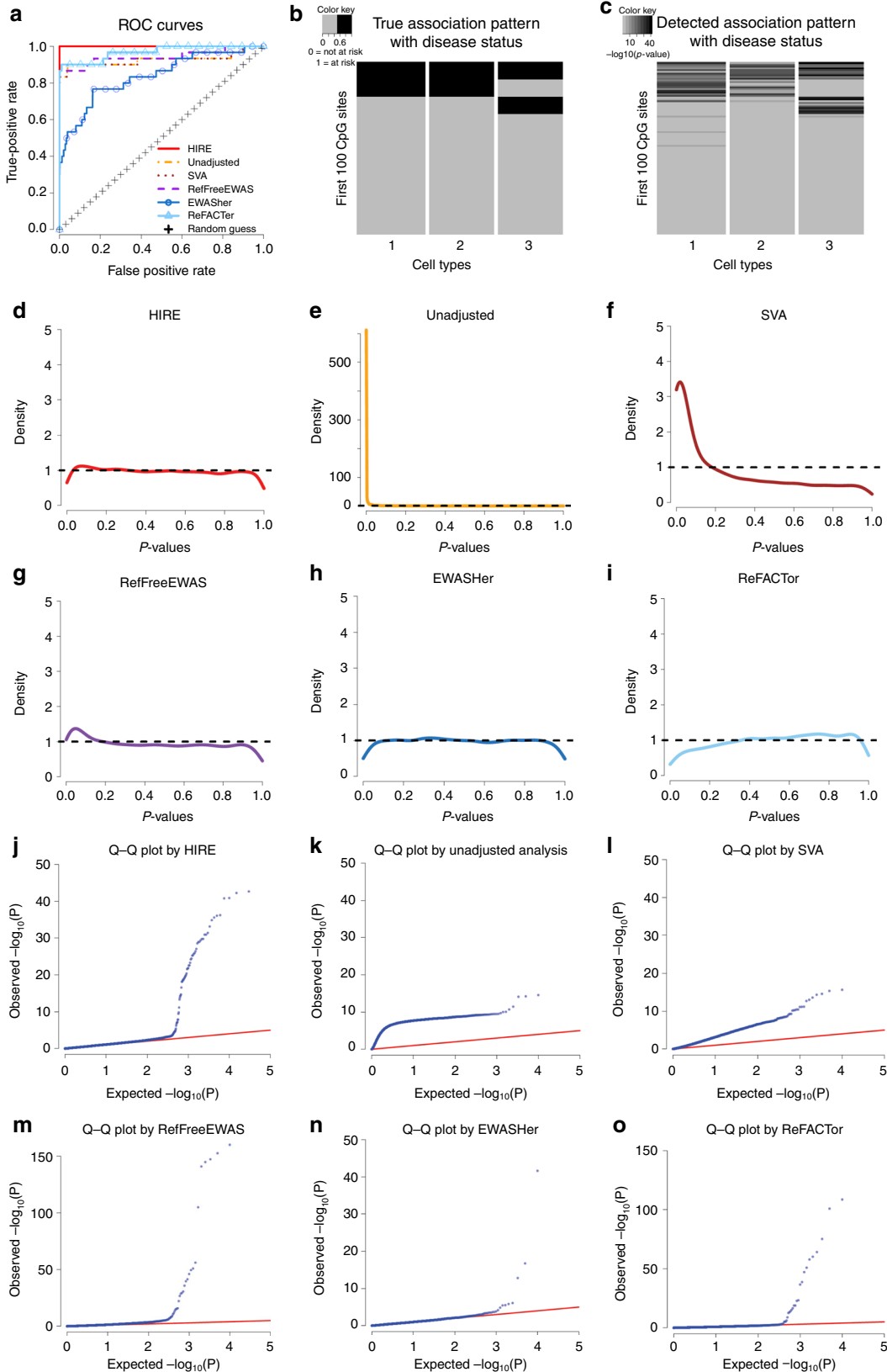

**Real data analysis**. HIRE also provides greater insight into real data than previous studies. The rheumatoid arthritis (RA) dataset[3] contains methylation profiles collected from the whole blood of 354 patients with RA and 335 normal participants. In addition to the RA status, other attributes such as gender, smoking history,

age, and batch information are available. We first corrected the batch effects and then applied HIRE to the dataset (Methods). Figure 3a displays the p-values regarding the association with the RA status for each CpG site in each cell type, in which HIRE selected six cell types (Supplementary Fig. 63a), consistent with

**Fig. 2** Association detection performance of HIRE and commonly used methods in the true alternative setting with $K = 3$ and $n = 180$. Source data are provided as a Source Data file. In all figures, red corresponds to HIRE; yellow indicates the unadjusted analysis; brown represents SVA; purple refers to RefFreeEWAS; dark blue indicates EWASher; and light blue corresponds to ReFACTor. **a** ROC curves of HIRE and commonly used methods. HIRE has the largest area under the curve among all of the methods. **b** True cell-type-specific association pattern with disease status for 10,000 simulated CpG sites; columns correspond to cell types, and the rows represent the CpG sites. Dark cells correspond to risk-CpG sites, and grey cells are CpG sites not associated with the disease status. **c** Detected cell-type-specific association pattern with disease status by HIRE. Darkness represents $-\log_{10}(p - \text{value})$ **d–i** The *p*-value density plots for association with disease status in the simulation dataset for **d** HIRE, **e** unadjusted analysis, **f** SVA, **g** RefFreeEWAS, **h** EWASher, and **i** ReFACTor. **j–o** The Q-Q plots for association with disease status for **j** HIRE, **k** unadjusted analysis, **l** SVA, **m** RefFreeEWAS, **n** EWASher, and **o** ReFACTor

### Table 2 Performance of HIRE in detecting cell-type-specific risk-CpG sites

| Phenotype | Cell type number | Sample size | | Cell type 1 | Cell type 2 | Cell type 3 | Cell type 4 | Cell type 5 | Cell type 6 | Cell type 7 |
|---|---|---|---|---|---|---|---|---|---|---|
| Disease status | $K = 3$ | $n = 180$ | FPR | 0.01% | 0.00% | 0.01% | | | | |
| Disease status | $K = 3$ | $n = 180$ | TPR | 83% | 85% | 92% | | | | |
| Disease status | $K = 3$ | $n = 300$ | FPR | 0.02% | 0.02% | 0.04% | | | | |
| Disease status | $K = 3$ | $n = 300$ | TPR | 74% | 85% | 95% | | | | |
| Disease status | $K = 3$ | $n = 600$ | FPR | 0.03% | 0.03% | 0.05% | | | | |
| Disease status | $K = 3$ | $n = 600$ | TPR | 99% | 98% | 100% | | | | |
| Disease status | $K = 5$ | $n = 180$ | FPR | 0.01% | 0 | 0.00% | 0.01% | 0.02% | | |
| Disease status | $K = 5$ | $n = 180$ | TPR | 35% | 46% | 44% | 39% | 75% | | |
| Disease status | $K = 5$ | $n = 300$ | FPR | 0.02% | 0.02% | 0.02% | 0.06% | 0.10% | | |
| Disease status | $K = 5$ | $n = 300$ | TPR | 66% | 73% | 67% | 43% | 43% | | |
| Disease status | $K = 5$ | $n = 600$ | FPR | 0.02% | 0.02% | 0.01% | 0.10% | 0.12% | | |
| Disease status | $K = 5$ | $n = 600$ | TPR | 81% | 77% | 92% | 52% | 56% | | |
| Disease status | $K = 7$ | $n = 180$ | FPR | 0 | 0 | 0.01% | 0 | 0.00% | 0 | 0.00% |
| Disease status | $K = 7$ | $n = 180$ | TPR | 13% | 28% | 32% | 20% | 21% | 15% | 69% |
| Disease status | $K = 7$ | $n = 300$ | FPR | 0.01% | 0.01% | 0.01% | 0.00% | 0.01% | 0.01% | 0.02% |
| Disease status | $K = 7$ | $n = 300$ | TPR | 20% | 48% | 60% | 52% | 40% | 23% | 78% |
| Disease status | $K = 7$ | $n = 600$ | FPR | 0.02% | 0.02% | 0.01% | 0.02% | 0.01% | 0.01% | 0.07% |
| Disease status | $K = 7$ | $n = 600$ | TPR | 37% | 79% | 90% | 52% | 71% | 66% | 98% |
| Age | $K = 3$ | $n = 180$ | FPR | 0.01% | 0.01% | 0.06% | | | | |
| Age | $K = 3$ | $n = 180$ | TPR | 68% | 76% | 96% | | | | |
| Age | $K = 3$ | $n = 300$ | FPR | 0.05% | 0.03% | 0.08% | | | | |
| Age | $K = 3$ | $n = 300$ | TPR | 95% | 95% | 90% | | | | |
| Age | $K = 3$ | $n = 600$ | FPR | 0.06% | 0.06% | 0.08% | | | | |
| Age | $K = 3$ | $n = 600$ | TPR | 94% | 99% | 95% | | | | |
| Age | $K = 5$ | $n = 180$ | FPR | 0.05% | 0.05% | 0.01% | 0.04% | 0.06% | | |
| Age | $K = 5$ | $n = 180$ | TPR | 67% | 61% | 82% | 69% | 97% | | |
| Age | $K = 5$ | $n = 300$ | FPR | 0.09% | 0.03% | 0.04% | 0.04% | 0.08% | | |
| Age | $K = 5$ | $n = 300$ | TPR | 78% | 85% | 97% | 85% | 91% | | |
| Age | $K = 5$ | $n = 600$ | FPR | 0.07% | 0.06% | 0.07% | 0.08% | 0.08% | | |
| Age | $K = 5$ | $n = 600$ | TPR | 88% | 84% | 94% | 83% | 94% | | |
| Age | $K = 7$ | $n = 180$ | FPR | 0.02% | 0.01% | 0.01% | 0 | 0.02% | 0.01% | 0 |
| Age | $K = 7$ | $n = 180$ | TPR | 39% | 62% | 58% | 68% | 38% | 54% | 85% |
| Age | $K = 7$ | $n = 300$ | FPR | 0.08% | 0.01% | 0.04% | 0.01% | 0.03% | 0.03% | 0.02% |
| Age | $K = 7$ | $n = 300$ | TPR | 46% | 62% | 79% | 84% | 79% | 73% | 93% |
| Age | $K = 7$ | $n = 600$ | FPR | 0.09% | 0.08% | 0.04% | 0.10% | 0.03% | 0.05% | 0.07% |
| Age | $K = 7$ | $n = 600$ | TPR | 52% | 77% | 85% | 84% | 77% | 79% | 84% |

Performance of HIRE in detecting cell-type-specific risk-CpG sites in the true alternative cases. The results are based on five replicates for each setting. A CpG site is claimed to be significant in a given cell type if its p-value is less than $\alpha/(mKq)$

the number of cell types in the previous study[13]. Despite potential batch effects and biological variability, three of the six cell types can be matched to known blood cell references—cell type 1 was matched to CD4+ T cells, cell types 2 and 4 were matched to neutrophils, and the remaining three cell types cannot be aligned to the references (Methods and Supplementary Fig. 64). HIRE detected 63 risk-CpG sites in cell type 3—the largest number of associations across all cell types—but no risk-CpG sites in cell type 1 (Supplementary Table 5). Therefore, the disease status affected some but not necessarily all cell types. Note that the significant CpG site cg06373940 called by HIRE is located on gene *ERCC3*. The level of *ERCC3*'s corresponding protein has been reported to increase in RA synovium[25]. Moreover, we found that five CpG sites had a significant association with smoking history (Supplementary Fig. 65 and Supplementary Table 6). One of them is cg05575921, which was recently linked to smoking in two other independent studies of blood samples[26,27]. However, these findings were missed by the association detection at the aggregate level in previous analyses of the same dataset[11,13]. The p-value density plots and Q-Q plots for the commonly used methods are also displayed in Fig. 3c–n; they present patterns

similar to those observed in the simulation study except for an obvious overcorrection by ReFACTor.

The high resolution provided by HIRE makes it a powerful tool for EWAS studies. Rahmani et al. used ReFACTor[13] to analyze the GALA II blood methylation dataset[28], which consists of 573 samples collected from a pediatric Latino population. Each sample includes the gender information and belongs to one of the following four populations: Mexican, Mixed Latino, Puerto Rican, and Other Latino. We applied HIRE to the dataset to investigate whether any cell-type-specific CpG sites were associated with gender and ethnicity. We created three dummy variables to represent the four ethnic groups. By taking the indicators of ethnicity as phenotypes in the model, HIRE automatically and simultaneously accounts for the population differences in cell composition and cell-type-specific methylation levels. HIRE correctly selected the number of cell types as six as reported in the previous study[13] (Supplementary Fig. 63b). According to cell-type alignment, cell types 1 and 5 can be annotated as CD4+ T cells; cell types 2, 3, and 4 belong to neutrophils; and cell type 6 was annotated as CD56+ natural killer cell (CD56+ NK) using the references (Supplementary Fig. 66). HIRE found that 1936

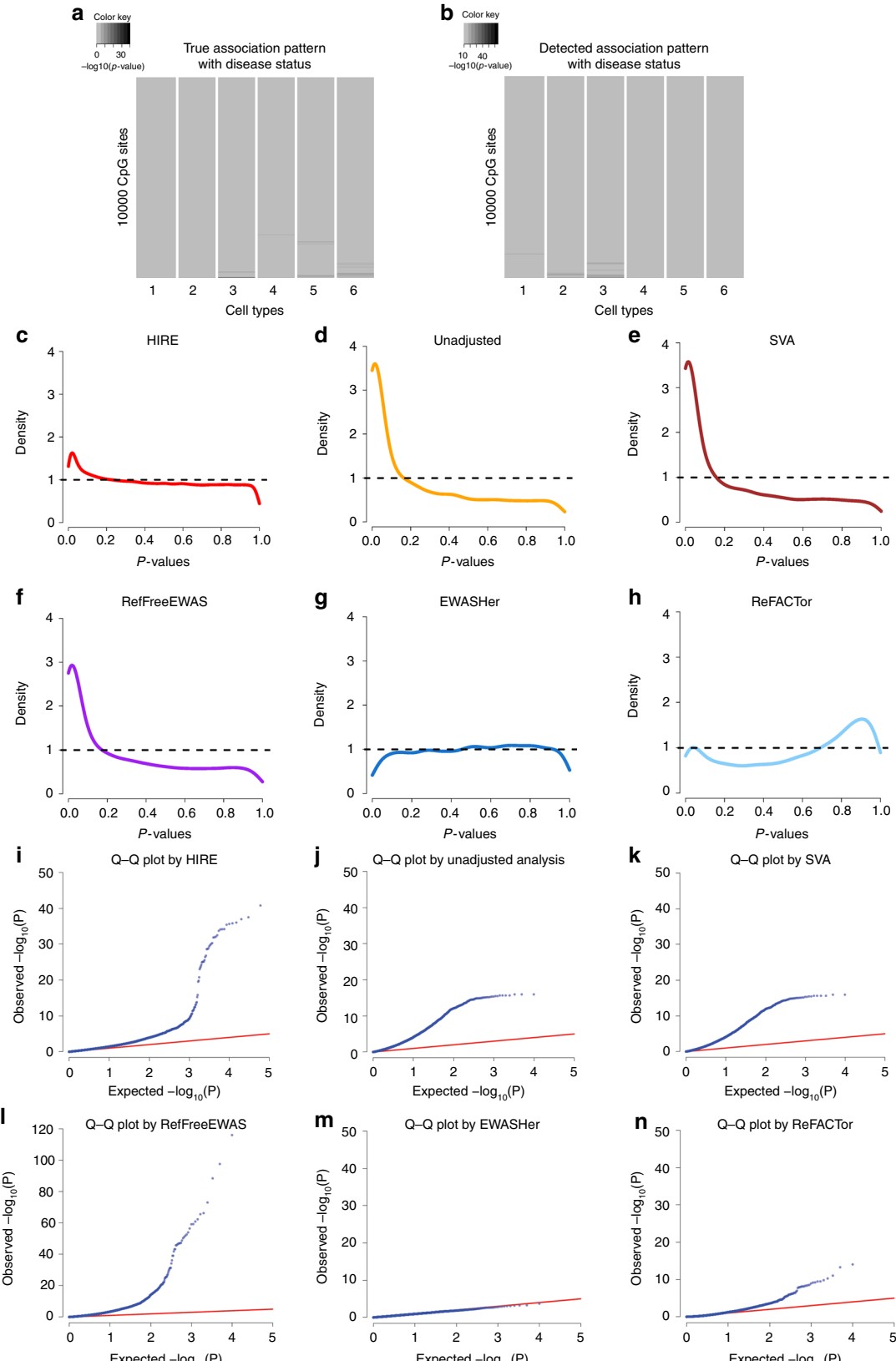

**Fig. 3** Application of HIRE and commonly used methods to two real methylation datasets: RA and GALA II. Source data are provided as a Source Data file. **a** Cell-type-specic association pattern with RA status detected by HIRE in the RA dataset. Darkness represents the −log10(p−value). **b** Cell-type-specic association pattern with gender detected by HIRE in the GALA II dataset. The darkness represents the −log10(p−value). **c−h** The p-value density plots for association with RA status in the RA dataset for **c** HIRE, **d** unadjusted analysis, **e** SVA, **f** RefFreeEWAS, **g** EWASHer, and **h** ReFACTor. **i-n** Q-Q plots for association with RA status in the RA dataset for **i** HIRE, **j** unadjusted analysis, **k** SVA, **l** RefFreeEWAS, **m** EWASHer, and **n** ReFACTor

CpG sites were associated with ethnicity across all cell types (Supplementary Fig. 67) and identified 14, 52, 155, 15, 18, and 14 risk-CpG sites for gender in cell types 1–6, respectively (Fig. 3b). Gene set enrichment analysis showed that the genes that harbored risk-CpG sites for gender were significantly enriched in seven canonical pathways (Supplementary Table 7), of which the PID_CMYB_PATHWAY was ranked the highest. The transcription factor $c − MYB$ in the PID_CMYB_PATHWAY enhances the progression of breast cancer[29]; therefore, the different occurrence rates of breast cancers in men and women may be linked to the differences at the epigenome level. In comparison, only one pathway was found to be enriched with the genes that host the risk-CpG sites claimed by ReFACTor at the aggregate level (Supplementary Table 8). All of these observations highlight the importance of the finer-scale resolutions of HIRE.

## Discussion

In reality, the phenotype may affect a risk-CpG site in some but not all of the cell types. HIRE can detect the cell-type-specific association pattern with each phenotype for EWAS. The identification of cell-type-specific risk-CpG sites will help epigenetic therapies to target the affected cell types in a more effective manner.

Statistically, instead of assuming fixed reference methylomes for all samples as the existing methods do[9,13,16], HIRE allows each sample's cell-type-specific methylation profiles to depend on its phenotypes. As a result, HIRE correctly models the multiplicative effects of the cellular compositions on the observed methylation levels, whereas the existing approaches all misspecify the cellular compositions as additive effects (Methods). As a result, HIRE enables the detection of cell-type-specific risk-CpG sites that cannot be feasibly detected with existing state-of-the-art methods. As a byproduct, HIRE also improves the statistical power of association detection at the aggregate level relative to existing state-of-the-art methods. Computationally, the time complexity of one iteration by HIRE is $O(nmKp + nK^3)$, which thus provides fast convergence when $K$ is moderate. The statistical and computational advantages equip HIRE to be scaled up for large-cohort EWAS.

So far, in the EWAS community, no gold-standard exists for the comparison of various methods. Ideally, we would like to have epigenetic spike-in experiments in which purified cell types are isolated, CpG-sites are epigenetically edited on a per-cell-type basis, and cell types are finally mixed in predetermined proportions. Given such experiments, the underlying knowledge of which CpGs are differentially methylated in each cell type and the cell mixing proportions for each sample are known. However, biotechnologies for epigenetic editing, such as CRISPR-Cas, are still not mature at this stage, with many off-target modifications[30]. Therefore, most computational EWAS studies refer to numerical simulation studies rather than to experimental studies when evaluating the performance of their algorithms[12,13]. Here, we follow the example of previous comparative studies and design our simulation studies to serve as the computational counterpart of experimental spike-in studies. With the rapid advances in epigenetic editing, we hope the community can devote greater effort in the near future to the creation of a gold-standard dataset, such as those generated in the early years for gene expression microarray studies[31].

The beta-values that represent methylation levels always lie between zero and one. As previous approaches to EWAS often assume normal distribution for the beta-values and show good performances in real applications[9,13], in HIRE, we also assume that the beta-values follow a normal distribution. Consequently, the fitted methylation level may lie outside the range of [0, 1]. Nevertheless, we do in fact constrain the baseline methylation profiles $\mu_{jk}$s to the closed interval [0, 1] and force the cellular compositions $p_{ki}$s to be non-negative and to add up to one: $\sum_{k=1}^{K} p_{ki} = 1$. As a result, because the phenotypes have no effect on most CpG sites, most observations, $O_{ji}$s, have their means $\sum_{k=1}^{K} \mu_{jk} p_{ki}$s in [0, 1]. In fact, for both the RA dataset and the GALA II dataset, more than 99.99% of the fitted methylation values $\hat{O}_{ji}$s based on HIRE estimates lie between zero and one. Therefore, the normal assumption fits the data reasonably well and does not have a large effect on the performance of HIRE.

One major issue for all of the cell-type deconvolution methods is that deconvolution cannot be achieved if the cellular compositions do not vary among samples. For example, assuming that the samples are mixtures of two cell types and $\mathbf{p}_i = \mathbf{p}$ for all of the samples, then the observed methylation profile $\mathbf{O}_i$ equals $\mathbf{u}_{i1} p_1 + \mathbf{u}_{i2} p_2 = (\mathbf{u}_{i1} + p_2 C) p_1 + (\mathbf{u}_{i2} − p_1 C) p_2 := \widetilde{\mathbf{u}}_{i1} p_1 + \widetilde{\mathbf{u}}_{i2} p_2$ for any constant $C$. As a result, $\mathbf{u}_{i1}$ and $\mathbf{u}_{i2}$ are not estimable. In our paper, we show mathematically that HIRE is identifiable under mild conditions in Theorem 1 and that condition (b) of Theorem 1 formulates the requirement for the variability of the cellular compositions (Methods). HIRE can accurately estimate cellular compositions of tissues with great cellular heterogeneity, such as blood. Although the mild conditions in Theorem 1 are easily met for real DNA methylation data, identification of both sufficient and necessary conditions for model identifiability is a theoretically interesting and challenging statistical problem that we will investigate in a future study.

HIRE requires a moderate sample size to obtain precise estimates because HIRE needs to learn $(1 + 2K + qK)m + (K − 1)n$ parameters with a total of $mn$ observed values. Our simulation studies show that HIRE performs very well at the aggregate level with 180 samples (Table 1). If the sample size drops below 150, say to 120, HIRE can still control the FPR well but begins to lose power (Supplementary Table 9). For small sample sizes, we have also developed a special case of HIRE by reparameterizing all $\sigma_{gk}^2$s as one single parameter $\sigma^2$, and we found that such a variance-stabilized approach can achieve even better inflation control (see Supplementary Figs. 71–76) and power comparable to HIRE (see Supplementary Table 10). Like the two datasets analyzed in the real application, a typical sample size for a current EWAS exceeds 500, thus guaranteeing a high TPR for HIRE. Given the decreasing cost of EWAS, we recommend that researchers collect at least 200 samples for their studies for association detection at the aggregate level and 600 samples for identification of cell-type-specific risk-CpG sites. A larger sample size can further boost the power.

With the popularity of EWAS, we believe that HIRE will be widely applied, and we hope that HIRE can motivate more researchers to mine out finer-scale results from EWAS.

## Methods

**Multiplicative effects of cellular composition on methylation**. In this section, we illustrate that the effects of the cell-type composition are actually multiplicative. Let us assume that the beta-values that represent the methylation levels are observed across $m$ CpG sites for $n$ samples. As the measured sample comprises cells of various types, the observed beta-value is a weighted average of the mean methylation levels of distinct cell types, and the weights correspond to the proportions of each cell type. Let $O_{ji}$ denote the measurement at CpG site $j$ for sample $i$. If we assume that there exist $K$ cell types in all samples and that the mean methylation level for CpG site $j$ in cell type $k$ is $\mu_{jk}$, then

$$O_{ji} = \sum_{k=1}^{K} \mu_{jk} p_{ki} + \epsilon_{ji},$$

where $p_{ki}$ is the proportion of cell type $k$ in sample $i$ with a natural constraint $\sum_{k=1}^{K} p_{ki} = 1$, and $\epsilon_{ji}$ is a random error.

Let us consider a case-control EWAS. Without loss of generality, we assume that CpG site $j$ is differentially methylated between cases and controls in cell type 1 with a mean shift $\delta_{j1}$ and that it is not differentially methylated in the remaining cell types. As a result, for case samples,

$$O_{ji} = (\mu_{j1} + \delta_{j1})p_{1i} + \sum_{k=2}^{K}\mu_{jk}p_{ki} + \epsilon_{ji} = \delta_{j1}p_{1i} + \sum_{k=1}^{K}\mu_{jk}p_{ki} + \epsilon_{ji}.$$

If we then use $Z_i$ to indicate the case-control status of sample $i$, the observed methylation level becomes

$$O_{ji} = \delta_{j1}p_{1i}Z_i + \sum_{k=1}^{K}\mu_{jk}p_{ki} + \epsilon_{ji}. \tag{3}$$

Therefore, the proportions of cell type 1—$p_{1i}$, $i = 1, \dots, n$—have multiplicative effects rather than additive effects on the mean difference between the case and control samples.

The existing methods, which either estimate the cell type proportions explicitly or approximate them implicitly with surrogate variables, add the estimated proportions and the case-control indicator $Z_i$ as the covariates to the regression as follows:

$$O_{ji} = \alpha_j + \tau_j Z_i + \sum_{k=1}^{K-1}b_{jk}\hat{p}_{ki} + \epsilon_{ji}, \tag{4}$$

where $b_{jk}$s are the regression coefficients. As a result, CpG site $j$ is called differentially methylated on the basis of hypothesis testing for $\tau_j = 0$. In general, $\tau_j$ in Eq. (4) is not equal to $\delta_{j1}$ in Eq. (3). Please see the Supplementary Notes for a numerical example. Moreover, testing for $\tau_j = 0$ loses the information regarding cell type in which CpG site $j$ may be at risk. To account for the multiplicative effects, we propose the HIRE model that conserves the individual cell-type level information, which is introduced in the next section.

**The HIRE model.** HIRE uses a hierarchical model to closely follow the data generation process for the EWAS data. To begin, we assume that the baseline methylation level for CpG site $j$ in cell type $k$ is $\mu_{jk}$. For sample $i$ with phenotypes $\mathbf{x}_i = (x_{i1}, \dots, x_{iq})$, the mean methylation value for CpG site $j$ in cell type $k$ is assumed to be $\mu_{jk} + \sum_{\ell=1}^{q}\beta_{jk\ell}x_{i\ell}$. In other words, the phenotypes have linear effects where $\beta_{jk\ell}$ characterizes the influence of phenotype $\ell$ on CpG site $j$ in cell type $k$. Let $u_{ijk}$ represent the signal from CpG site $j$ in cell type $k$ for sample $i$ with $\mathbf{x}_i$. We assume that $u_{ijk}$ follows a normal distribution with mean $\mu_{jk} + \sum_{\ell=1}^{q}\beta_{jk\ell}x_{i\ell}$ and standard deviation $\sigma_{jk}$,

$$u_{ijk} \sim N\left(\mu_{jk} + \sum_{\ell=1}^{q}\beta_{jk\ell}x_{i\ell}, \sigma_{jk}^2\right). \tag{5}$$

After $u_{ijk}$s are generated for all of the $K$ cell types, the observed methylation value $O_{ji}$ is sampled as follows:

$$O_{ji} \sim N\left(\sum_{k=1}^{K}u_{ijk}p_{ki}, \sigma_{\epsilon j}^2\right). \tag{6}$$

Collectively, $\mathbf{O} = \{O_{ji} : 1 \leq j \leq m, 1 \leq i \leq n\}$ denote the observed data; $\mathbf{u} = \{(u_{ij1}, \dots, u_{ijK})^T : 1 \leq i \leq n, 1 \leq j \leq m\}$ are the missing data; and $\boldsymbol{\mu}_j = (\mu_{j1}, \dots, \mu_{jK})^T$, $\mathbf{B}^{(j)} = (\beta_{jk\ell})_{K \times q}$, $\sigma_{\epsilon j}^2$, the diagonal matrix $\Sigma_j = diag(\sigma_{j1}^2, \dots, \sigma_{jK}^2)$ for $j = 1, \dots, m$, and $\mathbf{p}_i = (p_{1i}, \dots, p_{Ki})^T$ for $i = 1, \dots, n$ are the parameters. With $\Theta = \{\mathbf{p}_i, \boldsymbol{\mu}_j, \mathbf{B}^{(j)}, \Sigma_j, \sigma_{\epsilon j}^2 : 1 \leq j \leq m, 1 \leq i \leq n\}$, the complete data log-likelihood function, $l_c$, can be expressed as follows:

$$l_c(\Theta|\mathbf{O}, \mathbf{u}) = \sum_{i=1}^{n}\sum_{j=1}^{m}\left\{-\frac{1}{2}\log\sigma_{\epsilon j}^2 - \frac{(O_{ji} - \mathbf{u}_{ij}^T\mathbf{p}_i)^2}{2\sigma_{\epsilon j}^2} - \frac{1}{2}\sum_{k=1}^{K}\log\sigma_{jk}^2\right.$$
$$\left. -\frac{1}{2}(\mathbf{u}_{ij} - \boldsymbol{\mu}_j - \mathbf{B}^{(j)}\mathbf{x}_i)^T\Sigma_j^{-1}(\mathbf{u}_{ij} - \boldsymbol{\mu}_j - \mathbf{B}^{(j)}\mathbf{x}_i)\right\} + Constant.$$

Accordingly, we develop a generalized expectation-maximization algorithm[32] to estimate the parameters. In the expectation-maximization algorithm, a good initialization can lead to faster convergence than random starts. We adopt the cellular composition estimations from the methylation matrix decomposition algorithm[16] with slight modifications as the initializations. The initial values for the baseline methylation profiles $\mu_{jk}$ are accordingly estimated by simple linear regressions. As the number of risk-CpG sites is often small, all of the phenotype effects $\beta_{jk\ell}$ are set to zero at the beginning. For the standard deviations, the initial values are randomly sampled from inverse gamma distributions with small means. We choose the number of cell types $K$ by using a variant of the penalized Bayesian information criterion (pBIC)[20] (see details in Supplementary Methods).

For each phenotype $\ell$, we can conduct the hypothesis test $H_0 : \beta_{jk\ell} = 0$ versus $H_1 : \beta_{jk\ell} \neq 0$ for any cell type $k$ and any CpG site $j$. Combining Eqs. (5) and (6), we obtain the following equations:

$$E\left[O_{ji}\right] = \mu_{j1} + \sum_{k=2}^{K}(\mu_{jk} - \mu_{j1})p_{ki} + \sum_{k=1}^{K}\sum_{\ell=1}^{q}\beta_{jk\ell}x_{i\ell}p_{ki}, i = 1, \dots, n. \tag{7}$$

We can then take $(O_{j1}, \dots, O_{jn})$ as the response vector and concatenate $\mathbf{1}_n$, $(p_{k1}, \dots, p_{kn})$ $(k = 2, \dots, K)$ and $(x_{i\ell}p_{k1}, \dots, x_{n\ell}p_{kn})$ $(\ell = 1, \dots, q; k = 1, \dots, K)$ to a $n \times (p + 1) \cdot K$ design matrix in the linear regression. We plug in the estimated cellular compositions $\hat{p}_{ki\ell}$ and conduct the hypothesis test for $\beta_{jk\ell} = 0$ using the two-sided t-tests in the linear models. We claim that CpG site $j$ has an association with phenotype $\ell$ at the aggregate level if phenotype $\ell$ affects CpG site $j$ in at least one of the $K$ cell types. Note that in the regression we incorporate the estimated cellular compositions into the linear model as multiplicative effects rather than additive effects.

More technical details of the method and the algorithm are available in the Supplementary Methods.

**Data simulation.** We compared the performance of HIRE with five previous methods—unadjusted analysis, SVA, RefFreeEWAS, EWASHer, and ReFACTor— in 18 simulation settings. We set the sample size $n$ to 180, 300, and 600 and let the underlying cell type number $K$ be 3, 5, and 7. For each pair of $(n, K)$, we investigated the true null case and the true alternative case. As a result, we have in total 3 (the number of sample sizes) × 3 (the number of cell types) × 2 (the true null case and the true alternative case) = 18 simulation settings. For each setting, we considered 10,000 CpG sites and simultaneously accounted for the following factors.

Cell lineage. We first constructed the baseline methylation matrix $\boldsymbol{\mu} = (\mu_{jk})_{m \times K}$, in which each column corresponds to the baseline methylation levels of a cell type. To mimic the phenomenon in which cell types from the same lineage have similar methylation profiles, we assumed that $K_{sim}$ of the total $K$ cell types were similar. Specifically, without loss of generality, we assumed that the first $K_{sim}$ cell types came from the same cell lineage and that the remaining $K - K_{sim}$ cell types are irrelevant to one another. We set $K_{sim}$ to 2, 2, and 3 for $K = 3$, 5, and 7, respectively. We generated $\mu_{jk}$ for cell types $k = 1, K_{sim} + 1, \dots, K$ from the beta distribution $beta(3, 6)$ on each CpG site $j$ independently. For each of the remaining cell types $k' = 2, \dots, K_{sim}$, we randomly selected 20% of the CpG sites and drew their $\mu_{jk'}$s independently from $beta(3, 6)$; and for the remaining 80% of CpG sites, we let their $\mu_{jk'}$ be $\mu_{j1}$ plus a very small randomness, thus inducing the similarities among cell types 1 to $K_{sim}$.

Discrete and continuous phenotypes. We further generated a discrete and a continuous phenotype $\mathbf{x} = (\mathbf{x}_1, \mathbf{x}_2)^T$ for each individual $i$ $(i = 1, \dots, n)$. We let the first $n/3$ individuals be the control samples with $x_{i1} = 0$ for $i = 1, \dots, n/3$ and the remaining $2n/3$ individuals serve as cases with $x_{i1} = 1$ for $i = n/3 + 1, \dots, n$. The continuous phenotypes $\mathbf{x}_2 = (x_{12}, \dots, x_{i2}, \dots, x_{n2})^T$ were independently drawn from a $Unif(20, 50)$ to act as age.

Phenotype effects with different magnitudes and directions. We then simulated the phenotype effect $\beta_{jk\ell}$ of each phenotype $\ell$ on CpG site $j$ in cell type $k$. For the true null cases, all of the $\beta_{jk\ell}$s are zero. For a true alternative setting, we set nonzero phenotype effects as follows.

For phenotype 1—the case/control status, we let it affect the first 10 CpG sites in all of the cell types: $\beta_{jk1} \neq 0$ for $j = 1, \dots, 10$ and $k = 1, \dots, K$. We then assumed that the next 10 CpG sites were influenced by the disease status in the first $K_{sim}$ cell types which come from the same lineage but not the other cell types: $\beta_{jk1} \neq 0$ $(k = 1, \dots, K_{sim})$ and $\beta_{jk1} = 0$ $(k = K_{sim} + 1, \dots, K)$ for any $j = 11, \dots, 20$. Furthermore, for cell type $k \in \{K_{sim} + 1, \dots, K\}$, we let the disease status affect CpG sites $j = 20 + 10(k - K_{sim} - 1) + 1, \dots, 20 + 10(k - K_{sim})$ only in cell type $k$. We generated the cell-type-specific effects of age in a similar fashion for CpG site loci 21 to $40 + 10(K - K_{sim})$.

For each nonzero $\beta_{jk1}$, we let $\beta_{jk1} = r_{jk} \cdot \omega_{jk}$, where $\omega_{jk} \sim Unif(0.07, 0.15)$ and $r_{jk}$ takes values of 1 and −1 with equal probabilities. Thus, $\beta_{jk1}$s can have both positive and negative effects. In the same spirit, we generated nonzero $\beta_{jk2}$s with $r_{jk}'$s and $\omega_{jk}'$s where $\omega_{jk}' \sim Unif(0.007, 0.015)$.

Association between phenotypes and cellular compositions. Notice that the phenotypes may be associated with the cellular composition. Therefore, when $K = 3$, we drew $\mathbf{p}_i = (p_{1i}, \dots, p_{Ki})$ from a Dirichlet distribution $Dir(4, 4, 2 + 0.1x_{i2})$ if sample $i$ is a control and $\mathbf{p}_i \sim Dir(4, 4, 5 + 0.1x_{i2})$ if it is a case; when $K = 5$, we let $\mathbf{p}_i \sim Dir(3, 3, 3, 3, 2 + 0.1x_{i2})$ for a control sample and $\mathbf{p}_i \sim Dir(3, 3, 3, 3, 5 + 0.1x_{i2})$ for a case sample; when $K = 7$, we sampled $\mathbf{p}_i \sim Dir(1, 3, 3, 3, 2, 2, 2 + 0.1x_{i2})$ for controls and $\mathbf{p}_i \sim Dir(1, 3, 3, 3, 2, 2, 5 + 0.1x_{i2})$ for cases.

Finally, we generated the observed value $O_{ji}$ for CpG site $j$ of sample $i$ as follows: sample $u_{ijk}$ from $N(\mu_{jk} + \beta_{jk1}x_{i1} + \beta_{jk2}x_{i2}, 0.01^2)$ for $k = 1, \dots, K$; and sample $O_{ji}$ from $N(\sum_{k=1}^{K}u_{ijk}p_{ki}, 0.01^2)$. In case $O_{ji}$ lies outside the interval $(0, 1)$, we truncate it to zero if $O_{ji}$ is lower than zero and to one if $O_{ji}$ is greater than one.

**Semi-simulated dataset including samples with known cell mix proportions.** The GEO dataset GSE110554[24] contains purified cell-type-specific methylation profiles for six cell types: neutrophils, monocytes, B cells, CD4+ T, CD8+ T, and NK. Moreover, GSE110554 includes mixed samples whose methylation signals were aggregated from the six cell types with predetermined cell mix proportions. Therefore, because of the known cell type and cellular proportion information, GSE110554 is an ideal dataset with which to test HIRE's performance.

In GSE110554, the number of mixed samples is much smaller than the typical size of an EWAS and, as discussed in the manuscript, HIRE usually requires hundreds of samples to obtain accurate and stable results. Therefore, to increase the sample size, we first generated a simulated methylation dataset with

600 samples using the purified methylation profiles. We focused on 10k CpG sites, including the 450 IDOL CpG sites, which were previously identified as the optimal library of CpG sites for estimation of leukocyte subtype proportions[24], and another 9550 CpG sites whose methylation values across the purified cell types fell within the range of [0.2, 0.8] and had large standard deviations[11]. We then combined the 600 samples and six mixed samples (generated by method A)[24] available in GSE110554 to compose a semi-simulated dataset.

After applying HIRE to the semi-simulated data, we annotated the estimated cell types based on the methylation profiles from GSE110554. Supplementary Figure 69 shows the heatmap for the Pearson correlation matrix between inferred cell types and the underlying truth. The correlation signals on the diagonal are the strongest in each row. HIRE successfully recovers the six underlying cell types. We also compared the estimated cellular compositions with the underlying true proportions for the six mixed samples. Each panel in Supplementary Fig. 70 displays a scatter plot between the cellular proportion estimates and the true mix proportions for a given cell type; they all indicate that HIRE obtains good estimates for cellular compositions.

**Cell type matching protocol.** Assume that we have the reference methylation profiles for the $H$ annotated cell types. We first denote the methylation profile for cell type $h$ as $\phi_h = (\phi_{1h}, \ldots, \phi_{mh})$. We aim to annotate $\mu_k$ using the references. Following the previous study[33], first, we calculate the cosine similarity, the Pearson correlation, and the Spearman correlation between $\mu_k$ and $\phi_h$ for each cell type $h \in \{1, \ldots, H\}$. Notice that the three similarity measures lie between $-1$ and $1$, and a high positive value indicates great similarity between two vectors. Second, for each similarity measure $\ell$ ($\ell = 1, 2, 3$), we identify the cell type $h_\ell$ that has the maximal degree of similarity with $\mu_k$. If at least two out of the three similarity measures identify the same reference cell type $\tilde{h}$ and their corresponding similarity values are greater than 0.5, then we annotate $\mu_k$ with the reference cell type $\tilde{h}$. Otherwise, $\mu_k$ is believed to belong to a new cell type that is not included in the references. We repeat the above process for each methylation profile $\mu_k$ estimated from HIRE.

**Blood cell references.** The two real data sets analyzed in our applications were obtained from whole blood. Therefore, we prepared the references from a whole blood methylation study[34] with GEO accession code GSE35069. The study included seven isolated blood cell subpopulations—CD4+ T cells, CD8+ T cells, CD14 + monocytes, CD19+ B cells, CD56+ NK cells, neutrophils, and eosinophils—for six individuals. Accordingly, we define the reference profile $\phi_h$ for cell type $h$ as the average methylation profile of these individuals, i.e., $\phi_h := \frac{1}{6}\sum_{i=1}^{6} \phi_{hi}$.

**Data preprocessing.** The RA dataset is publicly available in GEO with accession number GSE42861. The dataset measures the methylation levels of the whole blood. The methylation data have been normalized by Illumina's control probe scaling procedure (see Liu et al.[3] "Illumina 450K microarray data preprocessing" section for details). The dataset includes 689 samples, and the RA status, age, gender, smoking history, and batch information are available for each sample. We removed two samples GSM1051535 and GSM1051691 because their smoking information is missing. CpG sites with a high methylation mean (>0.8) and a low methylation mean (<0.2) were discarded[11,13]. We adjusted the data for batch effects using COMBAT[35]. The correction process was justified because we did not observe a high degree of co-linearity between the RA status and the batches (Supplementary Fig. 68). The 10,000 most variable CpG sites were kept. For the RA status, we denoted RA patients with 1 and the normal control subjects with 0; we represented men with 1 and women with 0; for the smoking history, we used (0, 0, 0) to refer to "never," (1, 0, 0) to "ex," (0, 1, 0) to "current," and (0, 0, 1) to "occasional" smokers.

We downloaded the GALA II dataset from Gene Expression Omnibus (GEO) with accession number GSE77716. The dataset contains the whole-blood DNA methylation beta-values from 573 samples. The beta-values have been normalized by SWAN[36] and corrected for batch effects by COMBAT[35]. There are two types of covariates: gender and ethnicity. Ethnicity includes Mexican, Mixed Latino, Puerto Rican, and Other Latino. Out of the 573 samples, one sample "GSM2057284" has no gender information, so we removed it. As suggested by previous studies[11,13], CpG sites with a mean methylation value of less than 0.2 or higher than 0.8 were filtered out. We selected the 10,000 most variable of the remaining CpG sites. For gender, we denoted men with 1 and women with 0. For the ethnicity variables, we used three dummy variables to represent the four ethnicity categories. In particular, (0, 0, 0), (1, 0, 0), (0, 1, 0), and (0, 0, 1) corresponded to Mexican, Mixed Latino, Puerto Rican, and Other Latino, respectively.

For ReFACTor and EWASHer, according to their rules, we first filtered out CpG sites that were consistently hypomethylated or consistently hypermethylated and then regressed out the known covariates. We finally used the residuals to perform their analysis. Note that in their software these steps are processed automatically. For RefFreeEWAS, SVA, and the unadjusted analysis, the phenotypes and the covariates were regarded as the fixed effects in the regression model. In detail, for ReFACTor, in both GALA II and RA datasets, the cell type number "K" was specified to be six, which was the same as in their paper[13]. For RefFreeEWAS, we fixed the dimensionality of latent space "d" at six in the real data. For SVA, we also fixed the number of surrogate variables to six.

Gene enrichment analysis was carried out on the Broad Institute website http://software.broadinstitute.org/gsea/msigdb/annotate.jsp. The canonical pathways were selected as the basis gene sets, and only pathways with a false discovery rate of less than 0.05 were reported.

**Identifiability of HIRE.** Although the non-negative matrix factorization (NNMF) $\mathbf{O} = \mu\mathbf{P}$ has been widely applied in cell type deconvolution[16], where $\mathbf{O}$ is the observed methylation matrix, $\mu$ is the unknown methylation profile, and $\mathbf{P}$ is the unknown cellular compositions, model identifiability is rarely discussed. During the review period of our paper, Rahmani et al.[37] provided a setting under which the NMMF model is not identifiable.

Why then does NNMF always provide satisfactory cell type deconvolution results in real practice, and why can HIRE estimate all those parameters well? Here, we show mathematically that the HIRE model is identifiable under mild conditions that are easily met in reality.

Let us first introduce some notations and definitions. In the HIRE model, the whole parameter set is denoted by

$\Theta := \{\mathbf{P}_i, \mu_j, \mathbf{B}_\ell^{(j)}, \sigma_{jk}^2, \sigma_{ej}^2 : 1 \le j \le m, 1 \le i \le n, 1 \le k \le K, 1 \le \ell \le q\}$, where $\mathbf{p}_i$ is the cellular composition vector of sample $i$, $\mu_j$ is the baseline methylation vector of CpG site $j$, $\mathbf{B}_\ell^{(j)}$ is the phenotype $\ell$ effect vector on CpG site $j$, $\sigma_{jk}^2$ is the cell-type-$k$ noise variance on CpG site $j$, and $\sigma_{ej}^2$ is the overall noise variance on CpG site $j$.

The observed data in our study are the methylation matrix $\mathbf{O} = \{O_{ij} : 1 \le i \le n, 1 \le j \le m\}$ and the covariate matrix $\mathbf{X} = (\mathbf{x}_1, \ldots, \mathbf{x}_\ell, \ldots, \mathbf{x}_q)$, where $\mathbf{x}_\ell$ is the column vector that indicates phenotype-$\ell$ for the $n$ samples. The observed likelihood function $(\Theta|\mathbf{O}) = \prod_{i=1}^{n}\prod_{j=1}^{m} N(O_{ji} : \mathbf{P}_i^T\mu_j + \sum_{\ell=1}^{q} x_{i\ell}\mathbf{P}_i^T\mathbf{B}_\ell^{(j)}, \sum_{k=1}^{K}\sigma_{jk}^2 P_{ik}^2 + \sigma_{ej}^2)$ (see Eq. (S7) in the Supplementary Methods), where $N(O : \eta, \tau^2)$ indicates the normal density with mean $\eta$ and variance $\tau^2$ at value $O$.

We further define $\mathbf{1}_K = (1, 1, \ldots, 1)^T$ as a $K$-dimension column vector with all entries being one, an $n$ by $K$ matrix $\mathbf{J}_1$ as $\mathbf{1}_n\mathbf{1}_K^T$, and an $n$ by $K$ matrix $\mathbf{J}_{x_\ell}$ as $\mathbf{x}_\ell\mathbf{1}_K^T$ for each $1 \le \ell \le q$. We use $\odot$ to represent the entry-wise matrix product for two matrices $\mathbf{M}$ and $\mathbf{N}$ with the same dimension, i.e., $(\mathbf{M}\odot\mathbf{N})_{ij} := \mathbf{M}_{ij}\mathbf{N}_{ij}$.

*Theorem 1.* If (a) for each cell type $k$, there exists a CpG site $r_k$ such that $\mathbf{B}_\ell^{(r_k)} = 0$ for any phenotype $\ell$ and $\mu_{r_k k} = 1$ while $\mu_{r_k k'} = 0$ for $k' \ne k$, and (b) the cellular compositions $\mathbf{P}$ satisfies that $rank((\mathbf{J}_1 \odot \mathbf{P}^T, \mathbf{J}_{x_1} \odot \mathbf{P}^T, \ldots, \mathbf{J}_{x_\ell} \odot \mathbf{P}^T, \ldots, \mathbf{J}_{x_q} \odot \mathbf{P}^T)) = (q+1)K$ and $rank((\mathbf{1}_n, \mathbf{P}^T) \odot (\mathbf{1}_n, \mathbf{P}^T)) = K+1$, then the HIRE model is identifiable. In other words, $L(\Theta|\mathbf{O}) = L(\widetilde{\Theta}|\mathbf{O})$ for any $\mathbf{O}$ implies $\Theta = \widetilde{\Theta}$.

Proof: First, by integrating out all $\mathbf{O}$ elements except $O_{ji}$, $L(\Theta|\mathbf{O}) = L(\widetilde{\Theta}|\mathbf{O})$ implies $N(O_{ji} : \mathbf{P}_i^T\mu_j + \sum_{\ell=1}^{q} x_{i\ell}\mathbf{P}_i^T\mathbf{B}_\ell^{(j)}, \sum_{k=1}^{K}\sigma_{jk}^2 P_{ki}^2 + \sigma_{ej}^2)$ $= N(O_{ji} : \widetilde{\mathbf{P}}_i^T\widetilde{\mu}_j + \sum_{\ell=1}^{q} x_{i\ell}\widetilde{\mathbf{P}}_i^T\widetilde{\mathbf{B}}_\ell^{(j)}, \sum_{k=1}^{K}\widetilde{\sigma}_{jk}^2\widetilde{P}_{ki}^2 + \widetilde{\sigma}_{ej}^2)$. Because the univariate normal distribution is identifiable, we have

$$\mathbf{P}_i^T\mu_j + \sum_{\ell=1}^{q} x_{i\ell}\mathbf{P}_i^T\mathbf{B}_\ell^{(j)} = \widetilde{\mathbf{P}}_i^T\widetilde{\mu}_j + \sum_{\ell=1}^{q} x_{i\ell}\widetilde{\mathbf{P}}_i^T\widetilde{\mathbf{B}}_\ell^{(j)}, \tag{8}$$

$$\sum_{k=1}^{K}\sigma_{jk}^2 P_{ki}^2 + \sigma_{ej}^2 = \sum_{k=1}^{K}\widetilde{\sigma}_{jk}^2\widetilde{P}_{ki}^2 + \widetilde{\sigma}_{ej}^2. \tag{9}$$

Taking $j = r_k$ in Eq. (8), we have $LHS = \mathbf{P}_i^T\mu_{r_k} + \sum_{\ell=1}^{q} x_{i\ell}\mathbf{P}_i^T\mathbf{B}_\ell^{(r_k)} = \mathbf{P}_i^T\mu_{r_k} = 0 + P_{ki} \cdot 1 + 0 = P_{ki}$ and similarly $RHS = \widetilde{P}_{ki}$, so $P_{ki} = \widetilde{P}_{ki}$, which holds for any $i$ and $k$. Hence, we obtain $\mathbf{P} = \widetilde{\mathbf{P}}$. Next, we rewrite Eq. (8) into a matrix form.

$$(\mathbf{P}_i^T, x_{i1}\mathbf{P}_i^T, \ldots, x_{iq}\mathbf{P}_i^T)\begin{pmatrix}\mu_j \\ \mathbf{B}_1^{(j)} \\ \vdots \\ \mathbf{B}_q^{(j)}\end{pmatrix} = (\mathbf{P}_i^T, x_{i1}\mathbf{P}_i^T, \ldots, x_{iq}\mathbf{P}_i^T)\begin{pmatrix}\widetilde{\mu}_j \\ \widetilde{\mathbf{B}}_1^{(j)} \\ \vdots \\ \widetilde{\mathbf{B}}_q^{(j)}\end{pmatrix}, \quad i = 1, \ldots, n.$$

By combining these $n$ equations, it follows that

$$(\mathbf{J}_1 \odot \mathbf{P}^T, \mathbf{J}_{x_1} \odot \mathbf{P}^T, \ldots, \mathbf{J}_{x_\ell} \odot \mathbf{P}^T, \ldots, \mathbf{J}_{x_q} \odot \mathbf{P}^T)\begin{pmatrix}\mu_j \\ \mathbf{B}_1^{(j)} \\ \vdots \\ \mathbf{B}_q^{(j)}\end{pmatrix}$$
$$= (\mathbf{J}_1 \odot \mathbf{P}^T, \mathbf{J}_{x_1} \odot \mathbf{P}^T, \ldots, \mathbf{J}_{x_\ell} \odot \mathbf{P}^T, \ldots, \mathbf{J}_{x_q} \odot \mathbf{P}^T)\begin{pmatrix}\widetilde{\mu}_j \\ \widetilde{\mathbf{B}}_1^{(j)} \\ \vdots \\ \widetilde{\mathbf{B}}_q^{(j)}\end{pmatrix}. \tag{10}$$

Because the rank of $A := (\mathbf{J}_1 \odot \mathbf{P}^T, \mathbf{J}_{x_1} \odot \mathbf{P}^T, \dots, \mathbf{J}_{x_\ell} \odot \mathbf{P}^T, \dots, \mathbf{J}_{x_q} \odot \mathbf{P}^T)$ is $(q+1)K$ (full column rank), $A$ has a left inverse $A^{-1}$. Multiplying Eq. (10) by $A^{-1}$ from the left on both sides, we obtain $\mu_j = \tilde{\mu}_j$ and $\mathbf{B}_\ell^{(j)} = \tilde{\mathbf{B}}_\ell^{(j)}$ for $1 \le \ell \le q$. Therefore, we have $\mu = \tilde{\mu}$, $\mathbf{B} = \tilde{\mathbf{B}}$.

In addition, because Eq. (9) holds for any $i$, we can also rewrite it into a matrix form.

$$
\begin{pmatrix} 1 & P_{11}^2 & \dots & P_{K1}^2 \\ 1 & P_{12}^2 & \dots & P_{K2}^2 \\ \vdots & \vdots & \ddots & \vdots \\ 1 & P_{1n}^2 & \dots & P_{Kn}^2 \end{pmatrix} \begin{pmatrix} \sigma_{\epsilon j}^2 \\ \sigma_{j1}^2 \\ \vdots \\ \sigma_{jK}^2 \end{pmatrix} = \begin{pmatrix} 1 & P_{11}^2 & \dots & P_{K1}^2 \\ 1 & P_{12}^2 & \dots & P_{K2}^2 \\ \vdots & \vdots & \ddots & \vdots \\ 1 & P_{1n}^2 & \dots & P_{Kn}^2 \end{pmatrix} \begin{pmatrix} \tilde{\sigma}_{\epsilon j}^2 \\ \tilde{\sigma}_{j1}^2 \\ \vdots \\ \tilde{\sigma}_{jK}^2 \end{pmatrix}
$$

The left matrix is equal to $(\mathbf{1}_n, \mathbf{P}^T) \odot (\mathbf{1}_n, \mathbf{P}^T)$ which has a full column rank; therefore, it has a left inverse. Consequently, $\sigma_{\epsilon j}^2 = \tilde{\sigma}_{\epsilon j}^2$ and $\sigma_{jk}^2 = \tilde{\sigma}_{jk}^2$. As a result, $\Theta = \tilde{\Theta}$, and we have proven the identifiability of HIRE. $\square$

Conditions (a) and (b) are easily met for DNA methylation data. Condition (a) requires that for each cell type $k$, there exists a CpG site that is not associated with any phenotype and is only methylated in cell type $k$ but not methylated in any other cell type. Given the 450K CpG sites assayed by the microarray, we can expect that such CpG sites are not absent at all. Moreover, condition (a) can also be relaxed to the condition that for each cell type $k$, there exists a CpG site $r_k$ such that $\mathbf{B}_\ell^{(r_k)} = 0$ for any phenotype $\ell$ and $\mu_{r_k k} = 1$ while $\mu_{r_k k'} = 0$ for $k' \neq k$ or there exists a CpG site $r_k$ such that $\mathbf{B}_\ell^{(r_k)} = 0$ for any phenotype $\ell$ and $\mu_{r_k k} = 0$ while $\mu_{r_k k'} = 1$ for $k' \neq k$. The proof follows in a similar manner.

For condition (b), intuitively, the rank requirement of $(\mathbf{1}_n, \mathbf{P}^T) \odot (\mathbf{1}_n, \mathbf{P}^T)$ asks the cellular compositions to vary across subjects, which guards against the case in which all the subjects have the same cellular compositions and hence no cell type deconvolution is possible; the rank requirement on $(\mathbf{J}_1 \odot \mathbf{P}^T, \mathbf{J}_{x_1} \odot \mathbf{P}^T, \dots, \mathbf{J}_{x_\ell} \odot \mathbf{P}^T, \dots, \mathbf{J}_{x_q} \odot \mathbf{P}^T)$ is the same requirement as those in a standard linear regression, which requires that no collinearity exists among the covariates. Because the sample size $n$ is much larger than the underlying cell type number $K$ and the phenotype number $q$, the two rank requirements can commonly be satisfied in reality.

**Reporting summary**. Further information on research design is available in the Nature Research Reporting Summary linked to this article.

## Data availability

The RA whole blood methylation dataset is available in the Gene Expression Omnibus (GEO) with the accession number GSE42861. The GALA II whole blood methylation dataset can be downloaded from GEO with the accession number GSE77716. The accession number for the blood cell references is GSE35069. The purified methylation data and mixed samples used to generate the semisimulated dataset are taken from GSE110554.

## Code availability

The software and detailed documentations are available on Bioconductor with the software HIREewas page [http://www.bioconductor.org/packages/release/bioc/html/HIREewas.html].

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

## Acknowledgements

X.L. was supported in part by Fundamental Research Funds for the Central Universities, the Research Funds of Renmin University of China (19XNLG08), and the fund for building world-class universities (disciplines) of Renmin University of China. X.L. is grateful for the Hong Kong Ph.D. Fellowship (PF13-11656) from the Hong Kong Research Grants Council when X.L. was a Ph.D. student at the Chinese University of Hong Kong. C.Y. was supported in part by the National Science Funding of China [61501389]; the Hong Kong Research Grants Council [22302815, 12316116, 12301417 and 16307818]; The Hong Kong University of Science and Technology [startup grant R9405 and IGN17SC02]. Y.W. was supported in part by the Early Career Scheme 24301416 and General Research Fund 14306417 from the Research Grants Council of the Hong KongSpecial Administrative Region and Direct Grants from the Research Committee of the Chinese University of Hong Kong. We acknowledge Mingxuan Cai for his contribution to part of the HIREewas code. We are grateful to the High-Performance Computing Platform of Renmin University of China and the Department of Statistics at the Chinese University of Hong Kong for providing computing resources.

## Author contributions

Y.W. and C.Y. conceived the study. X.L. and Y.W. developed the method. Y.W. and X.L. proved the model identifiability. X.L. implemented the algorithm and prepared the software package. X.L. and C.Y. analyzed the data. X.L., Y.W., and C.Y. wrote the paper.

## Additional information

**Competing interests:** The authors declare no competing interests.

