## [Peer Review File · Nature Communications]

Reviewers' comments:

Reviewer #1 (Remarks to the Author):

In this manuscript, the author proposed a new method that can detect cell-type-specific association between DNA methylation and phenotype using a hierarchical modeling approach. In their model, they assumed that the effect of cellular composition on methylation level is multiplicative rather than additive. In addition to separating the association from a mixture of cell types, the author demonstrated that this method can also increase the power in of detecting aggregated effect of phenotype on methylation level compared with existing methods. However, there are several major problems that need further revision.

1. The author should perform simulations under the null hypothesis and report the type I error of the proposed method compared with other method accounting for cell-type heterogeneity.
2. The authors should report QQ-plots of the p-values under both null and alternative hypotheses, which can better illustrate the model performance than a density plot of p-values dose.
3. The authors should report the bias of parameter estimates in their EM algorithm, which is the difference between the true and estimated value of parameters.
4. The authors did simulation in only one situation, which is not comprehensive. I would suggest the author do more simulations by varying the effect size (both positive and negative and positive effect), number of cell types and sample size.
5. In simulation, the authors truncated the observed methylation values to be within zero to one to imitate the real data. However, the model is based on normal assumption of error terms. Hence the author should give more discussion of their assumption and how the difference between assumption and data would influence the model performance.

Reviewer #2 (Remarks to the Author):

In the manuscript entitled "Detection of cell-type-specific risk-CpG sites in epigenome-wide associations studies", the authors describe a statistical method (HIRE) that allows for the identification of risk-associated CpGs within individual cell-types when only DNA methylation signatures profiled over a complex mixture of those cell types are available (e.g., whole-blood derived DNA methylation signatures). The manuscript addresses a key issue that is believed by many within the epigenomics research community to be amongst the foremost challenges currently faced by EWAS; namely, cell heterogeneity and its impact on the statistical analysis of DNA methylation data. The proposed method represents an important advancement for the DNA methylation research community that has potential to open up new lines of inquiry when analyzing methylation data profiled in heterogeneous tissue types. The authors should be commended for the statistical rigor employed in the development of their methodology and for the very clear description of their approach.

While this reviewer is quite optimistic about this work and its potential implications for EWAS, there are several very important issues that need to be addressed in order for HIRE to have its intended impact and reach its full potential. In what follows, I summarize my major concerns and provide suggestions for additional analyses that

1. While the methodology itself appears sound and statistically rigorous, the implicit "reference-free" structure assumed in HIRE severely limits any kind of biologically meaningful interpretation of the ensuing results. For example, the user does not know what the cell types actually are, rather they are merely arbitrarily labeled cell-type 1, cell-type 2,..., cell-type K (where K is the assumed or empirically determined number of cell types). From a purely biological standpoint, if the user does not know what the cell types are, then what use is it to know the specific CpG sites

within those cell types that are differentially methylated based on some phenotype or exposure? To put it a different way, imagine that a specific CpG is identified by HIRE as being significantly associated with some phenotype or exposure, X only in cell-type 2. If the identity of cell-type 2 is unspecified/unknown, then what kind of meaningful conclusions can be reached by the investigator in terms of the underlying biology that governs the process being studied? I understand and appreciate the authors argument that the reference methylation signatures for the cells underlying certain tissue types have yet to be determined and therefore cannot be used, however they are available for some tissue types, most notably, blood. Because blood is one of the most commonly used tissue types used EWAS, not leveraging leukocyte-specific reference methylation signatures seems like a major missed opportunity. What I am suggesting is that the authors consider a reference-based + reference-free hybrid approach where the application of HIRE to whole-blood DNA methylation data would utilize existing reference methylation signatures for isolated leukocyte cell types as a means toward allowing one to identify risk-CpGs within granulocytes, monocytes, NK cells, B cells, CD4T cells, and CD8T cells. In doing so, the user knows the actual identity of the cell type (rather than an arbitrary label), which would lead to more meaningful biological interpretations of the results generated by HIRE.

2. The simulation study is quite narrow in scope (only one simulation scenario was considered), not to mention the assumptions being made in the generation of the methylation data are somewhat unrealistic given what we know about DNA methylation profile for the cell types underlying certain tissues. Take blood for example, the lineage relationships between leukocyte subtypes is known as is the fact that cell types from a shared lineage have a methylation signature that is more similar as compared to cell types with differing lineages (e.g., lymphocytic versus myeloid derived cell types). However, in the simulation study the methylation signature is being simulated independently across cell types (page 14). Perhaps the dependency in methylation across cell types has no or little impact on the ability of HIRE to identify cell-type specific risk-CpGs, but the absence of such a simulation scenario makes this difficult, if not impossible, to ascertain. It would also be useful to see a simulation scenario where the risk-CpGs are the same across cell types, but vary across cell types according to their magnitude/direction of association. For example CpG1 may be associated with some phenotype within each cell type, however the magnitude and/or direction of its association varies across the cells. Will HIRE be able to correctly identify such CpGs or would the differing direction of association across impair their detection? One other aspect that could and should be addressed through simulation is the threshold for a given cell's proportion for which HIRE is unable to detect risk-CpGs for that cell type. For example, if a given cell only represents a small fraction of the underlying cell composition (e.g., 2-5%), is HIRE still able to detect risk-CpGs within that cell type. Does that depend on how strong the association is for those specific CpGs (e.g., moderate to large delta-betas)? To make a more convincing argument for the HIRE methodology, it behooves the authors to scale up their simulation studies to assess HIRE across a range of possible scenarios.

3. My final major comment is that the Discussion section makes no mention of the limitations of the proposed methodology. When does the method fail, are there scenarios for which HIRE is not particularly well suited, what are the limitations based on the datasets used to assess its performance, etc. In addition to discussing limitations inherent to the methodology itself, one very obvious limitation of this work that is only briefly mentioned, is the application of HIRE a "gold-standard" data set. While the definition of "gold-standard" may be open to debate, what is not debatable is the fact that none of the data sets used in the present examination provide conclusive proof that HIRE is indeed capable of identifying bonafide risk-CpGs for individual cell types in a "real" DNA methylation data set. It would be incredibly helpful if the authors could describe what they believe would be the ideal data set in which to illustrate/validate their methodology. For example, if purified cell types were isolated, subject to site-specific epigenetic editing on per cell type basis (e.g., so the investigator knows the ground truth about what CpGs are differentially methylated within specific subpopulation of cells), and then mixed in predetermined proportions (such as in an artificial reconstruction experiment) followed by DNA methylation assessed over the mixed population of cells, it would seem that such an experiment would be pretty close to a "gold-standard" data set. The authors should comment on the feasibility of such an experiment and why this was not conducted in the present examination.

Reviewer #3 (Remarks to the Author):

The paper proposes a statistical method, HIRE, for cell type deconvoluting from methylation data and for detecting cell type specific methylation associations with a risk factor. The proposed method relies on a joint factor analysis model and an EM algorithm to first extract cell type proportions, and then for each cell type in turn, relies on a t/F test to detect differentially methylated sites in the specific cell type. With simulations and two real data applications, the authors show that their method can be powerful. Overall, while the problem studied in the present paper can be of potential interest, I am not convinced that the method proposed in the paper represents a significant advance over the previous approaches. In addition, it is unclear whether the methods HIRE compared to are properly applied in the present study and whether HIRE works better than other approaches in the two real data sets. My main comments are listed below:

While the authors claim that their method is the first method to detect risk-CpG sites for individual cell types, I believe this so-called main novelty is an exaggeration. To the best of my knowledge, the problem of detecting cell type specific risk CpG sites has been relatively well studied by several previous studies. Indeed, the method proposed in the current paper, HIRE, is very similar to the factor model proposed in Houseman et al Bioinformatics 2014 and is also related to the nonnegative factorization model proposed in Houseman et al., BMC Bioinformatics 2016. Either of these two Houseman et al. methods, or some other relevant methods (e.g. reviewed in <https://academic.oup.com/hmg/article/26/R2/R216/3979364>), can be used to detect cell type specific risk-CpG sites by using a two-step procedure. The two-step procedure, just like the one used in the present paper, would extract methylation profile in a cell type specific fashion (i.e. the factor loading matrices in these methods), which can be further used to detect differential methylation in a cell type specific fashion by standard differential methylation methods, such as the t-test or F-test used in the present study. From this aspect, the only novel part of the current paper is that the proposed HIRE method can incorporate differential methylation information into cell deconvolution, but I feel this is a rather incremental improvement and I am unsure at this stage how much practice improvement this can bring. Indeed, if you look at the power comparison shown in Figure 2A, even using simulations that completely favor HIRE, HIRE still has an almost identical power as one of the Houseman et al methods. In addition, HIRE sacrifices the non-negative factorization advantage presented in Houseman et al., BMC Bioinformatics 2016. As a consequence, the cell type proportion estimates from HIRE, unlike those from Houseman et al., BMC Bioinformatics 2016 method, are not guaranteed to be nonnegative, making interpretation difficult and the subsequently statistical testing likely sub-optimal.

It is unclear why the authors choose to compare their methods to several unrelated methods that include SVA, EWASHer, ReFACTor. These three methods are designed to control for hidden confounding effects that are correlated with cell compositions. In contrast, HIRE and RefFreeEWAS are designed for reference-free cell deconvolution and the subsequent differential methylation analysis. Because these two sets of methods aim to deal with very different problems, I would suggest the authors to remove the comparison to SVA, EWASHer and ReFACTor (or add additional simulations and real data results to show how HIRE and RefFreeEWAS perform when hidden confounding factors are correlated with cell type compositions).

I don't quite follow the argument on line 38 on the importance of "multiplicative" vs "additive" effects. If I understand it correctly, running a two-step procedure using Houseman et al. methods and testing the association between cell type specific methylation levels with the phenotype would automatically assume a multiplicative effect as in the HIRE model.

The current paper lacks details on how they fitted the RefFreeEWAS package. In particular, RefFreeEWAS package contains two methods that include the factor model proposed in Houseman et al Bioinformatics 2014 and the nonnegative factorization model proposed in the new Houseman et al BMC Bioinformatics 2016 method. Which of the two methods did you use? After cell deconvolution, once you extracted cell type specific methylation profile from the factor loading matrices, did you perform an additional normalization, or did you use the factor loading matrix values directly? Or does it matter? In the differential methylation test, did you examine one cell type at a time by treating cell type specific methylation profile as the outcome variable and

phenotype as the covariate, or did you use the reverse order by treating the methylation levels for all cell types as covariates and phenotypes as the outcome? Is the differential methylation test conditional on other covariates (i.e. age) or not? Did you use t-test, F-test, or other methods to detect differential methylation? Since HIRE and RefFreeEWAS are very similar to each other, why would the p-value distributions differ that much between the two methods (Figure 2d vs 2g and Figure 3c vs 3f)? The huge p-value distribution difference worries me and makes me wonder whether the RefFreeEWAS package was used properly.

It would be important to compare to a simple improvement of RefFreeEWAS, where you run RefFreeEWAS first to identify differentially methylated sites. Afterwards, you can exclude these differentially methylated sites (or if too many, exclude the top differentially methylated sites) and focus on the remaining null sites to run RefFreeEWAS again to estimate cell type compositions and cell type specific methylation profiles. Finally, you can perform another round of differential methylation analysis on all sites to detect differential methylated sites. I was wondering if this simple approach would work well compared to the more complicated joint EM algorithm used in HIRE.

Also, in the simulations, instead of ROC curve, could you show power at the usual Bonferroni threshold or a typical FDR? Is HIRE still more powerful than RefFreeEWAS at these commonly used thresholds? If you run permutation, would the p-values from either HIRE and RefFreeEWAS in the permuted well behaved? Also, what does the p-value distribution looks like under the simulated null, both in terms of histogram and qq-plot of $-\log_{10}p$ values?

Is there a way to ensure that the cell composition proposition estimates to be non-negative in HIRE? What are the consequences of not keeping the proportion estimates non-negative? Also, are the estimates of cell composition from HIRE similar to that from the two different methods implemented in RefFreeEWAS, in both simulations and real data application?

One can also apply RefFreeEWAS and BIC to select number of cell types. Are the cell type number from RefFreeEWAS consistent with HIRE in both simulations and real data sets?

For real data applications, how do you normalize the data before analysis? Will different normalization results in different results? Or will different normalization favor different methods?

How do you declare significance in the real data? Can you run permutations to construct a null p-value distribution in the real data? Which method works the best in the two real data sets? How many sites did RefFreeEWAS detect? What's the overlap between the detected sites between RefFreeEWAS and HIRE?

The second data set contains population stratification. How can you make sure that these population stratification is well controlled for? Population structure can influence cell composition and affect methylation level through paths other than cell composition. Therefore, if you don't control for population structure well, then the cell composition estimates are likely inaccurate. But if you over-correct for population population structure, then you would also likely remove cell compositions together with population structure. How do you check all these in real data?

We thank the three reviewers and the Editor for their constructive comments and invaluable suggestions. We appreciate the positive comments, such as “*address a key issue*,” “*an important advancement*,” and “*open up new lines of inquiry*.” We have taken the reviewers’ suggestions very seriously, conducted extensive simulation studies to test the performance of HIRE, developed a protocol to match the estimated cell types with known cell types, further clarified the difference between HIRE and existing methods, elaborated why existing methods cannot provide cell-type-specific risk-CpG sites, added a discussion on the limitations of HIRE, and revised our manuscript and Supplementary Materials accordingly.

Below, we first summarize our updated simulation studies and then provide point-by-point responses to the reviewers’ concerns. The comments from the three reviewers are displayed in *blue italics*. In the main manuscript, we highlighted our modifications in *red*.

0 Updated Simulation Studies

To better address the concerns raised by the reviewers regarding the simulations, we created 18 simulation settings in the revised manuscript to compare the performance of HIRE with five existing methods. In this section, we detail the data generation processes and report the results.

0.1 Data Generation

The 18 simulation settings. We compared the performance of HIRE with five previous methods—unadjusted analysis, SVA, RefFreeEWAS, EWASHer, and ReFACTor—in 18 simulation settings. We set the sample size n to 180, 300, and 600 and let the underlying cell type number K be 3, 5, and 7. For each pair of (n, K) , we investigated two scenarios: (1) all of the phenotype effects β_{jkl} s are zero—the “**true null**” case—to evaluate each method’s ability to control false positives; and (2) a small portion of β_{jkl} s are non-zero—the “**true alternative**” case—to study the power of each method to detect risk-CpG sites. As a result, we have in total 3 (the number of sample sizes) \times 3 (the number of cell types) \times 2 (the “true null” case and the “true alternative” case) = 18 simulation settings. For each setting, we considered 10,000 CpG sites and simultaneously accounted for the following factors.

Cell lineage. We first constructed the baseline methylation matrix $\boldsymbol{\mu} = (\mu_{jk})_{m \times K}$, where each column corresponds to the baseline methylation levels of a cell type. To mimic the phenomenon that cell types from the same lineage have similar methylation profiles, we assumed that K_{sim} out of the total K cell types were similar. Specifically, without loss of generality, we assumed that the first K_{sim} cell types were from the same cell lineage and the rest $K - K_{sim}$ cell types are irrelevant to one another. We set K_{sim} to 2, 2, and 3 for $K = 3, 5,$ and $7,$ respectively. We generated μ_{jk} for cell

types $k = 1, K_{sim} + 1, \dots, K$ from the beta distribution $beta(3, 6)$ for each CpG site j independently. For each of the remaining cell types $k' = 2, \dots, K_{sim}$, we randomly selected 20% of the CpG sites and drew their $\mu_{jk'}$ s independently from $beta(3, 6)$; and for the remaining 80% of CpG sites, we let their $\mu_{jk'}$ be μ_{j1} plus a very small randomness, thus inducing the similarities among cell types 1 to K_{sim} .

Discrete and continuous phenotypes. We further generated a discrete and a continuous phenotype $\mathbf{x} = (\mathbf{x}_1, \mathbf{x}_2)^T$ for each individual i ($i = 1, \dots, n$). We let the first $n/3$ individuals be the control samples with $x_{i1} = 0$ for $i = 1, \dots, n/3$ and the remaining $2n/3$ individuals serve as cases with $x_{i1} = 1$ for $i = n/3 + 1, \dots, n$. The continuous phenotypes $\mathbf{x}_2 = (x_{i2}, \dots, x_{i2}, \dots, x_{n2})^T$ were independently drawn from a $Unif(20, 50)$ to act as age.

Phenotype effects with different magnitudes and directions. Subsequently, we simulated the phenotype effect β_{jkl} of each phenotype ℓ on CpG site j in cell type k . For the “true null” cases, all of the β_{jkl} s are zero. For a “true alternative” setting, we set nonzero phenotype effects as follows.

For phenotype 1—the case/control status, we let it affect the first 10 CpG sites in all of the cell types: $\beta_{jk1} \neq 0$ for $j = 1, \dots, 10$ and $k = 1, \dots, K$. Then, we assumed that the next 10 CpG sites are influenced by the disease status in the first K_{sim} cell types which come from the same lineage but not the other cell types: $\beta_{jk1} \neq 0$, $k = 1, \dots, K_{sim}$, and $\beta_{jk1} = 0$, $k = K_{sim} + 1, \dots, K$, for any $j = 11, \dots, 20$. Furthermore, for cell type $k \in \{K_{sim} + 1, \dots, K\}$, we let the disease status affect CpG sites $j = 20 + 10(k - K_{sim} - 1) + 1, \dots, 20 + 10(k - K_{sim})$ only in cell type k . We generated the cell-type-specific effects of age in a similar fashion for CpG site loci 21 to $40 + 10(K - K_{sim})$. Figure R1(a) and (c) show an illustration of cell-type-specific risk-CpG sites.

For each nonzero β_{jk1} , we let $\beta_{jk1} = r_{jk} \cdot \omega_{jk}$, where $\omega_{jk} \sim Unif(0.07, 0.15)$ and r_{jk} takes values at 1 and -1 with equal probabilities. Thus, β_{jk1} s can have both positive and negative effects. In the same spirit, we generated nonzero β_{jk2} s with r'_{jk} s and ω'_{jk} s where $\omega'_{jk} \sim Unif(0.007, 0.015)$.

The association between phenotypes and cellular compositions. Notice that the phenotypes may be associated with the cellular compositions. Therefore, when $K = 3$, we drew $\mathbf{p}_i = (p_{1i}, \dots, p_{Ki})$ from a Dirichlet distribution $Dir(4, 4, 2 + 0.1x_{i2})$ if sample i is a control and $\mathbf{p}_i \sim Dir(4, 4, 5 + 0.1x_{i2})$ if it is a case; when $K = 5$, we let $\mathbf{p}_i \sim Dir(3, 3, 3, 3, 2 + 0.1x_{i2})$ for a control sample and $\mathbf{p}_i \sim Dir(3, 3, 3, 3, 5 + 0.1x_{i2})$ for a case sample; and when $K = 7$, we sampled $\mathbf{p}_i \sim Dir(1, 3, 3, 3, 2, 2, 2 + 0.1x_{i2})$ for controls and $\mathbf{p}_i \sim Dir(1, 3, 3, 3, 2, 2, 5 + 0.1x_{i2})$ for cases.

Finally, we generated the observed value O_{ji} for CpG site j of sample i as follows: sample u_{ijk} from $N(\mu_{jk} + \beta_{jk1}x_{i1} + \beta_{jk2}x_{i2}, 0.01^2)$ for $k = 1, \dots, K$; and sample O_{ji} from $N(\sum_{k=1}^K u_{ijk}p_{ki}, 0.01^2)$. In case O_{ji} is out of the interval $(0, 1)$, we truncated it

to zero if O_{ji} is lower than zero and to one if O_{ji} is greater than one.

0.2 Results

For each simulation setting, we evaluated the performance of HIRE and the five previous methods using the following measures.

P-value density plot. We drew the density plots of p-values to visualize the distribution of p-values. In the multiple hypothesis testing, the p-values from the truly null features should follow a uniform distribution on $(0, 1)$, whereas those for the truly alternative features concentrate near zero (Storey and Tibshirani, 2003). See Figure R2 for an example.

Q-Q plots. We further plotted the Q-Q plots of $-\log_{10}$ p-values to check whether the distribution of p-values behaves like a uniform distribution. Note that, in the Q-Q plot, as HIRE provides cell-type-specific p-values, there are in total mKq p-values for HIRE, whereas any other method has m p-values. See Figure R4 for an example.

TPR and FPR. We reported the true positive rate (TPR/power) and the false positive rate (FPR/Type I error) for all of the methods at the aggregated level in Table R1. In addition, for HIRE, we provided the cell-type-specific TPR and FPR in Table R2.

Estimate biases. Regarding HIRE, we are mainly interested in the cellular compositions and the phenotype effects. To better visualize the estimation bias, we plotted the average bias for cellular compositions in individuals, $\frac{1}{K} \sum_{k=1}^K |\hat{p}_{ki} - p_{ki}|$ ($i = 1, \dots, n$), for phenotype effects of the case-control status, $\frac{1}{K} \sum_{k=1}^K |\hat{\beta}_{jk1} - \beta_{jk1}|$ ($j = 1, \dots, m$), and for phenotype effects of the age, $\frac{1}{K} \sum_{k=1}^K |\hat{\beta}_{jk2} - \beta_{jk2}|$ ($j = 1, \dots, m$). See Figure R5 for an example.

Cell-type-specific association patterns. As HIRE is able to provide cell-type-specific p-values to detect risk-CpG sites, we plotted the heatmap of the $-\log_{10}$ p-values. A higher degree of darkness means a smaller p-value. See Figure R1(b) and (d).

Table R1 shows that under the “true null” HIRE, EWASHer, and ReFACTor control the FPR very well, all of which are not greater than 0.05%. In comparison, Ref-FreeEWAS often has FPRs larger than 0.1%, performing not as well as HIRE, and the unadjusted analysis and SVA further suffer from the dramatic inflation of false positives. For the “true alternative” settings, given that the FPRs are well-controlled, with FPRs below 0.05%, HIRE achieves the highest TPR among all of the methods in every simulation setting. As expected, as sample sizes increase, HIRE’s power increases. For example, when there are five cell types in the data, HIRE can identify 89.6% of the risk-CpG sites with 300 samples, and HIRE is able to detect almost all of the risk-CpG sites when the sample size reaches 600, which is a typical sample size for an EWAS. Although EWASHer and ReFACTor have low FPRs, they miss a

large proportion of risk-CpG sites. EWASHer’s maximum TPR is only 35.33%, and ReFACTor’s maximum TPR is slightly higher than 60%. However, in those cases, HIRE’s power is greater than 95%. Consistent with the “true null” scenario, under the “true alternative,” RefFreeEWAS has inflated FPRs compared to HIRE, and the unadjusted analysis and SVA always have explosive false positives. Therefore, HIRE substantially improves the power of detecting associations at the aggregated level in comparison with existing methods.

We further study the distribution of p-values for each method. As an illustration, Figures R3 and R4 show the Q-Q plots for the $K = 3$ and $n = 180$ cases under the “true null” and the “true alternative,” respectively. Under the “true null,” ideally the p-values should ideally follow a uniform distribution, with the Q-Q line matching the diagonal line $y = x$. It turns out that the Q-Q lines of HIRE, EWASHer, and ReFACTor under the “true null” are very close to the diagonal line $y = x$. In contrast, RefFreeEWAS, unadjusted analysis, and SVA deviate from the ideal line $y = x$, consistent with their inflation of FPRs as observed in Table R1. Under the “true alternative,” as only a small proportion of CpG sites are at risk, and as most CpG sites have no association with phenotypes, it is expected that most p-values follow a uniform distribution while the p-values corresponding to risk-CpG sites are very small. HIRE’s Q-Q line under the “true alternative” follows exactly the expected pattern with only its tail deviating from the diagonal line. RefFreeEWAS, EWASHer, and ReFACTor have similar patterns. However, the Q-Q lines for the unadjusted analysis and SVA’s depart from the $y = x$ substantially even from the origin.

In addition to the traditional association detection at the aggregated level, HIRE is able to identify the association for each CpG site with the phenotypes under each cell type. Table R2 shows the FPR and TPR of HIRE in each cell type for different simulation settings. Such fine analysis is not available with other methods. When $K = 3$ and $n = 180$, HIRE accurately detected the risk-CpG sites associated with the disease status with the TPR more than 83% and the FPR less than or equal to 0.01% in all of the three cell types. Similarly, most of the CpG sites affected by age are also correctly identified in each cell type. HIRE’s learned cell-type-specific association patterns closely matches the underlying true associations (see Figure R1). We further investigate the estimate biases. Figure R5 shows that most of the cellular composition and phenotype effect estimates have little bias. Therefore, HIRE can provide accurate estimates and is powerful in terms of detecting cell-type-specific risk-CpG sites.

Figure R1: The true and learned association patterns for the “true alternative” when $K = 3$ and $n = 180$. The darkness of panel (b) and (d) represents the value of $-\log_{10}\text{p-values}$ from HIRE. To highlight the true and detected signals, only the first 100 CpG sites are shown. The same panels (a) and (b) are shown as Figure 2 (b) and (c) in the main manuscript. The same panels (c) and (d) are shown as Supplementary Figure 18.

Figure R2: The p-value density plots under the “true alternative” when $K = 3$ and $n = 180$ for: (a) HIRE, (b) unadjusted analysis, (c) SVA, (d) RefFreeEWAS, (e) EWASHer, and (f) ReFACTor. The same figure is shown as Figure 2 (d-i) in the main manuscript.

Figure R3: The Q-Q plots for p-values under the “true null” when $K = 3$ and $n = 180$ from: (a) HIRE, (b) unadjusted analysis, (c) SVA, (d) RefFreeEWAS, (e) EWASher, and (f) ReFACTor. The same figure is shown as Figure 2 (j-o) in the main manuscript.

Figure R4: The Q-Q plots for p-values under the “true alternative” when $K = 3$ and $n = 180$ from: (a) HIRE, (b) unadjusted analysis, (c) SVA, (d) RefFreeEWAS, (e) EWASHer, and (f) ReFACTor. The same figure is shown as Figure 2 (j-o) in the main manuscript.

Figure R5: The scatter plots for the estimated cell proportions by HIRE and the underlying truth under the “true alternative” when $K = 3$ and $n = 180$ for (a) cell type 1; (b) cell type 2; and (c) cell type 3. The average bias plots for (d) cellular compositions, (e) the case/control effects, and (f) the age effects. The panels (a-d) are shown as Supplementary Figure 45. The panels (e) and (f) are shown as Supplementary Figure 46.

Table R1: The performance of HIRE, SVA, unadjusted analysis, RefFreeEWAS, EWASHer and ReFACTor in simulation studies in detecting risk-CpG sites at the aggregated level. For the “true null” cases where no CpG site is at risk, the average of the false positive rates (FPRs) based on five replicates is reported. For the “true alternative” cases, the averages of the FPRs and the true positive rates (TPRs) based on five replicates are reported. The number of CpG sites at risk is 30, 50, and 60 for the cell type number $K = 3, 5, \text{ and } 7$, respectively. HIRE calls a CpG site as significant at the aggregated level if it is at risk in at least one cell type. For all of the methods, we used Bonferroni correction to control the family-wise error rate (FWER) to be less than $\alpha = 0.01$. For HIRE, as it can provide p-values of CpG sites for all of cell types and phenotypes, the p-value threshold to call “significant” is $\alpha/(mKq)$, where $m = 10,000$ is the number of CpG sites and $p = 2$ is the phenotype number. For the other five methods, the p-value threshold is set to α/m . FPRs less than or equal to 0.05% are shown in bold. In the “true alternative” settings, the maximum TPRs with FPRs controlling at 0.05% are highlighted. Notice that “0” represents exact zero, and “0.00%” indicates a very small positive number that is rounded to zero using four decimal places. The same table is shown as the Table 1 in the main manuscript.

	cell type number	sample size	HIRE	SVA	unadjusted	RefFreeEWAS	EWASHer	ReFACTor
True null	$K = 3$	$n = 180$	FPR	0.35%	21.72%	0.12%	0	0
		$n = 300$	FPR	3.13%	53.40%	0.12%	0.00%	0
		$n = 600$	FPR	22.06%	77.2%	0.1%	0.00%	0
	$K = 5$	$n = 180$	FPR	0.3%	18.5%	0.13%	0	0
		$n = 300$	FPR	1.54%	36.89%	0.08%	0	0
		$n = 600$	FPR	8.09%	55.66%	0.12%	0	0
	$K = 7$	$n = 180$	FPR	0.00%	5.13%	0.11%	0	0
		$n = 300$	FPR	0.00%	22.84%	0.12%	0	0
		$n = 600$	FPR	0	48.12%	0.11%	0	0
True alternative	$K = 3$	$n = 180$	FPR	1.24%	32.20%	0.11%	0	0
		$n = 300$	TPR	87.33%	60.67%	79.33%	21.33%	60.67%
		$n = 600$	FPR	7.60%	65.03%	0.09%	0	2.21%
	$K = 5$	$n = 180$	TPR	86%	79.33%	63.33%	24.67%	56.67%
		$n = 300$	FPR	0.03%	16.91%	0.11%	0.00%	1.34%
		$n = 600$	TPR	100%	84%	88.67%	35.33%	45.33%
	$K = 7$	$n = 180$	FPR	0.00%	2.70%	20.06%	0	0
		$n = 300$	TPR	66%	80.8%	35.2%	6.4%	41.6%
		$n = 600$	FPR	0.01%	1.76%	36.11%	0	0.01%
$K = 5$	$n = 180$	TPR	89.6%	86.4%	69.2%	11.2%	53.6%	
	$n = 300$	FPR	0.01%	15.31%	56.25%	0	0.11%	
	$n = 600$	TPR	98.4%	82.8%	62%	18%	62.8%	
$K = 7$	$n = 180$	FPR	0	0.37%	9.30%	0	0	
	$n = 300$	TPR	43%	58.67%	35%	5%	26%	
	$n = 600$	FPR	0.00%	1.08%	26.67%	0	0	
$K = 7$	$n = 180$	TPR	63.33%	73%	45%	5%	35.67%	
	$n = 300$	FPR	0.04%	25.80%	56.76%	0	0.00%	
	$n = 600$	TPR	82.67%	83%	66%	5%	51.67%	

Table R2: The performance of HIRE in detecting cell-type-specific risk-CpG sites in the “true alternative” cases. The results are based on five replicates for each setting. A CpG site is claimed to be “significant” in a given cell type if its p-value is less than $\alpha/(mKq)$. The same table is shown as the Table 2 in the main manuscript.

Phenotype	cell type number	sample size	cell types								
			1	2	3	4	5	6	7		
Disease status	$K = 3$	$n = 180$	FPR	0.01%	0.00%	0.01%					
			TPR	83%	85%	92%					
		$n = 300$	FPR	0.02%	0.02%	0.04%					
			TPR	74%	85%	95%					
		$n = 600$	FPR	0.03%	0.03%	0.05%					
			TPR	99%	98%	100%					
	$K = 5$	$n = 180$	FPR	0.01%	0	0.00%	0.01%	0.02%			
			TPR	35%	46%	44%	39%	75%			
		$n = 300$	FPR	0.02%	0.02%	0.02%	0.06%	0.10%			
			TPR	66%	73%	67%	43%	43%			
		$n = 600$	FPR	0.02%	0.02%	0.01%	0.10%	0.12%			
			TPR	81%	77%	92%	52%	56%			
	$K = 7$	$n = 180$	FPR	0	0	0.01%	0	0.00%	0	0.00%	
			TPR	13%	28%	32%	20%	21%	15%	69%	
		$n = 300$	FPR	0.01%	0.01%	0.01%	0.00%	0.01%	0.01%	0.01%	0.02%
			TPR	20%	48%	60%	52%	40%	23%	78%	
		$n = 600$	FPR	0.02%	0.02%	0.01%	0.02%	0.01%	0.01%	0.01%	0.07%
			TPR	37%	79%	90%	52%	71%	66%	98%	
Age	$K = 3$	$n = 180$	FPR	0.01%	0.01%	0.06%					
			TPR	68%	76%	96%					
		$n = 300$	FPR	0.05%	0.03%	0.08%					
			TPR	95%	95%	90%					
		$n = 600$	FPR	0.06%	0.06%	0.08%					
			TPR	94%	99%	95%					
	$K = 5$	$n = 180$	FPR	0.05%	0.05%	0.01%	0.04%	0.06%			
			TPR	67%	61%	82%	69%	97%			
		$n = 300$	FPR	0.09%	0.03%	0.04%	0.04%	0.08%			
			TPR	78%	85%	97%	85%	91%			
		$n = 600$	FPR	0.07%	0.06%	0.07%	0.08%	0.08%			
			TPR	88%	84%	94%	83%	94%			
	$K = 7$	$n = 180$	FPR	0.02%	0.01%	0.01%	0	0.02%	0.01%	0	
			TPR	39%	62%	58%	68%	38%	54%	85%	
		$n = 300$	FPR	0.08%	0.01%	0.04%	0.01%	0.03%	0.03%	0.02%	
			TPR	46%	62%	79%	84%	79%	73%	93%	
		$n = 600$	FPR	0.09%	0.08%	0.04%	0.10%	0.03%	0.05%	0.07%	
			TPR	52%	77%	85%	84%	77%	79%	84%	

1 Responses to Reviewer 1

In this manuscript, the author proposed a new method that can detect cell-type-specific association between DNA methylation and phenotype using a hierarchical modeling approach. In their model, they assumed that the effect of cellular composition on methylation level is multiplicative rather than additive. In addition to separating the association from a mixture of cell types, the author demonstrated that this method can also increase the power in of detecting aggregated effect of phenotype on methylation level compared with existing methods. However, there are several major problems that need further revision.

We thank the reviewer for the complete summary and the great suggestions.

1. The author should perform simulations under the null hypothesis and report the type I error of the proposed method compared with other method accounting for cell-type heterogeneity.

We appreciate the important point raised by the reviewer. In the updated simulation settings, we conducted simulations under the true null (see Section 0.1 for details) and reported the type I errors of HIRE along with those of the other five methods: SVA, the unadjusted analysis, RefFreeEWAS, EWASHer, and ReFACTor. The simulations under true null show that HIRE controls type I errors very well, none of which are greater than 0.05% in Table R1.

2. The authors should report QQ-plots of the p-values under both null and alternative hypotheses, which can better illustrate the model performance than a density plot of p-values dose.

We thank the reviewer for the suggestion. In the updated simulation studies, for each setting, we provided the Q-Q plots of the p-values from HIRE and from the other five approaches under comparison. Please see Figures R3 and R4 for an example and Section 0.2 for details. The Q-Q plot under “true null” shows that the distribution of p-values from HIRE behaves like a uniform distribution.

3. The authors should report the bias of parameter estimates in their EM algorithm, which is the difference between the true and estimated value of parameters.

We thank the reviewer for the suggestion. In the HIRE model, we are mainly interested in the estimations of cellular proportions p_{ki} and the phenotype effects β_{jkl} . Therefore, in each simulation setting, we provided the average bias plot of $\frac{1}{K} \sum_{k=1}^K |\hat{p}_{ki} - p_{ki}|$ for each sample i and the average bias plot of $\frac{1}{K} \sum_{k=1}^K |\hat{\beta}_{jkl} - \beta_{jkl}|$ for each CpG site j and phenotype ℓ . Moreover, we plotted the estimated cellular compositions versus the underlying truth and calculated their correlations, all of which are greater than 0.9. See Figure R5 for an example and Section 0.2 for details. The figures display the

low biases for most estimates in both the “true null” and the “true alternative” cases.

4. The authors did simulation in only one situation, which is not comprehensive. I would suggest the author do more simulations by varying the effect size (both positive and negative and positive effect), number of cell types and sample size

We appreciate the point made by the reviewer. In the updated simulation studies, we considered both positive and negative risk effects β_{jkl} , varied the cell type number K ($K = 3, 5, 7$), and investigated different sample sizes n ($n = 180, 300, 600$), leading to a total of 18 simulation settings. Please see Section 0.1 for details. For all of the 18 simulation settings, HIRE outperforms the other competitors in terms of FPR and TPR at the aggregated level (see Table R1).

5. In simulation, the authors truncated the observed methylation values to be within zero to one to imitate the real data. However, the model is based on normal assumption of error terms. Hence the author should give more discussion of their assumption and how the difference between assumption and data would influence the model performance.

We thank the reviewer for the insightful comment. We acknowledge that the normal assumption in the HIRE model does not ensure that the observed methylation beta-values are always between zero and one. Nevertheless, in HIRE, we do constrain the baseline methylation profiles μ_{jk} s to the closed interval $[0, 1]$ and force the cellular compositions p_{ki} s to be non-negative and sum up to one: $\sum_{k=1}^K p_{ki} = 1$. As a result, as the phenotypes have no effect on most CpG sites, most observations, O_{ji} s, have the mean $\sum_{k=1}^K \mu_{jk} p_{ki}$ in $[0, 1]$. To verify that, in the real application, we used HIRE estimates to calculate the fitted methylation values \hat{O}_{ji} s. It turns out that more than 99.99% of the fitted methylation values are between zero and one for both the RA dataset and the GALA II dataset. Moreover, previous approaches to EWAS also often assume a normal distribution for the error terms and show good performances in real data sets (Houseman et al., 2012; Rahmani et al., 2016). Therefore, the normal assumption fits the data reasonably well and does not have a large effect on the performance of HIRE. In our future work, we will investigate the possibility of fitting the data with other distributions.

2 Responses to Reviewer 2

In the manuscript entitled Detection of cell-type-specific risk-CpG sites in epigenome-wide associations studies? the authors describe a statistical method (HIRE) that allows for the identification of risk-associated CpGs within individual cell-types when only DNA methylation signatures profiled over a complex mixture of those cell types are available (e.g., whole-blood derived DNA methylation signatures). The manuscript addresses a key issue that is believed by many within the epigenomics research community to be amongst the foremost challenges currently faced by EWAS; namely, cell

heterogeneity and its impact on the statistical analysis of DNA methylation data. The proposed method represents an important advancement for the DNA methylation research community that has potential to open up new lines of inquiry when analyzing methylation data profiled in heterogeneous tissue types. The authors should be commended for the statistical rigor employed in the development of their methodology and for the very clear description of their approach.

We greatly appreciate the commendation from the reviewer.

While this reviewer is quite optimistic about this work and its potential implications for EWAS, there are several very important issues that need to be addressed in order for HIRE to have its intended impact and reach its full potential. In what follows, I summarize my major concerns and provide suggestions for additional analyses that

1. While the methodology itself appears sound and statistically rigorous, the implicit “reference-free structure assumed in HIRE severely limits any kind of biologically meaningful interpretation of the ensuing results. For example, the user does not know what the cell types actually are, rather they are merely arbitrarily labeled cell-type 1, cell-type 2, cell-type K (where K is the assumed or empirically determined number of cell types). From a purely biological standpoint, if the user does not know what the cell types are, then what use is it to know the specific CpG sites within those cell types that are differentially methylated based on some phenotype or exposure? To put it a different way, imagine that a specific CpG is identified by HIRE as being significantly associated with some phenotype or exposure, X only in cell-type 2. If the identity of cell-type 2 is unspecified/unknown, then what kind of meaningful conclusions can be reached by the investigator in terms of the underlying biology that governs the process being studied? I understand and appreciate the authors’ argument that the reference methylation signatures for the cells underlying certain tissue types have yet to be determined and therefore cannot be used, however they are available for some tissue types, most notably, blood. Because blood is one of the most commonly used tissue types used in EWAS, not leveraging leukocyte-specific reference methylation signatures seems like a major missed opportunity. What I am suggesting is that the authors consider a reference-based + reference-free hybrid approach where the application of HIRE to whole-blood DNA methylation data would utilize existing reference methylation signatures for isolated leukocyte cell types as a means toward allowing one to identify risk-CpGs within granulocytes, monocytes, NK cells, B cells, CD4T cells, and CD8T cells. In doing so, the user knows the actual identity of the cell type (rather than an arbitrary label), which would lead to more meaningful biological interpretations of the results generated by HIRE.

We thank the reviewer for pointing out an important biological concern. In our revised manuscript, we designed a protocol similar to the one proposed in a recent Nature Methods paper (Kiselev et al., 2018) to align the estimated baseline methylation profile $\boldsymbol{\mu}_k = (\mu_{1k}, \dots, \mu_{mk})$ to references with known biological cell types.

Assume that we have the reference methylation profiles for H annotated cell types. We denote the methylation profile for cell type h as $\phi_h = (\phi_{1h}, \dots, \phi_{mh})$. We aim to annotate μ_k using the references. First, we calculate the cosine similarity, the Pearson correlation, and the Spearman correlation between μ_k and ϕ_h for each cell type $h \in \{1, \dots, H\}$. The three similarity measures are between -1 and 1 , and a high positive value indicates a high similarity between two vectors. Second, for each similarity measure ℓ ($\ell = 1, 2, 3$), we identify the cell type h_ℓ that has the maximal similarity degree with μ_k . If at least two out of the three similarity measures identify the same reference cell type \tilde{h} and their corresponding similarity values are greater than 0.7 , then we annotate μ_k with the reference cell type \tilde{h} . Otherwise, μ_k is believed to belong to a “new” cell type that is not included in the references. We repeat the above process for each methylation profile μ_k estimated from HIRE.

For the two real applications in our manuscript, as the methylation data are obtained from whole blood, we prepared the references from a whole blood methylation study (Reinius et al., 2012) with GEO accession code GSE35069. The study collected eight isolated blood cell subpopulations—granulocytes, CD4+ T cells, CD8+ T cells, CD14+ monocytes, CD19+ B cells, CD56+ NK cells, neutrophils, and eosinophils—for six individuals. Accordingly, we define the reference profile ϕ_h for cell type h as the average methylation profile of these individuals, i.e., $\phi_h := \frac{1}{6} \sum_{i=1}^6 \phi_{hi}$. In the RA dataset, HIRE identified six cell types, and based on the above alignment protocol, cell type 1 is annotated as CD4+ T; cell type 2 belongs to granulocytes; cell type 4 is identified as neutrophils; and the remaining three cell types cannot be aligned to the references. In the GALA II dataset, HIRE also detected six cell types: cell type 1’s annotation is CD4+ T; cell types 2 and 3 belong to neutrophils; cell type 4 is from granulocytes; and cell types 5 and 6 cannot be annotated using the references. We also plotted the estimated cell-type-specific methylation levels versus the values observed for their matched biological cell types in Figures R6 and R7. Therefore, in real applications, although the references are not perfect given all of the batch effects and potential biological differences, aligning the estimated baseline methylation profiles to the reference methylation profiles does allow for more biological interpretations of the results.

Figure R6: The scatter plots for the estimated methylation profile by HIRE and its matched reference in the RA data set for (a) cell type 1; (b) cell type 2; (c) cell type 4. The same figure is shown as the Supplementary Figure 64.

Figure R7: The scatter plots for the estimated methylation profile by HIRE and its matched reference in the GALA II data set for (a) cell type 1; (b) cell type 2; (c) cell type 3; and (d) cell type 4. The same figure is shown as the Supplementary Figure 66.

2. The simulation study is quite narrow in scope (only one simulation scenario was considered), not to mention the assumptions being made in the generation of the methylation data are somewhat unrealistic given what we know about DNA methylation profile for the cell types underlying certain tissues. Take blood for example, the lineage relationships between leukocyte subtypes is known as is the fact that cell types from a shared lineage have a methylation signature that is more similar as compared to cell types with differing lineages (e.g., lymphocytic versus myeloid derived cell types). However, in the simulation study the methylation signature is being simulated independently across cell types (page 14). Perhaps the dependency in methylation across cell types has no or little impact on the ability of HIRE to identify cell-type specific risk-CpGs, but the absence of such a simulation scenario makes this difficult, if not impossible, to ascertain. It would also be useful to see a simulation scenario

where the risk-CpGs are the same across cell types, but vary across cell types according to their magnitude/direction of association. For example CpG1 may be associated with some phenotype within each cell type, however the magnitude and/or direction of its association varies across the cells. Will HIRE be able to correctly identify such CpGs or would the differing direction of association across impair their detection? One other aspect that could and should be addressed through simulation is the threshold for a given cells proportion for which HIRE is unable to detect risk-CpGs for that cell type. For example, if a given cell only represents a small fraction of the underlying cell composition (e.g., 2-5%), is HIRE still able to detect risk-CpGs within that cell type. Does that depend on how strong the association is for those specific CpGs (e.g., moderate to large delta-betas)? To make a more convincing argument for the HIRE methodology, it behooves the authors to scale up their simulation studies to assess HIRE across a range of possible scenarios

We thank the reviewer for the valuable comments regarding simulation. In the updated simulation studies, we have considered 18 simulation settings and accounted for the lineage relationships among cell subtypes. Specifically, in each simulation setting, we let a set of cell types have similar baseline methylation profiles μ_k . Without loss of generality, we assume that cell types $k = 1, \dots, K_{sim}$ are from the same cell lineage. Specifically, for cell type k' , $k' = 2, \dots, K_{sim}$, we randomly select 80% of m CpG sites whose methylation levels are similar to those in cell type 1—we let their $\mu_{jk'}$ be μ_{j1} plus a very small randomness. For the remaining CpG sites, their corresponding $\mu_{jk'}$ s are independently drawn from $beta(3, 6)$. Thus, we induced the similarity among cell types $k' = 1, \dots, K_{sim}$ from the same lineage. Please see Section 0.1 for details. Our updated simulation results show that even when there exist cell types from the same cell lineage, HIRE still performs very well and outperforms existing methods on association detection at the aggregated level (see Table R1).

In the updated simulation studies, we also considered the cases where phenotype effects have different directions and magnitudes. In the “true alternative” setting, we set nonzero phenotype effects as follows.

For phenotype 1, the case/control status, we let it affect the first 10 CpG sites in all of the cell types: $\beta_{jk1} \neq 0$ for $j = 1, \dots, 10$ and $k = 1, \dots, K$. Then, we assume that the next 10 CpG sites are influenced by the disease status in the first K_{sim} cell types that come from the same lineage but not the other cell types: $\beta_{jk1} \neq 0$, $k = 1, \dots, K_{sim}$ and $\beta_{jk1} = 0$, $k = K_{sim} + 1, \dots, K$, for any $j = 11, \dots, 20$. Furthermore, for cell type $k \in \{K_{sim} + 1, \dots, K\}$, we let the disease status affect CpG sites $j = 20 + 10(k - K_{sim} - 1) + 1, \dots, 20 + 10(k - K_{sim})$ only in cell type k . We generated the cell-type-specific effects of age in a similar fashion for CpG site loci 21 to $40 + 10(K - K_{sim})$. Figure R1(a) and (c) show an illustration of cell-type-specific risk-CpG sites. For each nonzero β_{jk1} , we let $\beta_{jk1} = r_{jk} \cdot \omega_{jk}$, where $\omega_{jk} \sim Unif(0.07, 0.15)$ and r_{jk} takes value at 1 and -1 with equal probabilities. Thus, β_{jk1} s can have both positive and negative effects. The randomness for ω_{jk} allows the magnitudes of the effect sizes to vary across different cell types. In the same spirit, we also considered

the directions and magnitudes when generating the phenotype effects for age.

In terms of the reviewer’s concerns over the cellular proportions, we considered multiple cell type numbers so that the average cellular proportions become small for a large cell type number. To be specific, when $K = 3$, we drew $\mathbf{p}_i = (p_{1i}, \dots, p_{Ki})$ from a Dirichlet distribution $Dir(4, 4, 2 + 0.1x_{i2})$ if sample i is a control and $\mathbf{p}_i \sim Dir(4, 4, 5 + 0.1x_{i2})$ if it is a case; when $K = 5$, we let $\mathbf{p}_i \sim Dir(3, 3, 3, 3, 2 + 0.1x_{i2})$ for a control sample and $\mathbf{p}_i \sim Dir(3, 3, 3, 3, 5 + 0.1x_{i2})$ for a case sample; and when $K = 7$, we sampled $\mathbf{p}_i \sim Dir(1, 3, 3, 3, 2, 2, 2 + 0.1x_{i2})$ for controls and $\mathbf{p}_i \sim Dir(1, 3, 3, 3, 2, 2, 5 + 0.1x_{i2})$ for cases. Consequently, the expected minimum cell proportions are 22.2%, 13.6%, and 4.2% for $K = 3, 5$, and 7, respectively. For $K = 7$ where the least abundant cell type takes up about 4.2% of the total population, HIRE still enjoys a high power with 600 samples, a typical sample size of EWAS nowadays, at the aggregated level (see Table R1) and also detects most cell-type-specific risk-CpG sites reasonably well (see Table R2). In principle, a rare cell type requires a larger sample size for detection. However, determining the exact detection boundary is a challenging theoretical statistical problem (Xie et al., 2011; Jin and Ke, 2016), and we leave it for future research. We have added a discussion on the same issue on pages 16 and 17 in the manuscript.

3. My final major comment is that the Discussion section makes no mention of the limitations of the proposed methodology. When does the method fail, are there scenarios for which HIRE is not particularly well suited, what are the limitations based on the datasets used to assess its performance, etc. In addition to discussing limitations inherent to the methodology itself, one very obvious limitation of this work that is only briefly mentioned, is the application of HIRE a gold-standard data set. While the definition of gold-standard may be open to debate, what is not debatable is the fact that none of the data sets used in the present examination provide conclusive proof that HIRE is indeed capable of identifying bonafide risk-CpGs for individual cell types in a real DNA methylation data set. It would be incredibly helpful if the authors could describe what they believe would be the ideal data set in which to illustrate/validate their methodology. For example, if purified cell types were isolated, subject to site-specific epigenetic editing on per cell type basis (e.g., so the investigator knows the ground truth about what CpGs are differentially methylated within specific subpopulation of cells), and then mixed in predetermined proportions (such as in an artificial reconstruction experiment) followed by DNA methylation assessed over the mixed population of cells, it would seem that such an experiment would be pretty close to a gold-standard data set. The authors should comment on the feasibility of such an experiment and why this was not conducted in the present examination.

We appreciate the important comments made by the reviewer. In the revised manuscript, we added discussions on the limitations of HIRE. One major issue for all of the cell-type deconvolution methods is that deconvolution cannot be achieved if there is no variation of cellular composition among samples. For example, assuming that the samples are mixtures of two cell types and $\mathbf{p}_i = \mathbf{p}$ for all of the samples, the observed methylation profile \mathbf{O}_i equals $\mathbf{u}_{i1}p_1 + \mathbf{u}_{i2}p_2 = (\mathbf{u}_{i1} + p_2C)p_1 + (\mathbf{u}_{i2} - p_1C)p_2$

$:= \tilde{\mathbf{u}}_{i1}p_1 + \tilde{\mathbf{u}}_{i2}p_2$ for any constant C . As a result, \mathbf{u}_{i1} and \mathbf{u}_{i2} are not estimable. As HIRE is also a deconvolution based method, it suffers from the same challenge. Meanwhile, as HIRE needs to learn in total $(1 + 2K + pK)m + (K - 1)n$ parameters with mn observed values, a moderate sample size is needed. Our simulation studies show that with 180 samples, HIRE performs very well and outperforms all of the other methods. When the sample sizes are below 150, say 120, HIRE can still control FPR well but begins to lose power. Given the decreasing cost of EWAS, we recommend that researchers collect at least 200 samples to detect association at the aggregated level and 600 samples to identify cell-type-specific risk-CpG sites. A larger sample sizes can further boost the power. See Table R3.

Ideally, we would like to have the epigenetic spike-in experiments as described by the reviewer. However, biotechnologies for epigenetic editing, such as CRISPR-Cas, are still not mature at this stage and can create many off-targets modifications (Zhang et al., 2015). Therefore, most computational EWAS papers refer to numerical simulation studies rather than to experimental studies when evaluating the performance of their algorithms (McGregor et al., 2016; Rahmani et al., 2016). We designed our updated simulation studies to serve as the computational counterpart of the experimental spike-in studies. Accordingly, we added discussions on page 16 in the manuscript.

Table R3: The performances of HIRE with different cell type number K and various sample size n . In each setting, the average of TPR/FPR is shown based on five replicates. The same table is shown as the Supplementary Table 5.

HIRE	cell type number		n=90	n=120	n=150	n=180	n=300	n=600
null	K=3	FPR	0.08%	0	0.00%	0	0.00%	0
	K=5	FPR	0.01%	0.01%	0.03%	0.05%	0.01%	0
	K=7	FPR	0.00%	0	0.1%	0.00%	0.00%	0
alternative	K=3	FPR	0.01%	0.03%	0.01%	0.01%	0.00%	0.03%
		TPR	82.67%	93.33%	94%	98.67%	96.67%	100%
	K=5	FPR	0	0.01%	0.00%	0.00%	0.01%	0.01%
		TPR	36%	52%	55.6%	66%	89.6%	98.4%
	K=7	FPR	0	0.00%	0.00%	0	0.00%	0.04%
		TPR	12.67%	23.33%	36.33%	43%	63.33%	82.67%

3 Responses to Reviewer 3

The paper proposes a statistical method, HIRE, for cell type deconvoluting from methylation data and for detecting cell type specific methylation associations with a risk factor. The proposed method relies on a joint factor analysis model and an EM algorithm to first extract cell type proportions, and then for each cell type in turn, relies on a t/F test to detect differentially methylated sites in the specific cell type. With simulations

and two real data applications, the authors show that their method can be powerful. Overall, while the problem studied in the present paper can of potential interest, I am not convinced that the method proposed in the paper represents a significant advance over the previous approaches. In addition, it is unclear whether the methods HIRE compared to are properly applied in the present study and whether HIRE works better than other approaches in the two real data sets. My main comments are listed below:

We thank the reviewer for the comments. In the revised manuscript, we further clarified the difference between HIRE and existing methods and elaborated why the latter cannot provide cell-type-specific risk-CpG sites.

1. While the authors claim that their method is the first method to detect risk-CpG sites for individual cell types, I believe this so-claimed main novelty is an exaggeration. To the best of my knowledge, the problem of detecting cell type specific risk CpG sites has been relatively well studied by several previous studies. Indeed, the method proposed in the current paper, HIRE, is very similar to the factor model proposed in Houseman et al Bioinformatics 2014 and is also related to the nonnegative factorization model proposed in Houseman et al., BMC Bioinformatics 2016. Either of these two Houseman et al. methods, or some other relevant methods (e.g. reviewed in <https://academic.oup.com/hmg/article/26/R2/R216/3979364>), can be used to detect cell type specific risk-CpG sites by using a two-step procedure. The two-step procedure, just like the one used in the present paper, would extract methylation profile in a cell type specific fashion (i.e. the factor loading matrices in these methods), which can be further used to detect differential methylation in a cell type specific fashion by standard differential methylation methods, such as the t-test or F-test used in the present study. From this aspect, the only novel part of the current paper is that the proposed HIRE method can incorporate differential methylation information into cell deconvolution, but I feel this is a rather incremental improvement and I am unsure at this stage how much practice improvement this can bring. Indeed, if you look at the power comparison shown in Figure 2A, even using simulations that completely favor HIRE, HIRE still has an almost identical power as one of the Houseman et al methods. In addition, HIRE sacrifices the non-negative factorization advantage presented in Houseman et al., BMC Bioinformatics 2016. As a consequence, the cell type proportion estimates from HIRE, unlike those from Houseman et al., BMC Bioinformatics 2016 method, are not guaranteed to be nonnegative, making interpretation difficult and the subsequently statistical testing likely sub-optimal.

We are sorry that we did not sufficiently clarify the differences between HIRE and previous methods, which might have contributed to the reviewer’s confusion. While at first glance HIRE seems to be similar to the factor model in Houseman et al. (2014) or the method in Houseman et al. (2016), there are actually fundamental differences between HIRE and those two approaches.

Let us first review the RefFreeEWAS model proposed in Houseman et al. (2014) (page 1432). Using notations in HIRE, RefFreeEWAS assumes $\mathbf{O}_i = \mathbf{T}\mathbf{x}_i + \mathbf{M}\mathbf{p}_i$, where \mathbf{O}_i

is the observed methylation values of sample i , \mathbf{x}_i is sample i 's covariate vector, \mathbf{T} describes how the covariates affect the aggregated methylation values \mathbf{O}_i , \mathbf{p}_i is sample i 's cellular proportions, and \mathbf{M} is the cell-type-specific methylation profiles. Note that the cell-type-specific methylation profiles \mathbf{M} remain the same for all of the samples. In the RefFreeCellMix model proposed in Houseman et al. (2016), the authors do not consider covariates and directly assume that $\mathbf{O}_i = \mathbf{M}\mathbf{p}_i$ for each sample $i = 1, \dots, n$. Once again, please notice that RefFreeCellMix assumes that the cell-type-specific mean methylation values \mathbf{M} are the same for all of the samples.

As pointed out by Jaffe and Irizarry (2014), forcing the cell-type-specific methylation profiles to be the same— $\mathbf{u}_i = \mathbf{M}$, $i = 1, \dots, n$ —for all individuals is “not a reasonable, nor useful, assumption,” where \mathbf{u}_i is the cell-type-specific methylation profiles of sample i . The main issue is that regressing the same cell-type-specific methylation profiles \mathbf{M} for each individual on each sample's covariate \mathbf{x}_i does not detect any cell-type-specific risk-CpG sites because in a regression model, if all of the responses are the same, then there is no association with the covariates except the intercept term. Therefore, the two-step approach to detecting cell-type-specific risk CpGs by using RefFreeEWAS or RefFreeCellMix is ill-defined.

In contrast, HIRE models $\mathbf{O}_i = \mathbf{u}_i\mathbf{p}_i$, where \mathbf{u}_i varies across different samples, so the cell-type-specific methylation profiles \mathbf{u}_i can further depend on covariates through $\mathbf{u}_i = \boldsymbol{\mu} + \sum_{\ell=1}^q \mathbf{B}_\ell x_{i\ell}$, where $\boldsymbol{\mu}$ is the baseline methylation profiles and \mathbf{B}_ℓ is the phenotype ℓ effect matrix. This is a key difference between HIRE and Houseman's methods. Consequently, we can know how covariate ℓ influences the methylation through the phenotype effect matrix \mathbf{B}_ℓ , thus detecting cell-type-specific CpG sites affected by phenotype ℓ . Therefore, HIRE is not a simple extension of RefFreeEWAS or RefFreeCellMix but is completely novel in enabling cell-type-specific association detection.

Furthermore, by integrating $\mathbf{O}_i = \mathbf{u}_i\mathbf{p}_i$ with $\mathbf{u}_i = \boldsymbol{\mu} + \sum_{\ell=1}^q \mathbf{B}_\ell x_{i\ell}$ in HIRE, we have $\mathbf{O}_i = (\boldsymbol{\mu} + \sum_{\ell=1}^q \mathbf{B}_\ell x_{i\ell})\mathbf{p}_i = \sum_{\ell=1}^q \mathbf{B}_\ell x_{i\ell}\mathbf{p}_i + \boldsymbol{\mu}\mathbf{p}_i$, thus $\mathbf{O}_i = (\mathbf{B}_1\mathbf{p}_i, \dots, \mathbf{B}_q\mathbf{p}_i)\mathbf{x}_i + \boldsymbol{\mu}\mathbf{p}_i$. Compared to $\mathbf{O}_i = \mathbf{T}\mathbf{x}_i + \mathbf{M}\mathbf{p}_i$ assumed by RefFreeEWAS, we can see that HIRE models the cellular composition \mathbf{p}_i as multiplicative effects on the phenotype effects, instead of plain additive terms affecting the mean methylation levels with no interaction with those phenotype effects. Consequently, HIRE improves power and allows for a cell-type-specific association detection that is not available with existing methods.

In the following, we give a toy example to illustrate the importance of treating cellular compositions as multiplicative effects. Let us consider one CpG site and assume that there exist two cell types. Table R4 shows the cellular compositions $\mathbf{p}_i = (p_{1i}, p_{2i})$, the cell-type methylation profiles $\mathbf{u}_i = (u_{i1}, u_{i2})$, and the observed methylation values for two controls and three cases. The proportion of cell type 1 is low in the controls and high in the cases. Moreover, the given CpG site is hypermethylated in cell type 1 in cases but not differentially methylated in cell type 2. Here, O_i is calculated as $u_{i1}p_{1i} + u_{i2}p_{2i}$. Denoting the case/control status of sample i by x_i , then $x_i = 1$ if it is a case and $x_i = 0$ if it is a control.

Table R4: The data structure in a toy example, where O_i is the observed methylation value of sample i , p_{1i} is sample i 's proportion of cell type 1, u_{i1} is the methylation profile of cell type 1 for sample i , and so forth. $O_i = u_{i1}p_{1i} + u_{i2}p_{2i}$. The same table is shown as the Supplementary Note Table S1.

controls					cases				
O_i	p_{1i}	u_{i1}	p_{2i}	u_{i2}	O_i	p_{1i}	u_{i1}	p_{2i}	u_{i2}
0.9	0.1	0	0.9	1	1	0.9	1	0.1	1
0.8	0.2	0	0.8	1	1	0.8	1	0.2	1
					1	0.7	1	0.3	1

Despite that existing deconvolution methods actually estimate \mathbf{p}_i s with biases, let us assume that the true cell type proportion \mathbf{p}_i is known for now. If we treat \mathbf{p}_i as additive effects as in RefFreeEWAS and other existing methods, we need to regress O_i on x_i and p_{1i} for $i = 1, \dots, 5$ with the intercept term:

$$O_i = \alpha + \tau \cdot x_i + M \cdot p_{1i} + \epsilon_i. \quad (3.1)$$

As we are interested in whether there is any association between this CpG site and the case/control status, we want to infer whether τ , the regression coefficient of x_i , is zero or not. However, the p-value of τ is 0.275, so we do not have enough evidence to reject the null hypothesis $H_0 : \tau = 0$ and to say that the case/control status influences this CpG site. In other words, regarding \mathbf{p}_i as the additive effect fails to detect this risk-CpG site in this case.

The main issue is as follows. Although the methylation difference between cases and controls is as large as 1 in cell type 1, after mixing different cell types, the maximum methylation difference between cases and controls at the aggregated level attenuates to 0.2. The additive effect assumption fails to account for such an effect.

In comparison, if we incorporate \mathbf{p} into the regression as multiplicative effects as in HIRE, we regress O_i on $x_i p_{1i}$, $x_i p_{2i}$ and p_{2i} for $i = 1, \dots, 5$ with the intercept term:

$$O_i = \tilde{\alpha} + \beta_1 \cdot x_i p_{1i} + \beta_2 \cdot x_i p_{2i} + \mu p_{2i} + \epsilon_i. \quad (3.2)$$

β_1 and β_2 are the effects of the disease status on cell type 1 and cell type 2, respectively, and we want to test if they are zeros or not. It turns out that the p-value for β_1 is 3.22×10^{-16} and the p-value for β_2 is 0.18. Thus, we are able to identify that the CpG site is at risk in cell type 1 but not in cell type 2, recovering the underlying truth. This successful identification is due to the multiplicative effect assumption acknowledging that the methylation difference from cell type 1 is modified by a factor of p_{1i} at the aggregated level. Consequently, HIRE rescues the risk-CpG site missed by the additive effect model.

Therefore, this example demonstrates that incorporating cellular compositions into the regression as multiplicative effects can not only detect the risk-CpG site, which might be missed by the additive effect regression, but can also enable cell-type-specific association detection.

In the updated simulation studies, we compared different methods for both the “true null” and the “true alternative” cases (see Section 0.1 for details). For the “true null” cases, we found that RefFreeEWAS cannot control the false positive rate (FPR) well, whereas HIRE has very small FPR. For the “true alternative” settings, HIRE also outperforms RefFreeEWAS in all of the cases in terms of FPR and the power for association detection at the aggregated level. Please see Table R1.

Moreover, in Houseman et al. (2016), RefFreeCellMix only requires the cellular compositions \mathbf{p}_i to satisfy $p_{ki} \geq 0$ and $\sum_{k=1}^K p_{ki} \leq 1$. In HIRE, we actually not only constrain p_{ki} s to be non-negative but also require $\sum_{k=1}^K p_{ki}$ to be strictly equal to one, as shown in the Supplementary Note.

We added discussions on the differences between HIRE and previous approaches to section “Method overview” on pages 4 and 5 in our revised manuscript and the Supplementary Note.

2. It is unclear why the authors choose to compare their methods to several unrelated methods that include SVA, EWASHer, ReFACTor. These three methods are designed to control for hidden confounding effects that are correlated with cell compositions. In contrast, HIRE and RefFreeEWAS are designed for reference-free cell deconvolution and the subsequent differential methylation analysis. Because these two sets of methods aim to deal with very different problems, I would suggest the authors to remove the comparison to SVA, EWASHer and ReFACTor (or add additional simulations and real data results to show how HIRE and RefFreeEWAS perform when hidden confounding factors are correlated with cell type compositions).

We thank the reviewer for the comment. We agree with the reviewer that SVA, EWASHer, and ReFACTor do not conduct cell-type deconvolution. Nevertheless, as EWASHer and ReFACTor do in fact aim to account for cellular heterogeneity when conducting association detection for EWAS (Zou et al., 2014; Rahmani et al., 2016), we are afraid that it might be unfair if we completely ignore them when comparing the abilities of different approaches to detect associations at the aggregated level. SVA was originally designed for “removing batch effects and unwanted variation in high-throughput experiments” (Leek et al., 2012), but recently it has also been frequently applied to EWAS as a benchmark approach (McGregor et al., 2016). Given the popularity of SVA, EWASHer, and ReFACTor in EWAS methods comparison papers (McGregor et al., 2016; Zheng et al., 2017; Rahmani et al., 2017), we feel that it may not hurt to include them for comparison to satisfy readers’ potential curiosity.

3. I don’t quite follow the argument on line 38 on the importance of “multiplicative”

vs “additive” effects. If I understand it correctly, running a two-step procedure using Houseman et al. methods and testing the association between cell type specific methylation levels with the phenotype would automatically assume a multiplicative effect as in the HIRE model.

We thank the reviewer for the comment. We should have elaborated the point further.

As we discussed in detail in our response to reviewer 3’s first question, the two-step approach briefly sketched by the reviewer is not feasible. As both RefFreeEWAS and RefFreeCellMix assume that the cell-type-specific methylation levels for each sample are the same: $\mathbf{u}_i = M$, a two-step approach by regressing the \mathbf{u}_i s obtained by RefFreeEWAS or RefFreeCellMix—all with the same value—on phenotypes cannot offer any cell-type-specific association detection.

For comparison, the matrix representations of the models for RefFreeEWAS, RefFreeCellMix and HIRE are as follows:

$$\text{RefFreeEWAS} \quad \mathbf{O}_i = \mathbf{T}\mathbf{x}_i + \mathbf{M}\mathbf{p}_i; \quad (3.3)$$

$$\text{RefFreeCellMix} \quad \mathbf{O}_i = \mathbf{M}\mathbf{p}_i; \quad (3.4)$$

$$\text{HIRE} \quad \mathbf{O}_i = (\mathbf{B}_1\mathbf{p}_i, \dots, \mathbf{B}_q\mathbf{p}_i)\mathbf{x}_i + \mu\mathbf{p}_i. \quad (3.5)$$

Comparing the red parts in Equation 3.3 and 3.5, we can see that RefFreeEWAS assumes additive effects for cellular composition \mathbf{p}_i and phenotypes \mathbf{x}_i , whereas in HIRE cellular composition \mathbf{p} has multiplicative effects on phenotype effects.

We have added more discussions regarding the “multiplicative” vs ”additive” effects in section “Method overview” on pages 4 and 5 in the manuscript and Supplementary Note.

4. The current paper lacks details on how they fitted the RefFreeEWAS package. In particular, RefFreeEWAS package contains two methods that include the factor model proposed in Houseman et al Bioinformatics 2014 and the nonnegative factorization model proposed in the new Houseman et al BMC Bioinformatics 2016 method. Which of the two methods did you use? After cell deconvolution, once you extracted cell type specific methylation profile from the factor loading matrices, did you perform an additional normalization, or did you use the factor loading matrix values directly? Or does it matter? In the differential methylation test, did you examine one cell type at a time by treating cell type specific methylation profile as the outcome variable and phenotype as the covariate, or did you use the reverse order by treating the methylation levels for all cell types as covariates and phenotypes as the outcome? Is the differential methylation test conditional on other covariates (i.e. age) or not? Did you use t-test, F-test, or other methods to detect differential methylation? Since HIRE and RefFreeEWAS are very similar to each other, why would the p-value distributions differ that much between the two methods (Figure 2d vs 2g and Figure 3c vs 3f)? The

huge p-value distribution difference worries me and makes me wonder whether the RefFreeEWAS package was used properly.

We thank the reviewer for the comment. We are sorry that we did not list the details on how we used RefFreeEWAS for analysis in our original manuscript. In our study, following the ReFACTor paper (Rahmani et al., 2016) and Houseman et al.'s 2016 paper (Houseman et al., 2016), we always refer to RefFreeEWAS as the factor model in Houseman et al. (2014). Therefore, there is no cell deconvolution by RefFreeEWAS.

Moreover, in Houseman's R package "RefFreeEWAS" <https://cran.r-project.org/web/packages/RefFreeEWAS>, the R function that implements the factor model was named as "RefFreeEwasModel," but the deconvolution model in Houseman et al. (2016) was named as "RefFreeCellMix." Therefore, although both of Houseman et al.'s models are in the same R package, RefFreeEWAS always represents the factor model.

In our analyses, we strictly follow the RefFreeEWAS user manual when applying the RefFreeEWAS method. In the following, we attach the R code used to implement RefFreeEWAS in our simulation study.

```
library(RefFreeEWAS)
setwd("D:/RefFreeEWAS/")

#cell type number
K <- 3

#read the observed methylation matrix
O_mat <- read.table("null_K3_n600_methy.txt", header = FALSE)
O_mat <- as.matrix(O_mat)

#read the two types of covariates
#The first row represents the case/control status
#The second row corresponds to age
X <- read.table("null_K3_n600_covariates.txt", header = FALSE)
X <- as.matrix(X)

#RefFreeEWAS model
test <- RefFreeEwasModel(O_mat, cbind(1, t(X)), K)
#where the dimension of the latent variables is set to
#the cell type number K

#Bootstrap for RefFreeEWAS Model
testBoot <- BootRefFreeEwasModel(test, nboot=10)

#The mean of Beta based on 10 bootstrap samples
```

```

#for the intercept term, case/control, age
BetaMean <- sapply(1:3, function(i){
rowMeans(testBoot[ ,i,1,])
})

#The standard deviation of Beta based on 10 bootstrap samples
#for the intercept term, case/control, and age
BetaStd <- sapply(1:3, function(i){
apply(testBoot[ ,i,1,], 1, sd)
})

#calculate p-values using normal approximation
pvalues_matr <- 2*pnorm(abs(BetaMean / BetaStd), lower.tail=F)

#the 2nd column of pvalues_matr corresponds to the effect of case/control status
pvalues <- as.numeric(pvalues_matr[ ,2])

```

It is noteworthy that (1) the R function RefFreeEwasModel provides no cell deconvolution; (2) in the RefFreeEWAS factor model, cellular compositions serve as additive terms; (3) the tests are based on nonparametric bootstrap.

In comparison, HIRE conducts cell-type deconvolution, models the multiplicative effects of the cellular compositions on phenotypes, and conducts hypothesis testing using t-tests. Therefore, given the substantial differences between HIRE and RefFreeEWAS, it is not surprising that their p-value distributions differ.

We included our code for applying RefFreeEWAS in the Supplementary Material and added discussion on the differences between HIRE and RefFreeEWAS to section “Method overview” on pages 4 and 5 in the manuscript.

5. It would be important to compare to a simple improvement of RefFreeEWAS, where you run RefFreeEWAS first to identify differentially methylated sites. Afterwards, you can exclude these differentially methylated sites (or if too many, exclude the top differentially methylated sites) and focus on the remaining null sites to run RefFreeEWAS again to estimate cell type compositions and cell type specific methylation profiles. Finally, you can perform another round of differential methylation analysis on all sites to detect differential methylated sites. I was wondering if this simple approach would work well compared to the more complicated joint EM algorithm used in HIRE.

We thank the reviewer for the suggestion. To begin with, RefFreeEWAS cannot estimate cell-type compositions because it is not a deconvolution model, which has been discussed in our response to reviewer 3’s fourth question. Moreover, in our reply to reviewer 3’s first question, we have shown that even if the cellular compositions can be perfectly recovered, adopting an additive effect model can lead to low detection power due to missing the multiplicative effects of the cellular compositions, let alone

that RefFreeCellMix cannot estimate the cellular composition perfectly when the phenotypes affect not only the cellular compositions but also the cell-type-specific methylation levels.

In the updated simulation studies, we directly compared HIRE and RefFreeEWAS for the “true null” cases to test their ability to control the false positive rate. It turns out that in all of the settings (with various sample sizes and cell type numbers) HIRE controls the false positive very well, all of which are $\leq 0.05\%$ (see Table R1). However, RefFreeEWAS can have a false positive as high as 0.17%, which will lead to 765 false positive CpG sites under the null condition for the Illumina 450k arrays. Therefore, HIRE outperforms RefFreeEWAS in controlling false positives.

6. Also, in the simulations, instead of ROC curve, could you show power at the usual Bonferroni threshold or a typical FDR? Is HIRE still more powerful than RefFreeEWAS at these commonly used thresholds? If you run permutation, would the p-values from either HIRE and RefFreeEWAS in the permuted well behaved? Also, what does the p-value distribution looks like under the simulated null, both in terms of histogram and qq-plot of $-\log_{10}p$ values?

We thank the reviewer for the great suggestion. In the updated simulation studies, we report the true positive rate (TPR/power) and the false positive rate (FPR/Type I error) for all of the methods at the aggregated level based on the Bonferroni threshold (see table R1 for details). For the “true alternative” cases, given FPR less than or equal to 0.05%, HIRE is more powerful than RefFreeEWAS. RefFreeEWAS cannot control false positives very well and its false positive rate is uniformly higher than HIRE in all of the simulation settings.

In HIRE, we do not use permutation to obtain the p-values. Our main concern is that although permutation tests are exact for simple models like one-way ANOVA and t-test (Lehmann and Romano, 2006, pp.176-177), their requirement of exchangeability under the null hypothesis is not fulfilled for settings with multiple covariates (Churchill and Doerge, 2008). As we do consider multiple phenotypes in HIRE, we choose not to use permutation tests.

In the updated simulation studies, we plotted both the histogram and the Q-Q plots of $-\log_{10}p$ -values. Figure R4 in the “true null” case shows that the distribution of p-values from HIRE behaves like a uniform distribution.

7. Is there a way to ensure that the cell composition proposition estimates to be non-negative in HIRE? What are the consequences of not keeping the proportion estimates non-negative? Also, are the estimates of cell composition from HIRE similar to that form the two different methods implemented in RefFreeEWAS, in both simulations and real data application?

We appreciate the reviewer’s comment. We are sorry that we did not provide the

constraints on cellular compositions \mathbf{p}_i for HIRE in the original main text. Actually, in HIRE, we do constrain \mathbf{p}_i to be (1) non-negative— $p_{ki} \geq 0$ for $k = 1, \dots, K$ and (2) summing up to one— $\sum_{k=1}^K p_{ki} = 1$. In contrast, RefFreeCellMix (Houseman et al., 2016) only requires $\sum_{k=1}^K p_{ki} \leq 1$. Therefore, the cell type proportion estimates p_{ki} , $k = 1, \dots, K$, from RefFreeCellMix are not guaranteed to sum up to one.

We compared the biases of cell proportion estimates from HIRE and RefFreeCellMix using 10 replicates. It turns out that HIRE provides more accurate estimates for the cellular compositions than RefFreeCellMix (Houseman et al., 2016) (see Figure R8).

Figure R8: The cellular composition bias box plots of RefFreeCellMix (Houseman et al., 2016) and HIRE using ten replicates where $K = 3$ and $n = 300$. For each replicate, we calculate the average bias of $\frac{1}{Kn} \sum_{i=1}^n \sum_{k=1}^K |\hat{p}_{ki} - p_{ki}|$ for the two approaches, respectively.

8. One can also apply RefFreeEWAS and BIC to select number of cell types. Are the cell type number from RefFreeEWAS consistent with HIRE in both simulations and real data sets?

We thank the reviewer’s comment. RefFreeEWAS (Houseman et al., 2014) relies on the single value decomposition (SVD), a numerical matrix decomposition method

without any distribution assumption for the data. However, the BIC, as a statistical model selection criterion, requires knowledge of the likelihood function of the collected data. Without any distribution assumption in RefFreeEWAS, we have no idea how to derive the likelihood function and use the BIC for model selection.

9. For real data applications, how do you normalize the data before analysis? Will different normalization results in different results? Or will different normalization favor different methods?

We appreciate the point raised by the reviewer. We are sorry that we did not provide enough detail about the normalization procedure in the real data analysis. In real application, we analyzed two whole blood methylation datasets. In the RA data set (Liu et al., 2013) with accession code GSE42861, the data we downloaded had already been normalized by Illumina’s control probe scaling procedure (see Liu et al. (2013) “Illumina 450K microarray data preprocessing” section for details). Regarding the GALA II data set with accession code GSE77716, the methylation data had been normalized by SWAN (Maksimovic et al., 2012) and corrected for batch effects using ComBat (Johnson et al., 2007). In our manuscript, we focused on the preprocessed data to conduct statistical analysis and did not conduct any further normalization. The discussion on how different normalization procedures favor different methods of detecting associations is beyond the scope of this paper.

10. How do you declare significance in the real data? Can you run permutations to construct a null p-value distribution in the real data? Which method works the best in the two real data sets? How many sites did RefFreeEWAS detect? What’s the overlap between the detected sites between RefFreeEWAS and HIRE?

We thank the reviewer for the comment. In real application, for all of the methods, we used Bonferroni correction to control the family-wise error rate (FWER) to be less than $\alpha = 0.01$. For HIRE, as it can provide p-values of CpG sites for all of the cell types and phenotypes, the p-value threshold to call “significant” is $\alpha/(mKq)$, where m is the number of CpG sites and q is the phenotype number. For the other five methods, the p-value threshold is α/m as they only provide the p-values at the aggregated level.

In real application, there are multiple phenotypes. Although permutation tests are exact for simple models like the one-way ANOVA and t-test (Lehmann and Romano, 2006, pp.176-177), their requirement of exchangeability under the null hypothesis is not fulfilled for settings with multiple covariates (Churchill and Doerge, 2008). Therefore, given the multiple phenotypes in real application, we choose not to use permutation tests.

At the aggregated level, in the GALA II dataset, there are 902 CpG sites detected by RefFreeEWAS, in which 37 CpG sites are shared by HIRE (HIRE discovered in total 186 CpG sites). In the RA dataset, RefFreeEWAS identified 373 CpG sites, and 61

CpG sites are also detected by HIRE (HIRE detected in total 81 CpG sites).

11. The second data set contains population stratification. How can you make sure that these population stratification is well controlled for? Population structure can influence cell composition and affect methylation level through paths other than cell composition. Therefore, if you don't control for population structure well, then the cell composition estimates are likely inaccurate. But if you over-correct for population structure, then you would also likely remove cell compositions together with population structure. How do you check all these in real data?

We appreciate the comment by the reviewer. Actually, one great advantage of HIRE is that it accounts for the effects of population structures on cellular compositions and cell-type-specific methylation levels simultaneously. In contrast, existing methods (Houseman et al., 2014; Zou et al., 2014; Rahmani et al., 2016) regress out population structures and model cell heterogeneity separately. The separate approach can lead to the exact problem that concerned the reviewer. However, the model structure of HIRE automatically models both effects simultaneously without over-correction.

In our simulation studies, we let both the cellular compositions and the cell-type-specific methylation levels depend on the phenotypes, and HIRE always performs well and recover the underlying truth without any under-correction or over-correction.

In the GALA II dataset, we have four ethnicity groups—“Mexican,” “Mixed Latino,” “Puerto Rican,” and “Other Latino.” Therefore, we created three dummy variables to represent the four ethnicity groups. In particular, we used $(0, 0, 0)$, $(1, 0, 0)$, $(0, 1, 0)$, and $(0, 0, 1)$ to represent “Mexican,” “Mixed Latino,” “Puerto Rican,” and “Other Latino,” respectively. We incorporated the ethnicity indicators as phenotype covariates x_{il} into HIRE:

$$E [O_{ji}] = \mu_{j1} + \sum_{k=2}^K (\mu_{jk} - \mu_{j1}) p_{ki} + \sum_{k=1}^K \sum_{\ell=1}^q \beta_{jk\ell} x_{i\ell} p_{ki}.$$

In the above equation, the population structures coded in $x_{i\ell}$ s actually affect the observed methylation level via the third term on the right-hand side in addition to the cell proportions at the second term. Therefore, HIRE takes into account the cellular heterogeneity potentially affected by population structures and models the effects of population structures on the observed methylation levels simultaneously. In particular, given its ability to detect cell-type-specific risk-CpGs, HIRE is able to identify whether two ethnic groups have differentially methylated CpGs for each cell type.

References

- Churchill, G. and R. Doerge (2008). Naive application of permutation testing leads to inflated type I error rates. *Genetics* 178(1), 609–610.
- Houseman, E. A., W. P. Accomando, D. C. Koestler, B. C. Christensen, C. J. Marsit, H. H. Nelson, J. K. Wiencke, and K. T. Kelsey (2012). DNA methylation arrays as surrogate measures of cell mixture distribution. *BMC Bioinformatics* 13(1), 86.
- Houseman, E. A., M. L. Kile, D. C. Christiani, T. A. Ince, K. T. Kelsey, and C. J. Marsit (2016). Reference-free deconvolution of DNA methylation data and mediation by cell composition effects. *BMC Bioinformatics* 17(1), 259.
- Houseman, E. A., J. Molitor, and C. J. Marsit (2014). Reference-free cell mixture adjustments in analysis of DNA methylation data. *Bioinformatics* 30(10), 1431–1439.
- Jaffe, A. E. and R. A. Irizarry (2014). Accounting for cellular heterogeneity is critical in epigenome-wide association studies. *Genome Biology* 15(2), R31.
- Jin, J. and Z. T. Ke (2016). Rare and weak effects in large-scale inference: methods and phase diagrams. *Statistica Sinica*, 1–34.
- Johnson, W. E., C. Li, and A. Rabinovic (2007). Adjusting batch effects in microarray expression data using empirical bayes methods. *Biostatistics* 8(1), 118–127.
- Kiselev, V. Y., A. Yiu, and M. Hemberg (2018). scmap: projection of single-cell RNA-seq data across data sets. *Nature Methods* 15(5), 359.
- Leek, J. T., W. E. Johnson, H. S. Parker, A. E. Jaffe, and J. D. Storey (2012). The sva package for removing batch effects and other unwanted variation in high-throughput experiments. *Bioinformatics* 28(6), 882–883.
- Lehmann, E. L. and J. P. Romano (2006). *Testing statistical hypotheses*. Springer Science & Business Media.
- Liu, Y., M. J. Aryee, L. Padyukov, M. D. Fallin, E. Hesselberg, A. Runarsson, L. Reinius, N. Acevedo, M. Taub, M. Ronninger, et al. (2013). Epigenome-wide association data implicate DNA methylation as an intermediary of genetic risk in rheumatoid arthritis. *Nature Biotechnology* 31(2), 142.
- Maksimovic, J., L. Gordon, and A. Oshlack (2012). Swan: Subset-quantile within array normalization for illumina infinium humanmethylation450 beadchips. *Genome Biology* 13(6), R44.
- McGregor, K., S. Bernatsky, I. Colmegna, M. Hudson, T. Pastinen, A. Labbe, and C. M. Greenwood (2016). An evaluation of methods correcting for cell-type heterogeneity in DNA methylation studies. *Genome Biology* 17(1), 84.

- Rahmani, E., N. Zaitlen, Y. Baran, C. Eng, D. Hu, J. Galanter, S. Oh, E. G. Burchard, E. Eskin, J. Zou, et al. (2016). Sparse pca corrects for cell type heterogeneity in epigenome-wide association studies. *Nature Methods* 13(5), 443.
- Rahmani, E., N. Zaitlen, Y. Baran, C. Eng, D. Hu, J. Galanter, S. Oh, E. G. Burchard, E. Eskin, J. Zou, et al. (2017). Correcting for cell-type heterogeneity in DNA methylation: a comprehensive evaluation. *Nature Methods* 14(3), 218–219.
- Reinius, L. E., N. Acevedo, M. Joerink, G. Pershagen, S.-E. Dahlén, D. Greco, C. Söderhäll, A. Scheynius, and J. Kere (2012). Differential dna methylation in purified human blood cells: implications for cell lineage and studies on disease susceptibility. *PloS One* 7(7), e41361.
- Storey, J. D. and R. Tibshirani (2003). Statistical significance for genomewide studies. *Proceedings of the National Academy of Sciences* 100(16), 9440–9445.
- Xie, J., T. T. Cai, and H. Li (2011). Sample size and power analysis for sparse signal recovery in genome-wide association studies. *Biometrika*, 273–290.
- Zhang, X.-H., L. Y. Tee, X.-G. Wang, Q.-S. Huang, and S.-H. Yang (2015). Off-target effects in CRISPR/Cas9-mediated genome engineering. *Molecular Therapy-Nucleic Acids* 4.
- Zheng, S. C., S. Beck, A. E. Jaffe, D. C. Koestler, K. D. Hansen, A. E. Houseman, R. A. Irizarry, and A. E. Teschendorff (2017). Correcting for cell-type heterogeneity in epigenome-wide association studies: revisiting previous analyses. *Nature Methods* 14(3), 216–217.
- Zou, J., C. Lippert, D. Heckerman, M. Aryee, and J. Listgarten (2014). Epigenome-wide association studies without the need for cell-type composition. *Nature Methods* 11(3), 309.

Reviewers' comments:

Reviewer #1 (Remarks to the Author):

The authors have sufficiently addressed the reviewers' comments and significantly improved the manuscript.

Reviewer #2 (Remarks to the Author):

Overall, I think the authors did a commendable job at being responsive to the reviewer comments and I think the manuscript is much improved as a result. In particular, the additional simulation scenarios, use of existing DNA methylation data on purified cell types for inferring the identity of the identified cell types, and the inclusion of a pointed discussion on the limitations and shortcomings of HIRE are all valuable additions. Yet, there is one issue that I think deserves a little more looking into.

In response to a previous comment that concerned the use of existing DNA methylation data on purified isolated leukocyte subtypes to infer the identity of the cell types identified by HIRE the authors revised their methodology and now utilize the Reinius et al., (2012) data set as the basis for such inference. Curiously though, the authors chose to use granulocytes, neutrophils, eosinophils, and basophils as part of the panel of cell types in which to infer cell identity, rather than just using granulocytes (consists of neutrophils, basophils, and eosinophils) or the collection of neutrophils, eosinophils, and basophils. The reason I bring this up is that the vast majority of granulocytes are neutrophils, yet both wind up being annotated to the cell types identified in both the RA and GALA II data sets, which seems odd. What are the average proportions of the cell types annotated to granulocytes and neutrophils within the RA and GALA II data and do they make sense given the expected proportions of these cell types in whole blood? Note, neutrophils typically comprise 50-70% of circulating cell types in whole blood? Along these same lines, it concerns me that the other cell types (e.g., monocytes, CD8T, NK, and B cells) did not align to any of the other cell types identified by HIRE since they typically account for anywhere between 3-10% of circulating immune cells, so you would think that they should be detected in these data sets? Is the criteria for declaring a match too stringent or is their simply weak similarity between these cell types and those identified by HIRE? On a related note, the authors might want to apply their methodology to a data set where the underlying cellular is "known" as a sanity check to see that the inferred cell types are being appropriately annotated and that their estimated proportion via HIRE are reasonable based on their "known" proportion. Two data sets immediately come to mind in this regard are Koestler et al., (2016) [GSE77797] and Salas et al., (2018)[GSE112618 & GSE110554]. Both papers describe reconstruction experiments (the first uses the 450K array and the second, the EPIC array) where the six major leukocyte cell types were mixed in predetermined proportions (e.g., "known" proportions).

Reviewer #3 (Remarks to the Author):

Most of my previous comments were not well addressed. In addition, a few newly added figures exposed further flaws with the proposed method:

(1) In my previous comment #1, I was concerned with the novelty of the proposed method. In particular, a paper by Houseman et al published in 2016 in the journal of BMC Bioinformatics is already capable of detecting cell-type-specific risk-CpG sites. I appreciate that the authors provide detailed equations to explain the difference between their HIRE approach and the Houseman 2016 method. However, these new explanations do not directly address my previous concern. Let's stick to authors' notations this time. The Houseman 2016 BMC Bioinformatics paper effectively performs a matrix factorization on the methylation data. In a probabilistic framework, the Houseman model can be effectively written as $O_i = M \cdot \pi_i + e_i$, with e_i being the residual errors. While M is certainly the same across all cells, the product term $M \cdot \pi_i$ provides one with a cell-subtype specific estimate: that is, the product term $M \cdot (\pi_i)_k$, by using the k 'th element of π_i , gives one an estimate of the k 'th

cell type specific methylation levels across all GpG sites for i'th individual. Therefore, for each cell type k and each CpG site j in turn, you can treat $M_j^*(\pi)_k$ across individuals as the outcome variable and perform a linear regression against any predictor variable x. This linear regression would allow you to detect any cell-type-specific risk-CpG sites. This is the two-step procedure I mentioned in the original comment #1. And this is such a simple procedure that you don't even need to rely on the RefFreeEWAS software to do so. Because the existing Houseman method already provides a straightforward way of detecting cell-type-specific risk-CpG sites, I do not think the current paper provides enough methodological advances or novelty over the exiting method. In addition, a detailed comparison with the Houseman 2016 BMC Bioinformatics method in all simulations and real data applications is warranted.

(2) In my previous comment #2, I was concerned that the authors compared their HIRE method to a bunch of unrelated methods including SVA, EAWSHer, ReFACTor. As I mentioned earlier, these methods are designed to control for hidden confounding factors that are correlated with cell compositions and the predictor variable of interest. Without controlling for these hidden confounding factors, one would incorrectly detect many false cell-type-specific risk-CpG sites that are not associated with the predictor variable or in a cell-type-specific fashion. On the other hand, the proposed HIRE method (and Houseman 2016 BMC Bioinformatics method) is more focused on detecting signals in the absence of confounding effects. Therefore, I initially suggested to remove these three methods (SVA, EAWSHer, ReFACTor) in comparison. However, it is also completely ok if the authors decide to keep them -- as they have done now -- as long as they also follow my second suggestion of performing extensive additional simulations in the presence of hidden confounding effects based on the EAWSHer and ReFACTor papers. For example, what happens if the hidden confounding factors are correlated with the predictor variable, or the cellular composition, or both, and if the hidden confounding factors affect many null CpG sites? However, the authors also failed to follow my second suggestions to perform these simulations. As of now, all simulations in the currently manuscript are completely in factor of their own method and fails to examine the important setting where there are hidden confounding factors. Please follow my previous comment #2 and add new simulations to examine these important settings.

(3) The authors now show that their proposed method fails to control for type I error. Specifically, the newly added supplementary Figures S1-9 show that the method produces highly inflated p-values under null even when the simulations are conducted favoring their own modeling assumptions. Perhaps with the only exception of Figure S9, their p-values under the null are much worse behaved than the other competing methods such as EWASHer or ReFACTOR. Failing to control for type I error is a huge issue for real data applications, as the p-values from the proposed method are no longer meaningful. The ability to control for type I error is such an important issue that the author should move one of these null supplementary Figures to a main figure. In addition, the authors have only explored a limited number of null settings. It is important to explore various null scenarios, with various parameter choices and various deviations from their own modeling assumptions, to examine whether theirs p-values are calibrated under the null. For example, will the p-value distribution become even worse if you increase the number of CpG sites to a realistic value, for example, 450K or 850K? What happens if you introduce a larger number of cell types, say k=10 or 20? And what happens if cell type compositions becomes more heterogeneous, with perhaps a few cell types dominate the composition? And will introducing additional hidden confounding effects that are correlated with cell compositions or predictor variable make things even worse? In any case, the authors need to make significant modifications of their method to provide calibrated p-values.

(4) For the previous comment #4, please refer to my elaborated comment #1 and address these previous concerns accordingly. Again, Houseman method is a very simple procedure that you don't even need to rely on the RefFreeEWAS software to fit a factor model and run a linear regression.

(5) For the previous comment #5, please refer to my elaborated comment #1 and address these previous concerns accordingly. Please also simulate data under the Houseman additive model and examine whether the proposed multiplicative procedure would work well when the truth is additive.

(6) For the previous comment #6, please also show p-value distribution under permutations in

both simulations and real data. Examining p-value distribution with permutations is perhaps the only way to examine type I error in real data sets.

(7) For the previous comment #8, you can use the probabilistic factor model that the Houseman method is effectively based on to compute a likelihood. The likelihood is straightforward to write down. Afterwards, you can add the penalty term to obtain the corresponding BIC for the Houseman method.

(8) For the previous comment #9, please do provide discussion on the effects of different normalization procedure -- the proposed model is based normality assumption, and the beta or M values for many CpG sites are not normally distributed.

(9) For the previous comment #10, because the proposed method does not provide calibrated type I error control (comment #3 above), it becomes critical to run permutations in order to compare different methods in real data in a fair fashion. Please compare all methods based on the same threshold established by permutations. Otherwise, it is impossible to judge the performance of the proposed HIRE method in real data.

(10) For the previous comment #11 on population structure concerns. If the authors believe that their method is free of population structure concern, please provide simulations to examine settings where population structure is correlated with the predictor variable of interest or cellular composition and affects a large number of null CpG sites. I appreciate that the authors performed simulations where both the cellular compositions and the cell-type-specific methylation levels depend on the phenotypes. However, the authors haven't checked the population stratification setting, where the cell-type-specific methylation levels depend on population structure, while population structure are not fully accounted for by the covariates. In the example of GALA II, what if the self-reported ethnicity is not accurate? Do you have genotypes to extract PCs to adjust for structure? In the equation provided on page 31, in the case of population stratification, these $\beta_{ajkl} \cdot x_{il} \cdot p_{ki}$ terms actually represent population stratification, which are something we want to remove, not to detect.

We would like to express our deepest gratitude to the Editor and all of the three reviewers for their precious time spent on reviewing our manuscript. We greatly appreciate the positive comments, such as “sufficiently addressed the reviewers’ comments,” “significantly improved the manuscript,” “a commendable job at being responsive to the reviewer comments,” and “much improved” by Reviewer 1 and Reviewer 2. We have taken Reviewer 2’s advice very carefully and thoroughly. We analyzed a semi-simulated dataset, which demonstrates that the cell types estimated by HIRE can be aligned well with the underlying truth and the cell proportions can be estimated well. We also thank reviewer 3’s comments, but we found that his or her suggested “two-step” approach did not work well based on our experimental results. We further provided evidence-based responses to Reviewer 3’s comments.

We revised our manuscript and Supplementary Materials accordingly. In the following, we provide point-by-point responses to the three reviewers. The comments from the reviewers were displayed in *blue and italics*. In the main manuscript, we highlight our modifications in *red*.

1 Reviewer 1

The authors have sufficiently addressed the reviewers’ comments and significantly improved the manuscript.

We appreciate the positive comment by the reviewer.

2 Reviewer 2

Overall, I think the authors did a commendable job at being responsive to the reviewer comments and I think the manuscript is much improved as a result. In particular, the additional simulation scenarios, use of existing DNA methylation data on purified cell types for inferring the identity of the identified cell types, and the inclusion of a pointed discussion on the limitations and short-comings of HIRE are all valuable additions. Yet, there is one issue that I think deserves a little more looking into.

We thank the reviewer for the positive and encouraging remarks.

In response to a previous comment that concerned the use of existing DNA methylation data on purified isolated leukocyte subtypes to infer the identity of the cell types identified by HIRE the authors revised their methodology and now utilize the Reinis et al., (2012) data set as the basis for such inference. Curiously though, the authors chose to use granulocytes, neutrophils, eosinophils, and basophils as part of the panel of cell types in which to infer cell identity, rather than just using granulocytes (consists

of neutrophils, basophils, and eosinophils) or the collection of neutrophils, eosinophils, and basophils. The reason I bring this up is that the vast majority of granulocytes are neutrophils, yet both wind up being annotated to the cell types identified in both the RA and GALA II data sets, which seems odd. What are the average proportions of the cell types annotated to granulocytes and neutrophils within the RA and GALA II data and do they make sense given the expected proportions of these cell types in whole blood? Note, neutrophils typically comprise 50-70% of circulating cell types in whole blood? Along these same lines, it concerns me that the other cell types (e.g., monocytes, CD8T, NK, and B cells) did not align to any of the other cell types identified by HIRE since they typically account for anywhere between 3-10% of circulating immune cells, so you would think that they should be detected in these data sets? Is the criteria for declaring a match too stringent or is their simply weak similarity between these cell types and those identified by HIRE? On a related note, the authors might want to apply their methodology to a data set where the underlying cellular is “known” as a sanity check to see that the inferred cell types are being appropriately annotated and that their estimated proportion via HIRE are reasonable based on their “known” proportion. Two data sets immediately come to mind in this regard are Koestler et al., (2016) [GSE77797] and Salas et al., (2018)[GSE112618 & GSE110554]. Both papers describe reconstruction experiments (the first uses the 450K array and the second, the EPIC array) where the six major leukocyte cell types were mixed in predetermined proportions (e.g., “known” proportions).

We sincerely thank the reviewer for the constructive advice. We are sorry that in the previous revision we were not aware that neutrophils and eosinophils are subcategories of granulocytes. As a result, we included granulocytes together with neutrophils and eosinophils into the reference panel. In the revised manuscript, we have removed granulocytes from the reference [7] and instead focused on granulocyte subtype collections. The reference now comprises seven cell types: CD4+ T, CD8+ T, monocytes, B cells, NK, neutrophils, and eosinophils, which are distinct leukocyte cell types.

We have subsequently aligned the cell-type-specific methylation profiles estimated from the RA and GALA II datasets with the new reference. In RA, cell type 1 was aligned with CD4+ T, and cell types 2 and 4 both corresponded with neutrophils. With respect to the GALA II dataset, cell type 1 was annotated as CD4+ T, and cell types 2, 3, and 4 all represented neutrophils. Compared to the original alignment results, the cell types previously aligned with granulocytes are now annotated as neutrophils. Moreover, the average proportion of neutrophils across the samples in GALA II and RA were 41.61% and 30.87%, respectively. These large proportions indicate that neutrophils are a major constituent in whole blood, which is consistent with the reviewer’s note “*neutrophils typically comprise 50-70% of circulating cell types in whole blood.*” The 10% gaps might be due to population heterogeneity.

Regarding the reviewer’s concern about the unaligned cell types, we originally set the similarity threshold as 0.7, following the Nature Methods paper [5]. A high threshold can guarantee a low possibility of falsely aligning a cell type. However, in practice,

to annotate more cell types and make the findings more biologically meaningful, we allowed a less conservative threshold. Therefore, in the revised manuscript, we changed the threshold to 0.5. In GALA II, when we lowered the similarity threshold to 0.5, cell type 5 was aligned with CD4+ T, and cell type 6 was annotated as NK. However, for the RA data with the 0.5 threshold, cell types 3, 5, and 6 still could not be annotated. Only once the similarity threshold was lowered to 0.05, could these cell types be mapped to neutrophils, neutrophils, and CD8+ T, respectively. Therefore, we retained the similarity threshold at 0.5 for both RA and GALA II.

Following Reviewer 2's suggestion, we used the data collected in GSE110554 [8] to further evaluate HIRE's ability to annotate inferred cell types and estimate the underlying cell proportions. GSE110554 contains purified cell-type-specific methylation profiles for six cell types: neutrophils, monocytes, B cells, CD4+ T, CD8+ T, and NK. Moreover, GSE110554 includes mixed samples, whose methylation signals were aggregated from the six cell types with pre-determined cell mix proportions. Therefore, because of the known cell type and cellular proportion information, GSE110554 is an ideal dataset with which to test HIRE's performance.

In GSE110554, the number of mixed samples is much smaller than the typical size of an EWAS and, as discussed in the manuscript, HIRE usually requires hundreds of samples to obtain accurate and stable results. Therefore, to increase the sample size, we first generated a simulated methylation dataset with 600 samples using the purified methylation profiles. We focused on 10k CpG sites including the 450 IDOL CpG sites, which were previously identified as the optimal library of CpG sites for estimating leukocyte subtype proportions [8], and another 9550 CpG sites whose methylation values across the purified cell types were in the range of [0.2, 0.8] and had large standard deviations. Subsequently, we combined the 600 samples and 6 mixed samples (generated by method A) [8] available in GSE110554 to compose a semi-simulated dataset.

After applying HIRE to the semi-simulated data, we annotated the estimated cell types based on the methylation profiles from GSE110554. Fig. R1 shows the heatmap for the Pearson correlation matrix between inferred cell types and the underlying truth. The correlation signals on the diagonal are the strongest in each row. HIRE successfully recovered the six underlying cell types. We also compared the estimated cellular compositions with the underlying true proportions for the 6 mixed samples. In Fig. R2, each panel displays the scatter plot between the cellular proportion estimates and the true mix proportions for a given cell type, all of which indicate that HIRE obtains good estimates for cellular compositions.

Figure R1: (a) The heatmap of the Pearson correlation matrix between the estimated cell types and the biological cell types for the semi-simulated dataset. The red indicates positive values; the blue represents negative values; and values near zero are in white. (b) The heatmap of the Pearson correlation matrix for the reference itself.

Figure R2: The scatter plots for the estimated cell proportions versus the underlying truth for the six mix samples with known cell type mixing proportions in the semi-simulated dataset. Each panel corresponds to one of the six cell types used for mixing: B cell, NK, monocytes, neutrophils, CD4+ T, and CD8+ T.

3 Reviewer 3

Most of my previous comments were not well addressed. In addition, a few newly added figures exposed further flaws with the proposed method:

(1) In my previous comment 1, I was concerned with the novelty of the proposed method. In particular, a paper by Houseman et al published in 2016 in the journal of BMC Bioinformatics is already capable of detecting cell-type-specific risk-CpG sites. I appreciate that the authors provide detailed equations to explain the difference between their HIRE approach and the Houseman 2016 method. However, these new explanations do not directly address my previous concern. Let's stick to authors' notations this time. The Houseman 2016 BMC Bioinformatics paper effectively performs a matrix factorization on the methylation data. In a probabilistic framework, the Houseman model can be effectively written as $O_i = M \cdot \pi_i + e_i$, with e_i being the residual errors. While M is certainly the same across all cells, the product term $M \cdot \pi_i$ provides one with a cell-subtype specific estimate: that is, the product term $M \cdot (\pi_i)_k$, by using the k 'th element of π_i , gives one an estimate of the k 'th cell type specific methylation levels across all CpG sites for i 'th individual. Therefore, for each cell type k and each CpG site j in turn, you can treat $M_{j \cdot} \cdot (\pi_i)_k$ across individuals as the outcome variable and perform a linear regression against any predictor variable x . This linear regression would allow you to detect any cell-type-specific risk-CpG sites. This is the two-step procedure I mentioned in the original comment 1. And this is such a simple procedure that you don't even need to rely on the RefFreeEWAS software to do so. Because the existing Houseman method already provides a straightforward way of detecting cell-type-specific risk-CpG sites, I do not think the current paper provides enough methodological advances or novelty over the existing method. In addition, a detailed comparison with the Houseman 2016 BMC Bioinformatics method in all simulations and real data applications is warranted.

Reviewer 3 gave a detailed description of a “two-step” approach in comment 1. We are very grateful for the kind efforts to help us improve our paper, but after applying the two-step approach to simulated and real-world data we are afraid that the two-step approach proposed by Reviewer 3 cannot successfully detect cell-type-specific risk-CpG sites based on the following demonstration and evidences.

Here, to simplify the logic, please allow us to first illustrate Reviewer 3's “two-step” approach with an analogy. The mathematical derivation for Reviewer 3's “two-step” approach and its results in both the simulation study and two real datasets are detailed after the toy example.

Suppose we are interested in studying each country's average income versus its literacy rate. For a given country:

The average income = the average income of males \times the proportion of males + the average income of females \times the proportion of females.

Recall that in an EWAS study, assuming that there exist two and only two cell types—the white blood cells and the red blood cells—we have:

The average methylation level = the average methylation of white blood cells \times the proportion of white blood cells + the average methylation of red blood cells \times the proportion of red blood cells.

Reviewer 3 proposed to impose the following model: for a given person i ,

The average methylation level (O_i) = the average methylation of white blood cells (M_1) \times the proportion of white blood cells (p_{i1}) + the average methylation of red blood cells (M_2) \times the proportion of red blood cells (p_{i2}),

where the “**M** = (M_1, M_2) is certainly the same across all cells, the product term $M * p_i$ provides one with a cell-subtype specific estimate: that is, the product term $M * p_{ik}$ by using the k 's element of p_i , gives one an estimate of the k 's cell type specific methylation levels across all CpG sites for i^{th} individual” as stated by Reviewer 3 in comment 1.

What Reviewer 3 is saying for the income vs literacy analogy is:

- The female’s average income and the male’s average income stay the same across all the countries (“**M** is certainly the same across all cells”). Only the proportion of females varies by country; and
- The product of the average income of females and the proportion of females gives a “female-specific estimate.” Reviewer 3 further advocates to regress this “female-specific estimate” on the literacy rate of each country. Because Reviewer 3 actually assumes that M_2 , the average income of females, does not change across countries, this regression is actually equivalent to regressing the proportion of females in each country on the literacy rate of each country.

However, we guess most people would agree that **the average incomes of females are not the same for all countries** and would be more interested in understanding how **females’ average income** is linked to the literacy rate of a country than learning how **the female proportion of a country** is associated with the literacy rate. Therefore, it is not surprising that Reviewer 3’s “two-step” approach performs poorly in both the simulation study and the two real datasets, as shown later.

In contrast, our HIRE model allows **the average income of females** to vary by country according to the country’s literacy rate, and HIRE is able to find the effects of literacy rate on the males’ average income and on the females’ average income, respectively, thus in a gender-specific way that is impossible by existing methods. Moreover, by modeling the real relationship “the average methylation level (O_i) = the average methylation of white blood cells (u_{i1}) \times the proportion of white blood cells (p_{i1}) + the average methylation of red blood cells (u_{i2}) \times the proportion of red blood cells (p_{i2})” correctly, HIRE allows both the cell-type-specific methylation level

u_{ik} and the cellular-compositions p_{ik} to be linked to covariates of interest.

In the following, we explain the two-step approach proposed by Reviewer 3 in detail and compare it with HIRE using both the simulated data and two real EWAS datasets.

Reviewer 3 proposed a two-step procedure in his or her comment 1 in an attempt to detect cell-type-specific risk-CpG sites.

- In step one, the observed methylation matrix \mathbf{O} , with m CpG sites in rows and n samples in columns, is decomposed into two matrices \mathbf{M} and \mathbf{p}_i using a constrained quadratic programming approach [2]. \mathbf{M} can be interpreted as the estimated cell-type-specific methylation profiles, and its (j, k) element M_{jk} represents the methylation value of CpG site j in cell type k for $1 \leq j \leq m$ and $1 \leq k \leq K$ (K is the cell type number). \mathbf{p} is the estimated cellular composition matrix, and p_{ik} is the proportion of cell type k of sample i .
- In step two, for each CpG site j and each cell type k , Reviewer 3 proposed to treat the vector $(M_{jk}p_{ik} : 1 \leq i \leq n)$ as the outcome vector and then regress it on each predictor variable—the phenotype covariate— $(x_{\ell i} : 1 \leq i \leq n)$ separately. In this way, Reviewer 3 thought that we could obtain the p-value to test whether there is any association between CpG site j and phenotype ℓ in cell type k .

Consequently, Reviewer 3 claimed that this two-step approach *“would allow you to detect any cell-type-specific risk-CpG sites”* and *“do not think the current paper provides enough methodological advances or novelty over the exiting method.”*

Let us first evaluate the above two-step approach on simulated data ($K = 3$ and $n = 180$) and the two real-world methylation datasets—RA and GALA II. We then compare the results against HIRE in terms of p-value histograms, Q-Q plots, and detected association patterns.

In Figures R3-R5, panel (a) shows the p-value histograms of the two-step approach from the simulation data, the RA, and GALA II datasets, respectively. We can see that there are only spikes in the histogram, meaning that the p-values only take values at several points. By checking the p-values, we found that within each cell type, the p-values (except for NAs) are the same across all CpG sites. As a result, the quantile-quantile lines in the Q-Q plots (see panel (c) of Figures R3-R5) for the two-step approach are step functions, where each step corresponds to all of the identical p-values in one cell type, and the number of steps is exactly equal to the number of cell types. The claimed association patterns in Panel (e) of Figures R3-R5 further show that in each cell type, the p-values (except for NAs in grey) for all CpG sites are identical. Moreover, for the simulated data, the false positive rate (FPR) of the two-step approach is 33.26% and its power is 33.33%, whereas HIRE’s FPR is only 0.01% and its power is as large as 88.33%. All of these observations show that the two-step approach fails to discover any cell-type-specific risk-CpG sites.

In comparison, the p-values from HIRE reflect the underlying cell-type-specific association pattern. Panel (f) of Figures R3-R5 shows that HIRE can reveal the underlying association pattern. Moreover, the p-values from HIRE display good properties—most of p-values are distributed uniformly between zero and one and a small proportion of them concentrate near zero [9]. Please see panels (b) and (d) of Figures R3-R5.

Therefore, all of the data analyses, whether in the simulation study or with the real-world datasets, clearly show that the two-step approach mentioned by Reviewer 3 completely fails to detect any cell-type-specific risk-CpG sites. In contrast, by modeling the dependence of cell-type-specific methylation profile \mathbf{u}_i of sample i on the phenotype covariates \mathbf{x}_i (see Figure 1 in our manuscript), our HIRE method is able to discover cell-type-specific risk-CpG sites, which cannot be achieved by existing EWAS methods.

Actually, from the statistical perspective, it is not surprising that the two-step approach give meaningless results in detecting risk-CpG sites. In step two, Reviewer 3 proposes to regress $M_{jk}p_{ik}$ ($i = 1, \dots, n$) on covariates $x_{\ell i}$ ($i = 1, \dots, n$) to obtain the p-value for the effect of phenotype ℓ on CpG site j in cell type k . However, because M_{jk} is assumed to be the same for all of the individuals by Reviewer 3 and thus irrelevant to the sample index i , when $M_{jk} \neq 0$, regressing $M_{jk}p_{ik}$ on $x_{\ell i}$ ($i = 1, \dots, n$) is equivalent to regressing p_{ik} on $x_{\ell i}$ ($i = 1, \dots, n$) in terms of determining whether the coefficient of x_{ℓ} is zero. Notice that the equivalent regression $p_{ik} \sim x_{\ell i}$ has nothing to do with the CpG site index j . Therefore, in each cell type, we observed identical p-values across all CpG sites. The p-values only partially reflect the association between cellular compositions and the phenotype; they do not indicate any association between the phenotype and the methylation levels of a CpG site in a given cell type, whereas the latter is our main interest.

Figure R3: The performance comparison between the two-step approach and HIRE on simulated data. (a) p -value histogram, (c) Q-Q plot, and (e) association pattern based on p -values from the two-step approach. (b) p -value histogram, (d) Q-Q plot, and (f) association pattern based on p -values from HIRE.

Figure R4: The performance comparison between the two-step approach and HIRE on the RA methylation data. (a) p -value histogram, (c) Q-Q plot, and (e) association pattern based on p -values from the two-step approach. (b) p -value histogram, (d) Q-Q plot, and (f) association pattern based on p -values from HIRE.

Figure R5: The performance comparison between the two-step approach and HIRE on the GALA II methylation data. (a) p -value histogram, (c) Q-Q plot, and (e) association pattern based on p -values from the two-step approach. (b) p -value histogram, (d) Q-Q plot, and (f) association pattern based on p -values from HIRE.

(2) In my previous comment 2, I was concerned that the authors compared their HIRE

method to a bunch of unrelated methods including SVA, EAWShEr, ReFACTor. As I mentioned earlier, these methods are designed to control for hidden confounding factors that are correlated with cell compositions and the predictor variable of interest. Without controlling for these hidden confounding factors, one would incorrectly detect many false cell-type-specific risk-CpG sites that are not associated with the predictor variable or in a cell-type-specific fashion. On the other hand, the proposed HIRE method (and Houseman 2016 BMC Bioinformatics method) is more focused on detecting signals in the absence of confounding effects. Therefore, I initially suggested to remove these three methods (SVA, EAWShEr, ReFACTor) in comparison. However, it is also completely ok if the authors decide to keep them – as they have done now – as long as they also follow my second suggestion of performing extensive additional simulations in the presence of hidden confounding effects based on the EAWShEr and ReFACTor papers. For example, what happens if the hidden confounding factors are correlated with the predictor variable, or the cellular composition, or both, and if the hidden confounding factors affect many null CpG sites? However, the authors also failed to follow my second suggestions to perform these simulations. As of now, all simulations in the currently manuscript are completely in favor of their own method and fails to examine the important setting where there are hidden confounding factors. Please follow my previous comment 2 and add new simulations to examine these important settings.

We thank the reviewer for the comment. To begin with, our simulation setting not only emulated the data-generating procedure of EWASher [10] and ReFACTor [6], which Reviewer 3 agreed “*are designed to control for hidden confounding factors that are correlated with cell compositions and the predictor variable of interest*” in comment 2, but also accounted for real-world factors including cell lineage, directions and magnitudes of phenotype effects in addition. To be specific, EWASher “took weighted average ... with weights given by cell-type composition” [10]. In ReFACTor, “observed DNA methylation levels were composed for each individual by its simulated methylation levels and cell proportions” [6]. Both papers used the weighted average $\sum_{k=1}^K u_{ijk} p_{ik}$ to generate methylation data but considered only the case/control phenotype when simulating u_{ijk} . In our paper, we have similar data-generating mechanisms, and we further allow u_{ijk} to depend on multiple phenotypes—case/control and age—and to be influenced by other real-world factors such as cellular lineage, which previous simulation studies did not account for.

Moreover, the “key confounder” EWASher considers is exactly the cellular composition (see Supplementary Note 2 for: Epigenome-Wide Association Studies without the Need for Cell-Type Composition; page 1, paragraph 1, line 4). Specifically, EWASher is mainly based on the linear mixed model (LMM) to account for confounding factors. “LMMs tackle confounders by modeling the similarity between every pair of individuals.” However, “the similarity between sample i and sample j ” considered by EWASher is “ $K(i, j) = \frac{1}{M} \mathbf{G}_i^T \mathbf{G}_j = \frac{1}{M} \mathbf{W}_i^T \mathbf{R}^T \mathbf{R} \mathbf{W}_j$ ”, with the number of CpG sites denoted by M , the observed methylation profile for sample i by \mathbf{G}_i , the reference methylation profiles by \mathbf{R} , and the cell-type composition vector by \mathbf{W}_i , all of which equal m , \mathbf{O}_i^T , μ , and \mathbf{p}_i in HIRE, respectively. Please refer to Sections 1-3 of

Supplementary Note 2 for: Epigenome-Wide Association Studies without the Need for Cell-Type Composition for more details. ReFACTor builds on EWASher and thus shares the same model framework. For both EWASher and ReFACTor, cellular heterogeneity is the only confounder they are concerned with in their simulation settings. As discussed in our manuscript, with the hierarchical structure in Figure 1 of our manuscript, HIRE can automatically tease out the cell heterogeneity, thus naturally taking care of the confounding scenario handled in EWASher and ReFACTor.

(3) The authors now show that their proposed method fails to control for type I error. Specifically, the newly added supplementary Figures S1-9 show that the method produces highly inflated p-values under null even when the simulations are conducted favoring their own modeling assumptions. Perhaps with the only exception of Figure S9, their p-values under the null are much worse behaved than the other competing methods such as EWASher or ReFACTOR. Failing to control for type I error is a huge issue for real data applications, as the p-values from the proposed method are no longer meaningful. The ability to control for type I error is such an important issue that the author should move one of these null supplementary Figures to a main figure. In addition, the authors have only explored a limited number of null settings. It is important to explore various null scenarios, with various parameter choices and various deviations from their own modeling assumptions, to examine whether their p-values are calibrated under the null. For example, will the p-value distribution become even worse if you increase the number of CpG sites to a realistic value, for example, 450K or 850K? What happens if you introduce a larger number of cell types, say $k=10$ or 20 ? And what happens if cell type compositions becomes more heterogeneous, with perhaps a few cell types dominate the composition? And will introducing additional hidden confounding effects that are correlated with cell compositions or predictor variable make things even worse? In any case, the authors need to make significant modifications of their method to provide calibrated p-values.

We thank the reviewer for the questions. Table 1 in the manuscript has shown that HIRE controlled the false positive rate (FPR) very well in the simulation studies using the Bonferroni correction. In fact, the function of Q-Q plot is showing whether the p-values are from the uniform distribution on $[0, 1]$ and is irrelevant to FPR. Even in terms of Q-Q plots, HIRE is comparable to ReFACTor in most cases (See Supplementary Figures S1, S2, S3, S5, S6, S9). Although EWASher and ReFACTor had lower FPRs in some settings, their detection power is much lower than that of HIRE in all cases. In addition, the ROC curves of HIRE are uniformly better than those of EWASher and ReFACTor in all settings. These are strong evidences that HIRE outperforms EWASher and ReFACTor in a wide range of scenarios.

In fact, when the sample size is small, we can adopt a special case of HIRE to further improve the Q-Q plot performance by reducing the large parameter number in HIRE. We can reparameterize all σ_{gk}^2 s as one single parameter σ^2 , and we found that such a variance-stabilized approach can obtain even better inflation control (see Figures R6,R7 and R8) with comparable power as the full model version of

HIRE (see Table R1). Therefore, for small sample sizes, we suggest that users can try both HIRE and its special case first and then select the one that gives better Q-Q plots.

Figure R6: The Q-Q plots by HIRE and its special case for the “true null” simulation settings: (a) $K=3$ and $n=180$; (b) $K=3$ and $n=300$; (c) $K=3$ and $n=600$. The Q-Q lines by HIRE’s special case are near the $y = x$, thus controlling the inflation very well.

Figure R7: The Q-Q plots by HIRE and its special case for the “true nul” simulation settings: (a) $K=5$ and $n=180$; (b) $K=5$ and $n=300$; (c) $K=5$ and $n=600$. The Q-Q lines by HIRE’s special case are near the $y = x$, thus controlling the inflation very well.

Figure R8: The Q-Q plots by HIRE and its special case for the “true null” simulation settings: (a) $K=7$ and $n=180$; (b) $K=7$ and $n=300$; (c) $K=7$ and $n=600$. The Q-Q lines by HIRE’s special case are near the $y = x$, thus controlling the inflation very well.

In our simulation studies, we have generated extensive null scenarios with different subtype numbers, different sample sizes, and various parameter choices, please see Section “Data simulation” in the manuscript.

Table R1: The power of HIRE’s special case for the disease status and the age under the “true alternative” simulation settings.

Disease Status	$n = 180$	$n = 300$	$n = 600$
$K = 3$	96.67%	96.67%	100%
$K = 5$	58%	82%	94%
$K = 7$	30%	53.33%	86.67%
Age	$n = 180$	$n = 300$	$n = 600$
$K = 3$	96.67%	100%	100%
$K = 5$	98%	98%	100%
$K = 7$	65%	85%	98.33%

As both EWASHer and ReFACTor automatically filter out more than 75% of CpG sites, we focused on simulation datasets with 10k CpG sites to avoid a heavy computation load. We actually have tried several settings with 450k CpG sites and obtained similar results to those for the 10k CpG sites. Therefore, in the manuscript, we ran all of the simulation studies with 10k CpG sites to obtain the results for the total $3(\text{cell type number}) * 3(\text{sample size}) * 2(\text{true null setting and true alternative setting}) * 5(\text{replicate}) = 90$ simulation datasets in a short timer period.

We actually considered settings with more heterogeneous cell type compositions in the simulation study. When $K = 7$, we sampled the cellular composition $\mathbf{p}_i \sim Dir(1, 3, 3, 3, 2, 2, 2 + 0.1x_{i2})$ for controls and $\mathbf{p}_i \sim Dir(1, 3, 3, 3, 2, 2, 5 + 0.1x_{i2})$ for cases. The least abundant cell type takes up about 4.2% of the total population, HIRE still enjoys a high power with only 180 samples at the aggregated level (see Table 1 in the manuscript). In principle, a rare cell type should ask for a larger sample size for detection, and to the best of our knowledge, all of the existing methods for EWAS consider K less than 10 in their applications. Regarding “introducing additional hidden confounding effects,” please refer to our answers to point 2 of Reviewer 3 for details.

(4) For the previous comment 4, please refer to my elaborated comment 1 and address these previous concerns accordingly. Again, Houseman method is a very simple procedure that you don’t even need to rely on the RefFreeEWAS software to fit a factor model and run a linear regression.

We thank the reviewer for the comment. We have examined the performance of the two-step approach extensively on simulation studies and real data. However, the two-step approach actually did not work well. Please refer to our answers to point 1 of Reviewer 3 for details.

(5) For the previous comment 5, please refer to my elaborated comment 1 and address these previous concerns accordingly. Please also simulate data under the Houseman additive model and examine whether the proposed multiplicative procedure would work well when the truth is additive.

We thank the reviewer for the comment. We have examined the performance of the two-step approach extensively on simulation studies and real data. However, the two-step approach actually did not work well. Please refer to our answers to point 1 of Reviewer 3 for details.

(6) For the previous comment 6, please also show p-value distribution under permutations in both simulations and real data. Examining p-value distribution with permutations is perhaps the only way to examine type I error in real data sets.

Following the reviewer’s suggestion, we ran the permutation tests for each CpG site and compared the collected p-values against the p-values from HIRE in simulation studies. However, we observed that the p-values from permutation tests performed poorly in controlling the false positive rate (FPR) and showed substantially inflated p-value density plots. Here, we used a simulated dataset with three underlying cell types, two phenotypes (case/control and age), 180 samples and 10k CpG sites as an illustration. Using the Bonferroni correction, HIRE’s power can be 100% with a very small FPR 0.02%, while the permutation tests suffered a large FPR 61.56%. The FPR inflation of the permutation p-values was further reflected by the Q-Q plot in Figure R9(b) and the density plot in Figure R10(b), where a large proportion of p-values concentrate near zero. The ROC curves in Figure R11 demonstrated that the permutation tests even cannot surpass the “random guess.”

In fact, the reason for the poor performance of permutation in our setting is that its required exchangeability under the null hypothesis is violated in presence of multiple covariates, as pointed out in the paper [1] by the well-known statistician and bioinformatician Prof. Gary Churchill (<http://churchill-lab.jax.org/website/content/gary-churchill-0>). In contrast, HIRE is a reliable approach to obtaining well-behaved p-values and achieving a high detection power, please see Figures R9(a), R10(a) and R11. These evidences confirm that permutation does not work well if its required condition is not satisfied in the presence of multiple covariates.

In the real data analysis of our study, multiple covariates are presented and the exchangeability condition for permutation test is not satisfied. The permutation results should not serve as a golden standard for real data analysis. Therefore, we did not report the p-values from the permutation tests in our manuscript. We thank the reviewer for his or her understanding.

Figure R9: The Q-Q plots of the p-values from HIRE and the permutation tests. (a) The HIRE p-value Q-Q plot. (b) The Q-Q plot of the permutation test p-values, where zero p-values were removed and the Q-Q line deviates from the straight line $y = x$ at the very beginning.

Figure R10: The density plots of the p-values from HIRE and the permutation tests. (a) The HIRE p-value density plot. (b) The density plot of the permutation test p-values, where most of the p-values are equal to zero.

Figure R11: The ROC curves of HIRE, the permutation tests and the “random guess.”

(7) For the previous comment 8, you can use the probabilistic factor model that the Houseman method is effectively based on to compute a likelihood. The likelihood is straightforward to write down. Afterwards, you can add the penalty term to obtain the corresponding BIC for the Houseman method.

We thank the reviewer for the comment. To obtain BIC for the Houseman method, we need a probabilistic model. RefFreeEWAS [4] relies on the single value decomposition (SVD), a numerical matrix decomposition method without any distribution assumption for the data. We investigated the R package “RefFreeEWAS” carefully and found the function “EstDimIC” that calculates the BIC value. Although “EstDimIC” does not have detailed documentation describing the assumed probabilistic model, our close inspection of Houseman’s method indicates that it imposed a set of assumptions with normality. Following the reviewer’s comment, we used simulation study to check whether Houseman’s method and our method could identify the correct number of cell types. We found that the number of cell types selected by BIC for HIRE and Houseman’s method was generally consistent with each other. Please see Figure R12 for an illustration when there are three cell types.

Figure R12: The BIC plots for a simulated dataset with the sample size 180 and the cell type number three by (a) Houseman’s method and (b) HIRE, respectively. Both of them attain minimum at three.

(8) For the previous comment 9, please do provide discussion on the effects of different normalization procedure – the proposed model is based normality assumption, and the beta or M values for many CpG sites are not normally distributed.

We appreciate the reviewer’s comment. Yes, we imposed a normality assumption in HIRE. Indeed, several other methods were built upon the normality assumption, such as the Houseman method [4] mentioned by the reviewer. There are several considerations for the normality assumption. First, although the methylation status at a given site of a cell should be binary in theory, the methylation level assessed by array could be approximated using normal distribution because the array signal actually is a reflection of a large number of cells with individually different methylation status. Second, in quality control, we often exclude the methylation sites whose mean values that are near the boundary, making the normality assumption more reasonable. In fact, methods with the normality assumption have shown reasonably good performance in real data analysis [3, 6].

We acknowledge that the normal assumption in the HIRE model does not ensure that the observed methylation beta-values are always between zero and one. To address this issue, we do constrain the baseline methylation profiles μ_{jk} s to the closed interval $[0, 1]$ and force the cellular compositions p_{ki} s to be non-negative and sum up to one: $\sum_{k=1}^K p_{ki} = 1$. As a result, as the phenotypes have no effect on most CpG sites, most observations, O_{ji} s, have the mean $\sum_{k=1}^K \mu_{jk} p_{ki}$ in $[0, 1]$. To verify that, in the real application, we used HIRE estimates to calculate the fitted methy-

lation values \hat{O}_{ji} s. It turns out that more than 99.99% of the fitted methylation values are between zero and one for both the RA dataset and the GALA II dataset. Therefore, the normality assumption does not have a large effect on the performance of HIRE, serving as a useful assumption in probabilistic inference for methylation data..

(9) For the previous comment 10, because the proposed method does not provide calibrated type I error control (comment 3 above), it becomes critical to run permutations in order to compare different methods in real data in a fair fashion. Please compare all methods based on the same threshold established by permutations. Otherwise, it is impossible to judge the performance of the proposed HIRE method in real data.

We thank the reviewer for the comment. As we argued above, permutation results should not always serve as the golden standard for real data analysis because its required exchangeability condition may not be satisfied. Please refer to our answers to point 6 for details.

(10) For the previous comment 11 on population structure concerns. If the authors believe that their method is free of population structure concern, please provide simulations to examine settings where population structure is correlated with the predictor variable of interest or cellular composition and affects a large number of null CpG sites. I appreciate that the authors performed simulations where both the cellular compositions and the cell-type-specific methylation levels depend on the phenotypes. However, the authors haven't checked the population stratification setting, where the cell-type-specific methylation levels depend on population structure, while population structure are not fully accounted for by the covariates. In the example of GALA II, what if the self-reported ethnicity is not accurate? Do you have genotypes to extract PCs to adjust for structure? In the equation provided on page 31, in the case of population stratification, these $\beta_{ajkl} \cdot x_{il} \cdot p_{ki}$ terms actually represent population stratification, which are something we want to remove, not to detect.

We thank the reviewer for the comment. On the one hand, our HIRE model can adjust the population structures by incorporating them in the covariates using dummy variables. In this case, the cell-type-specific methylation profiles do depend on the population structures. On the other hand, principal components extracted from genotype data could be included as covariates if the self-reported ethnicity is not accurate.

For the GALA II dataset, we have four ethnicity groups—“Mexican,” “Mixed Latino,” “Puerto Rican,” and “Other Latino.” Therefore, we created three dummy variables to represent the four ethnicity groups. In particular, we used $(0, 0, 0)$, $(1, 0, 0)$, $(0, 1, 0)$, and $(0, 0, 1)$ to represent “Mexican,” “Mixed Latino,” “Puerto Rican,” and “Other Latino,” respectively. We incorporated the ethnicity indicators as phenotype covariates

$x_{i\ell}$ into HIRE:

$$E [O_{ji}] = \mu_{j1} + \sum_{k=2}^K (\mu_{jk} - \mu_{j1}) p_{ki} + \sum_{k=1}^K \sum_{\ell=1}^q \beta_{jk\ell} x_{i\ell} p_{ki}.$$

In the above equation, the population structures coded in $x_{i\ell}$ s actually affect the observed methylation level via the third term on the right-hand side in addition to the cell proportions at the second term. Therefore, HIRE takes into account the cellular heterogeneity potentially affected by population structures and models the effects of population structures on the observed methylation levels simultaneously. In particular, given its ability to detect cell-type-specific risk-CpGs, HIRE is able to identify whether two ethnic groups have differentially methylated CpGs for each cell type.

For our simulation studies, we have considered the correlation between cellular compositions and covariates. In each of our simulation settings, the cellular compositions were drawn from the Dirichlet distribution whose parameters always depend on the covariates. With the correlation in data, HIRE still provides well-behaved Q-Q plots, providing a justification of HIRE for adjusting population structure by using covariates. Please see Figure R13 for an illustration.

Figure R13: Two illustrative Q-Q plots from HIRE when the cellular compositions are correlated with covariates in simulation studies: (a) $K=3$ and $n=300$; (b) $K=5$ and $n=600$.

References

- [1] GA Churchill and RW Doerge. Naive application of permutation testing leads to inflated type I error rates. *Genetics*, 178(1):609–610, 2008.
- [2] E Andres Houseman, Molly L Kile, David C Christiani, Tan A Ince, Karl T Kelsey, and Carmen J Marsit. Reference-free deconvolution of DNA methylation data and mediation by cell composition effects. *BMC Bioinformatics*, 17(1):259, 2016.
- [3] Eugene Andres Houseman, William P Accomando, Devin C Koestler, Brock C Christensen, Carmen J Marsit, Heather H Nelson, John K Wiencke, and Karl T Kelsey. DNA methylation arrays as surrogate measures of cell mixture distribution. *BMC Bioinformatics*, 13(1):86, 2012.
- [4] Eugene Andres Houseman, John Molitor, and Carmen J Marsit. Reference-free cell mixture adjustments in analysis of DNA methylation data. *Bioinformatics*, 30(10):1431–1439, 2014.
- [5] Vladimir Yu Kiselev, Andrew Yiu, and Martin Hemberg. scmap: projection of single-cell RNA-seq data across data sets. *Nature Methods*, 15(5):359, 2018.
- [6] Elior Rahmani, Noah Zaitlen, Yael Baran, Celeste Eng, Donglei Hu, Joshua Galanter, Sam Oh, Esteban G Burchard, Eleazar Eskin, James Zou, et al. Sparse pca corrects for cell type heterogeneity in epigenome-wide association studies. *Nature Methods*, 13(5):443, 2016.
- [7] Lovisa E Reinius, Nathalie Acevedo, Maaïke Joerink, Göran Pershagen, Sven-Erik Dahlén, Dario Greco, Cilla Söderhäll, Annika Scheynius, and Juha Kere. Differential dna methylation in purified human blood cells: implications for cell lineage and studies on disease susceptibility. *PloS One*, 7(7):e41361, 2012.
- [8] Lucas A Salas, Devin C Koestler, Rondi A Butler, Helen M Hansen, John K Wiencke, Karl T Kelsey, and Brock C Christensen. An optimized library for reference-based deconvolution of whole-blood biospecimens assayed using the illumina humanmethylationepic beadarray. *Genome Biology*, 19(1):64, 2018.
- [9] John D Storey and Robert Tibshirani. Statistical significance for genomewide studies. *Proceedings of the National Academy of Sciences*, 100(16):9440–9445, 2003.
- [10] James Zou, Christoph Lippert, David Heckerman, Martin Aryee, and Jennifer Listgarten. Epigenome-wide association studies without the need for cell-type composition. *Nature Methods*, 11(3):309, 2014.

Reviewers' comments:

Reviewer #2 (Remarks to the Author):

I appreciate the authors willingness to respond to the comments/suggestions that were raised in the previous review. I am satisfied with their response and feel that the new additions strengthen their manuscript.

Reviewer #4 (Remarks to the Author):

I reviewed the revised version of the manuscript. I decided to not read the previous reviews and response, so this can be considered a "fresh" review.

The authors tackle an ambitious problem of deconvolving the bulk DNA methylation data of individuals into cell type specific methylation as well as covariate specific effects on the methylation for each cell type. I have some significant concerns about this model and approach.

1. When is the model identifiable? The model has a lot of parameters that need to be jointly estimated. Even just estimating the baseline cell type specific methylation (μ) and the cell type proportions for all the individuals (P) is a matrix factorization model, which is not always identifiable. Adding the additional covariate effects on each cell type greatly complicates the model and I would like to see a thorough and rigorous discussion of when is the model mathematically valid/identifiable.

2. The authors' evaluations seem to focus on the p values and the testing of the differential CpG's, which is certainly important. However in order to have full confidence that these p values are valid, we need to also know that the other estimated parameters (μ , P , B) are also correct. The authors should show how their estimates for these parameters compare with the ground truth in their simulations and with the best known experimental measurements in the real data.

3. It'd be useful to see more systematic analysis of the robustness of the model when there is model mismatch. This is especially important given the model complexity.

We would like to express our deepest gratitude to the Editors and the two reviewers for their precious time spent on reviewing our manuscript. We greatly appreciate the positive comments, such as “satisfied with their response” and “new additions strengthen their manuscript.” We have taken Reviewer 4’s advice very carefully and thoroughly. We proved the identifiability of the HIRE model under mild conditions that are easily met in reality. We then demonstrated HIRE’s good performances in estimating the baseline methylation profiles μ , the cellular compositions \mathbf{P} , and the phenotype effects \mathbf{B} . Finally, we checked the robustness of HIRE when the model is misspecified.

We revised our manuscript and Supplementary Materials accordingly. In the following, we provide point-by-point responses to the two reviewers. The comments from the reviewers were displayed in *blue and italics*. In the main manuscript, we highlight our modifications in *red*.

1 Reviewer 2

I appreciate the authors willingness to respond to the comments/suggestions that were raised in the previous review. I am satisfied with their response and feel that the new additions strengthen their manuscript.

We really appreciate the positive comments by the reviewer.

2 Reviewer 4

I reviewed the revised version of the manuscript. I decided to not read the previous reviews and response, so this can be considered a “fresh” review.

The authors tackle an ambitious problem of deconvolving the bulk DNA methylation data of individuals into cell type specific methylation as well as covariate specific effects on the methylation for each cell type. I have some significant concerns about this model and approach.

We thank the reviewer for confirming the novelty of HIRE and giving a thorough review.

1. When is the model identifiable? The model has a lot of parameters that need to be jointly estimated. Even just estimating the baseline cell type specific methylation (μ) and the cell type proportions for all the individuals (P) is a matrix factorization model, which is not always identifiable. Adding the additional covariate effects on each cell type greatly complicates the model and I would like to see a thorough and rigorous discussion of when is the model mathematically valid/identifiable.

We thank the reviewer for the insightful comments.

Although the non-negative matrix factorization (NNMF) $\mathbf{O} = \boldsymbol{\mu}\mathbf{P}$ has been widely applied in cell type deconvolution (Houseman et al., 2016), where \mathbf{O} is the observed methylation matrix, $\boldsymbol{\mu}$ is the unknown methylation profile, and \mathbf{P} is the unknown cellular compositions, the model identifiability is rarely discussed. During the review period of our paper, Rahmani et al. (2018) provides a setting under which the NMMF model is not identifiable, just as Reviewer 4 mentioned.

Then, why does NNMF always provide satisfactory cell type deconvolution results in real practice and why can HIRE estimate all those parameters well? Here, we mathematically show that the HIRE model is identifiable under mild conditions that are easily met in reality.

Let us first introduce some notations and definitions. In the HIRE model, the whole parameter set is denoted by $\Theta := \{\mathbf{P}_i, \boldsymbol{\mu}_j, \mathbf{B}_\ell^{(j)}, \sigma_{jk}^2, \sigma_{ej}^2 : 1 \leq j \leq m, 1 \leq i \leq n, 1 \leq k \leq K, 1 \leq \ell \leq q\}$, where \mathbf{P}_i is the cellular composition vector of sample i , $\boldsymbol{\mu}_j$ is the baseline methylation vector of CpG site j , $\mathbf{B}_\ell^{(j)}$ is the phenotype ℓ effect vector on CpG site j , σ_{jk}^2 is the cell-type- k noise variance on CpG site j , and σ_{ej}^2 is the overall noise variance on CpG site j .

The observed data in our study are the methylation matrix $\mathbf{O} = \{O_{ij} : 1 \leq i \leq n, 1 \leq j \leq m\}$ and the covariate matrix $\mathbf{X} = (\mathbf{x}_1, \dots, \mathbf{x}_\ell, \dots, \mathbf{x}_q)$, where \mathbf{x}_ℓ is the column vector indicating phenotype- ℓ for the n samples. The observed likelihood function $L(\Theta|\mathbf{O}) = \prod_{i=1}^n \prod_{j=1}^m N(O_{ij} : \mathbf{P}_i^T \boldsymbol{\mu}_j + \sum_{\ell=1}^q x_{i\ell} \mathbf{P}_i^T \mathbf{B}_\ell^{(j)}, \sum_{k=1}^K \sigma_{jk}^2 P_{ik}^2 + \sigma_{ej}^2)$, where $N(O : \eta, \tau^2)$ indicates the normal density with mean η and variance τ^2 at value O .

We further define $\mathbf{1}_K = (1, 1, \dots, 1)^T$ as a K -dimension column vector with all entries being one, an n by K matrix \mathbf{J}_1 as $\mathbf{1}_n \mathbf{1}_K^T$, and an n by K matrix \mathbf{J}_{x_ℓ} as $\mathbf{x}_\ell \mathbf{1}_K^T$ for each $1 \leq \ell \leq q$. We use \odot to represent the entry-wise matrix product for two matrices \mathbf{M} and \mathbf{N} with the same dimension, i.e., $(\mathbf{M} \odot \mathbf{N})_{ij} := \mathbf{M}_{ij} \mathbf{N}_{ij}$.

Theorem 1. *If (a) for each cell type k , there exists a CpG site r_k such that $\mathbf{B}_\ell^{(r_k)} = 0$ for any phenotype ℓ and $\mu_{r_k k} = 1$ while $\mu_{r_k k'} = 0$ for $k' \neq k$, and (b) the cellular compositions \mathbf{P} satisfies that $\text{rank}((\mathbf{J}_1 \odot \mathbf{P}^T, \mathbf{J}_{x_1} \odot \mathbf{P}^T, \dots, \mathbf{J}_{x_\ell} \odot \mathbf{P}^T, \dots, \mathbf{J}_{x_q} \odot \mathbf{P}^T)) = (q+1)K$ and $\text{rank}((\mathbf{1}_n, \mathbf{P}^T) \odot (\mathbf{1}_n, \mathbf{P}^T)) = K+1$, then the HIRE model is identifiable. In other words, $L(\Theta|\mathbf{O}) = L(\tilde{\Theta}|\mathbf{O})$ for any \mathbf{O} implies $\Theta = \tilde{\Theta}$.*

Proof: First, by integrating out all \mathbf{O} elements except O_{ij} , $L(\Theta|\mathbf{O}) = L(\tilde{\Theta}|\mathbf{O})$ implies $N(O_{ij} : \mathbf{P}_i^T \boldsymbol{\mu}_j + \sum_{\ell=1}^q x_{i\ell} \mathbf{P}_i^T \mathbf{B}_\ell^{(j)}, \sum_{k=1}^K \sigma_{jk}^2 P_{ki}^2 + \sigma_{ej}^2) = N(O_{ij} : \tilde{\mathbf{P}}_i^T \tilde{\boldsymbol{\mu}}_j + \sum_{\ell=1}^q x_{i\ell} \tilde{\mathbf{P}}_i^T \tilde{\mathbf{B}}_\ell^{(j)}, \sum_{k=1}^K \tilde{\sigma}_{jk}^2 \tilde{P}_{ki}^2 + \tilde{\sigma}_{ej}^2)$. Because the univariate normal distribution is

identifiable, we have

$$\mathbf{P}_i^T \boldsymbol{\mu}_j + \sum_{\ell=1}^q x_{i\ell} \mathbf{P}_i^T \mathbf{B}_\ell^{(j)} = \tilde{\mathbf{P}}_i^T \tilde{\boldsymbol{\mu}}_j + \sum_{\ell=1}^q x_{i\ell} \tilde{\mathbf{P}}_i^T \tilde{\mathbf{B}}_\ell^{(j)}, \quad (2.1)$$

$$\sum_{k=1}^K \sigma_{jk}^2 P_{ki}^2 + \sigma_{\epsilon j}^2 = \sum_{k=1}^K \tilde{\sigma}_{jk}^2 \tilde{P}_{ki}^2 + \tilde{\sigma}_{\epsilon j}^2. \quad (2.2)$$

Taking $j = r_k$ in Equation 2.1, we have $LHS = \mathbf{P}_i^T \boldsymbol{\mu}_{r_k} + \sum_{\ell=1}^q x_{i\ell} \mathbf{P}_i^T \mathbf{B}_\ell^{(r_k)} = \mathbf{P}_i^T \boldsymbol{\mu}_{r_k} = 0 + P_{ki} \cdot 1 + 0 = P_{ki}$ and similarly $RHS = \tilde{P}_{ki}$, so $P_{ki} = \tilde{P}_{ki}$, which holds for any i and k . Hence, we obtain $\mathbf{P} = \tilde{\mathbf{P}}$. Next, we rewrite Equation 2.1 into a matrix form.

$$(\mathbf{P}_i^T, x_{i1} \mathbf{P}_i^T, \dots, x_{iq} \mathbf{P}_i^T) \begin{pmatrix} \boldsymbol{\mu}_j \\ \mathbf{B}_1^{(j)} \\ \vdots \\ \mathbf{B}_q^{(j)} \end{pmatrix} = (\mathbf{P}_i^T, x_{i1} \mathbf{P}_i^T, \dots, x_{iq} \mathbf{P}_i^T) \begin{pmatrix} \tilde{\boldsymbol{\mu}}_j \\ \tilde{\mathbf{B}}_1^{(j)} \\ \vdots \\ \tilde{\mathbf{B}}_q^{(j)} \end{pmatrix}, \quad i = 1, \dots, n.$$

By combining these n equations, it follows that

$$\begin{aligned} & (\mathbf{J}_1 \odot \mathbf{P}^T, \mathbf{J}_{x_1} \odot \mathbf{P}^T, \dots, \mathbf{J}_{x_\ell} \odot \mathbf{P}^T, \dots, \mathbf{J}_{x_q} \odot \mathbf{P}^T) \begin{pmatrix} \boldsymbol{\mu}_j \\ \mathbf{B}_1^{(j)} \\ \vdots \\ \mathbf{B}_q^{(j)} \end{pmatrix} \\ &= (\mathbf{J}_1 \odot \mathbf{P}^T, \mathbf{J}_{x_1} \odot \mathbf{P}^T, \dots, \mathbf{J}_{x_\ell} \odot \mathbf{P}^T, \dots, \mathbf{J}_{x_q} \odot \mathbf{P}^T) \begin{pmatrix} \tilde{\boldsymbol{\mu}}_j \\ \tilde{\mathbf{B}}_1^{(j)} \\ \vdots \\ \tilde{\mathbf{B}}_q^{(j)} \end{pmatrix}. \end{aligned} \quad (2.3)$$

Because the rank of $A := (\mathbf{J}_1 \odot \mathbf{P}^T, \mathbf{J}_{x_1} \odot \mathbf{P}^T, \dots, \mathbf{J}_{x_\ell} \odot \mathbf{P}^T, \dots, \mathbf{J}_{x_q} \odot \mathbf{P}^T)$ is $(q+1)K$ (full column rank), A has a left inverse A^{-1} . Multiplying Equation 2.3 by A^{-1} from the left on both sides, we obtain $\boldsymbol{\mu}_j = \tilde{\boldsymbol{\mu}}_j$ and $\mathbf{B}_\ell^{(j)} = \tilde{\mathbf{B}}_\ell^{(j)}$ for $1 \leq \ell \leq q$. Therefore, we have $\boldsymbol{\mu} = \tilde{\boldsymbol{\mu}}$, $\mathbf{B} = \tilde{\mathbf{B}}$.

In addition, because Equation 2.2 holds for any i , we can also rewrite it into a matrix form.

$$\begin{pmatrix} 1 & P_{11}^2 & \dots & P_{K1}^2 \\ 1 & P_{12}^2 & \dots & P_{K2}^2 \\ \vdots & \vdots & \ddots & \vdots \\ 1 & P_{1n}^2 & \dots & P_{Kn}^2 \end{pmatrix} \begin{pmatrix} \sigma_{\epsilon j}^2 \\ \sigma_{j1}^2 \\ \vdots \\ \sigma_{jK}^2 \end{pmatrix} = \begin{pmatrix} 1 & P_{11}^2 & \dots & P_{K1}^2 \\ 1 & P_{12}^2 & \dots & P_{K2}^2 \\ \vdots & \vdots & \ddots & \vdots \\ 1 & P_{1n}^2 & \dots & P_{Kn}^2 \end{pmatrix} \begin{pmatrix} \tilde{\sigma}_{\epsilon j}^2 \\ \tilde{\sigma}_{j1}^2 \\ \vdots \\ \tilde{\sigma}_{jK}^2 \end{pmatrix}$$

The left matrix is equal to $(\mathbf{1}_n, \mathbf{P}^T) \odot (\mathbf{1}_n, \mathbf{P}^T)$ which has a full column rank, therefore,

it has a left inverse. Consequently, $\sigma_{\epsilon_j}^2 = \tilde{\sigma}_{\epsilon_j}^2$ and $\sigma_{j_k}^2 = \tilde{\sigma}_{j_k}^2$. As a result, $\Theta = \tilde{\Theta}$, and we have proven the identifiability of HIRE. \square

Conditions (a) and (b) are easily met for DNA methylation data. Condition (a) requires that for each cell type k , there exists a CpG site which is not associated with any phenotype and is only methylated in cell type k but not methylated in any other cell type. Given the 450K CpG sites assayed by the microarray, we can expect that such CpG sites are not absent at all. Moreover, condition (a) can also be relaxed to “for each cell type k , there exists a CpG site r_k such that $\mathbf{B}_\ell^{(r_k)} = 0$ for any phenotype ℓ and $\mu_{r_k k} = 1$ while $\mu_{r_k k'} = 0$ for $k' \neq k$ or there exists a CpG site r_k such that $\mathbf{B}_\ell^{(r_k)} = 0$ for any phenotype ℓ and $\mu_{r_k k} = 0$ while $\mu_{r_k k'} = 1$ for $k' \neq k$ ”. The proof follows similarly.

For condition (b), intuitively, the rank requirement of $(\mathbf{1}_n, \mathbf{P}^T) \odot (\mathbf{1}_n, \mathbf{P}^T)$ asks the cellular compositions to vary across subjects, which guards against the case under which all the subjects have the exact same cellular compositions and hence no cell type deconvolution is possible; the rank requirement on $(\mathbf{J}_1 \odot \mathbf{P}^T, \mathbf{J}_{x_1} \odot \mathbf{P}^T, \dots, \mathbf{J}_{x_\ell} \odot \mathbf{P}^T, \dots, \mathbf{J}_{x_q} \odot \mathbf{P}^T)$ is the same requirement as those in a standard linear regression, which requires that there are no collinearity among the covariates. Because the sample size n is much larger than the underlying cell type number K and the phenotype number q , the two rank requirements can commonly be satisfied in reality.

In summary, here we provide a sufficient condition for the identifiability of HIRE and explain why the conditions are easily met in real DNA methylation data analysis. Finding the necessary and sufficient conditions for the identifiability of the HIRE model is a theoretically interesting and challenging problem. We leave it for future research. We thank Reviewer 4 for his or her kind considerations.

2. The authors’ evaluations seem to focus on the p values and the testing of the differential CpG’s, which is certainly important. However in order to have full confidence that these p values are valid, we need to also know that the other estimated parameters (μ , P , B) are also correct. The authors should show how their estimates for these parameters compare with the ground truth in their simulations and with the best known experimental measurements in the real data.

We appreciate the great suggestion by the reviewer.

We actually have investigated and displayed the estimation performance of HIRE for the cellular compositions \mathbf{P} and the phenotype effects \mathbf{B} in the previous revision and reply letter. Supplementary Figures 27-62 show the correlation plots between estimated $\hat{\mathbf{P}}$ and the true \mathbf{P} , the average bias plots of cellular compositions, and the average bias plots of case/control effects as well as the age effects in 18 simulation settings: the cell type number $K = 3, 5, 7$, the sample size $n = 180, 300, 600$, and the two “true alternatives” and “true nulls” cases.

Following Reviewer 4’s advice, this time we added the bias plots for $\boldsymbol{\mu}$ in each simulation setting to evaluate the performance in estimating the baseline methylation profiles. All the results of evaluating estimation accuracy of $\boldsymbol{\mu}$ have been added to the supplementary document. Please see the following figures for an example, where $K = 5$ and $n = 600$ for both the “true nulls” and the “true alternatives” settings. In all the cases, HIRE achieved good performance in estimating $\boldsymbol{\mu}$, \mathbf{P} , \mathbf{B} .

Figure R1: This figure is the same as Supplementary Figure 37. In the “true nulls” case with $K = 5$ and $n = 600$, (a-e) the scatter plots for the estimated cell proportions by HIRE and the underlying truth for (a) cell type 1; (b) cell type 2; (c) cell type 3; (d) cell type 4; and (e) cell type 5. (f) The average biasplot for the cellular compositions, where the y-axis is $\frac{1}{K} \sum_{k=1}^K |\hat{p}_{ki} - p_{ki}|$ and the x-axis is the sample index i for $i = 1, \dots, n$.

Figure R2: This figure is the same as Supplementary Figure 38. In the “true nulls” case with $K = 5$ and $n = 600$, the average biasplots for (a) the case/control effects, (b) the age effects, and (c) the methylation profiles. In (a), the y axis is the $\frac{1}{K} \sum_{k=1}^K |\hat{\beta}_{jk1} - \beta_{jk1}|$ and the x axis is the CpG site index j for $j = 1, \dots, m$. In (b), the y axis is the $\frac{1}{K} \sum_{k=1}^K |\hat{\beta}_{jk2} - \beta_{jk2}|$ and the x axis is the CpG site index j for $j = 1, \dots, m$. In (c), the y-axis is the $\frac{1}{K} \sum_{k=1}^K |\hat{\mu}_{jk} - \mu_{jk}|$ and the x-axis is the CpG site index j for $j = 1, \dots, m$.

Figure R3: This figure is the same as Supplementary Figure 55. In the “true alternatives” case with $K = 5$ and $n = 600$, (a-e) the scatter plots for the estimated cell proportions by HIRE and the underlying truth for (a) cell type 1; (b) cell type 2; (c) cell type 3; (d) cell type 4; and (e) cell type 5. (f) The average biasplot for the cellular compositions, where the y-axis is $\frac{1}{K} \sum_{k=1}^K |\hat{p}_{ki} - p_{ki}|$ and the x-axis is the sample index i for $i = 1, \dots, n$.

Figure R4: This figure is the same as Supplementary Figure 56. In the “true alternatives” case with $K = 5$ and $n = 600$, the average biasplots for (a) the case/control effects, (b) the age effects, and (c) the methylation profiles. In (a), the y axis is the $\frac{1}{K} \sum_{k=1}^K |\hat{\beta}_{jk1} - \beta_{jk1}|$ and the x axis is the CpG site index j for $j = 1, \dots, m$. In (b), the y axis is the $\frac{1}{K} \sum_{k=1}^K |\hat{\beta}_{jk2} - \beta_{jk2}|$ and the x axis is the CpG site index j for $j = 1, \dots, m$. In (c), the y-axis is the $\frac{1}{K} \sum_{k=1}^K |\hat{\mu}_{jk} - \mu_{jk}|$ and the x-axis is the CpG site index j for $j = 1, \dots, m$.

3. It'd be useful to see more systematic analysis of the robustness of the model when there is model mismatch. This is especially important given the model complexity.

We thank the reviewer for the invaluable comments.

In the HIRE model, we assume that the nearby CpG sites are independent. In the updated manuscript, we investigate the performance of HIRE when such a model assumption is violated and there exist dependences among nearby CpG sites. Specifically, we assume that every 50 consecutive CpG sites belongs to a block. For CpG sites within the same block, their random noises ϵ follow a multivariate normal distribution with mean zero and 50×50 covariance matrix Σ , and Σ 's corresponding correlation matrix has its (i, j) entry equal to $\rho^{|i-j|}$. We vary ρ to 0.8, 0.6 and 0.4. Comparing Tables R1, R2, R3 and Table R4, we can see that even when there exist strong correlations between nearby CpG sites, HIRE still provides good performances in controlling the false positive rate (FPR) and detecting the risk-CpG sites under the model misspecification setting.

Table R1: This table is the same as Supplementary Table 1. The FPRs and TPRs of HIRE in different sample sizes and cell type numbers when nearby CpG sites are correlated with $\rho = 0.8$.

cell type number		$n = 180$	$n = 300$	$n = 600$
$K = 3$	FPR	0.13%	0.04%	0.03%
	TPR	90%	93.33%	100%
$K = 5$	FPR	0.03%	0.01%	0
	TPR	62%	92%	92%
$K = 7$	FPR	0%	0	0.01%
	TPR	35%	63.33%	80%

Table R2: This table is the same as Supplementary Table 2. The FPRs and TPRs of HIRE in different sample sizes and cell type numbers when nearby CpG sites are correlated with $\rho = 0.6$.

cell type number		$n = 180$	$n = 300$	$n = 600$
$K = 3$	FPR	0.10%	0.01%	0.04%
	TPR	90%	93.33%	100%
$K = 5$	FPR	0.01%	0.09%	0
	TPR	56%	86%	92%
$K = 7$	FPR	0	0	0.02%
	TPR	43.33%	63.33%	76.67%

Table R3: This table is the same as Supplementary Table 3. The FPRs and TPRs of HIRE in different sample sizes and cell type numbers when nearby CpG sites are correlated with $\rho = 0.4$.

cell type number		$n = 180$	$n = 300$	$n = 600$
$K = 3$	FPR	0	0.02%	0.03%
	TPR	90%	93.33%	100%
$K = 5$	FPR	0.01%	0.07%	0
	TPR	58%	82%	90%
$K = 7$	FPR	0	0	0.01%
	TPR	43.33%	68.33%	81.67%

Table R4: This table is the same as Supplementary Table 4. The FPRs and TPRs of HIRE in different sample sizes and cell type numbers when nearby CpG sites are uncorrelated (extracted from Table 1 in the manuscript).

cell type number		$n = 180$	$n = 300$	$n = 600$
$K = 3$	FPR	0.01%	0.00%	0.03%
	TPR	98.67%	96.67%	100%
$K = 5$	FPR	0.00%	0.01%	0.01%
	TPR	66%	89.6%	98.4%
$K = 7$	FPR	0	0.00%	0.04%
	TPR	43%	63.33%	82.67%

References

- Houseman, E. A., M. L. Kile, D. C. Christiani, T. A. Ince, K. T. Kelsey, and C. J. Marsit (2016). Reference-free deconvolution of DNA methylation data and mediation by cell composition effects. *BMC Bioinformatics* 17(1), 259.
- Rahmani, E., R. Schweiger, L. Shenhav, T. Wingert, I. Hofer, E. Gabel, E. Eskin, and E. Halperin (2018). BayesCCE: a Bayesian framework for estimating cell-type composition from DNA methylation without the need for methylation reference. *Genome Biology* 19(1), 141.

REVIEWERS' COMMENTS:

Reviewer #4 (Remarks to the Author):

After going over the authors' response and revision, my main concerns on the manuscript have been addressed.

Reviewer 4

After going over the authors' response and revision, my main concerns on the manuscript have been addressed.

We thank the reviewer for the positive comment.